# Genetic diversity of Collaborative Cross mice implicates FFAR3 as a target for ILC2 anti-inflammatory reprogramming

Mark Rusznak [1,2] ✉, Shinji Toki [1], Yajing Hao[3], Marc J. C. Todd[1], Liddy Malone[1], Julia F. Goodhead[1], Catherine DuPuy[1], Weisong Zhou[1], Dominique Babin[1], Christian M. Warren[4], Masako Abney[1], Matthew T. Stier [1], Christopher M. Thomas[1], Jing Li[2,5], Justin Jacobse [5], Andrew P. Pahnke [2], Mark I. Petrovic[1], Jacqueline-Yvonne Cephus[1], Shelby N. Kuehnle [1], M. Wade Calcutt[6], Allison E. Norlander[7], Fang Yan [8,9], Jeremy A. Goettel[2,5], Darla R. Miller[10], Rachel M. Lynch [10], Daniel P. Cook [11], Dawn C. Newcomb[1,2], Fei Zou [3,10] & R. Stokes Peebles Jr [1,2,4] ✉

Pulmonary group 2 innate lymphoid cells (ILC2s) are key drivers of Type 2 inflammation in diseases like asthma, yet the molecular mechanisms regulating their function are incompletely understood. Using the genetically diverse Collaborative Cross (CC) mouse panel, we mapped a quantitative trait locus (QTL) that governs ILC2 prevalence in the lung after aeroallergen exposure. This QTL induces a large population of ILC2s in the lung that are resistant to activation and have diminished Type 2 effector function. We identified free-fatty acid receptor 3 (*Ffar3*) as a gene responsible for this effect and demonstrated that FFAR3 signaling reprograms ILC2s to an anti-inflammatory state by promoting their survival, reducing Type 2 cytokine production, and enhancing IL-10 expression. This anti-inflammatory state is dependent on IL-2 signaling, is characterized by decreased ST2 expression, and is distinct from previously described IL-10-producing ILC2 phenotypes. FFAR3-dependent reprogramming is mediated by epidermal growth factor receptor (EGFR) upregulation, and FFAR3's anti-inflammatory effect is partially conserved in human ILC2s.

Asthma is one of the most prevalent chronic diseases in the United States. A myriad of factors including the rising rates of allergic diseases, modern Western diets and lifestyles, urbanization, and antibiotic usage all contribute to its development[1]. In addition to environmental contributors, asthma has numerous genetic determinants of susceptibility, highlighting the complexity of the immunologic mechanisms that govern disease pathogenesis[2]. Investigating the specific molecular regulators of asthma has allowed for the

[1]Division of Allergy, Pulmonary, and Critical Care Medicine, Department of Medicine, Vanderbilt University Medical Center, Nashville, TN, USA. [2]Department of Pathology, Microbiology, and Immunology, Vanderbilt University Medical Center, Nashville, TN, USA. [3]Department of Biostatistics, University of North Carolina at Chapel Hill, Chapel Hill, NC, USA. [4]United States Department of Veterans Affairs, Tennessee Valley Healthcare System, Nashville, TN, USA. [5]Department of Medicine, Division of Gastroenterology, Hepatology and Nutrition, Vanderbilt University Medical Center, Nashville, TN, USA. [6]Mass Spectrometry Research Center and Department of Biochemistry, Vanderbilt University School of Medicine, Nashville, TN, USA. [7]Anatomy, Cell Biology, and Physiology, Indiana University School of Medicine, Indianapolis, IN, USA. [8]Department of Cell and Developmental Biology, Vanderbilt University, Nashville, TN, USA. [9]Department of Pediatrics, Vanderbilt University Medical Center, Nashville, TN, USA. [10]Department of Genetics, University of North Carolina at Chapel Hill, Chapel Hill, NC, USA. [11]Department of Internal Medicine, University of Iowa, Iowa City, IA, USA. ✉e-mail: markrusznak@gmail.com; stokes.peebles@vanderbilt.edu

development of targeted biological therapeutics, which have proven effective in treating poorly-controlled and severe asthma[3–5]. The heterogeneity of the disease and the recent success of these targeted therapies demonstrates the vital importance of uncovering new molecular regulators of asthma to expand our therapeutic target repertoire.

Mouse research is commonly used to identify disease targets, but most of this research relies on a small number of traditional inbred strains. While restricting work to a few strains has practical benefits, research conducted in this manner ignores the phenotypic heterogeneity that exists within the mouse species and potentially limits the breadth of discovery. For a heterogeneous disease like asthma in the diverse human population, it is possible that the contributions of important genetic regulators are not appropriately represented in just a few mouse genotypes. We endeavored to implicate disease-relevant targets that may have been missed due to this limited sampling by harnessing the genetic diversity in the mouse species with the Collaborative Cross (CC) recombinant inbred (RI) mouse panel, one of the most ambitious mouse genetics projects undertaken in recent history.

The CC is an RI panel created from 8 founder strains, 5 classical laboratory strains (A/J, C57BL/6J, 129S1/SvImJ, NOD/LtJ, NZO/HiLtJ) and 3 wild derived strains (CAST/EiJ, PWK/PhJ, and WSB/EiJ). Owing to these diverse founder strains, the panel's breeding architecture encompasses roughly 90% of known genetic variation in laboratory mice[6]. These founders were crossed to yield the CC RI strains. The genotype of each CC RI strain is unique and contains genetic character from each of the 8 founders. The genomes of these recombinant strains have been reconstructed as probabilistic mosaics of founder haplotypes, meaning that one can know from which of the 8 founders a specific position on the genome was inherited[7,8]. The vast genetic diversity in the population and high-marker density makes the CC an excellent tool for the high resolution of mapping complex traits[9–13].

In this study, we performed a quantitative trait locus (QTL) mapping experiment to leverage the genetic diversity of the CC to identify disease-relevant targets not previously described. QTL mapping studies involve quantifying phenotypic differences among a large set of progeny and use the segregation of genetic markers in those progeny to identify genomic loci that associate with the measured phenotype[14]. In principle, this approach can experimentally associate genomic regions as regulators of any complex trait, provided that the phenotypic trait being observed is quantifiable and continuous (i.e. not qualitative or discrete). In addition, the success of a QTL mapping study is determined by the density of genomic markers used, the number of genetically distinct progeny, the phenotypic differences across these progeny, and the effect sizes of the genes regulating the trait[15,16]. To uncover genetic regulators of asthma pathogenesis with the QTL method in the CC, we selected a quantitative phenotype relating to group 2 innate lymphoid cells (ILC2s).

The adaptive immune system had traditionally been considered primarily responsible for the pathologic Type 2 inflammation that characterizes many variants of asthma. However, this adaptive immunity-centric view was challenged with the first descriptions of group 2 innate lymphoid cells (ILC2s) in 2010[17–19]. ILC2s are tissue resident immune cells derived from the common lymphoid progenitor that are capable of secreting large quantities of Type 2 cytokines, namely IL-5 and IL-13, which in turn act to propagate eosinophil survival and goblet cell mucin production, respectively. ILC2s do not participate in antigen recognition for activation; instead, they are activated primarily by alarmin cytokines like IL-33, thymic stromal lymphopoietin (TSLP), and IL-25 released by the airway epithelial cells in response to aeroallergens[20]. ILC2s play a pivotal role in the rapid induction of inflammation seen in allergen-exacerbated asthma[21,22]. Many key regulators of ILC2 biology, including secreted factors, cell surface receptors, and transcription factors have been extensively described and associated with asthma pathogenesis in humans[23–26]. For

this reason, ILC2s represent a high yield cell type where the discovery of novel regulators has a high likelihood of identifying a target with real therapeutic potential for asthma.

In this study, we used the diversity of the CC to identify genes that are regulators of ILC2 function. To do this, we mapped a locus that associated with the prevalence of ILC2s in the lung after aeroallergen challenge, a quantitative and continuous phenotype that is a function of ILC2 activation. We demonstrate a successful attempt at mapping a QTL relating to ILC2 prevalence and effector function using the CC, allowing us to establish the role of a gene in ILC2 biology and asthma pathogenesis.

## Results
### Different strains of the CC have different ILC2 responses to aeroallergen challenge
In this study, we use the *Alternaria alternata* fungal extract (*Alt* Ex) airway challenge model to induce the Type 2 inflammation that mirrors what would occur during allergen-exacerbated asthma[27–29]. This *Alt* Ex contains the *Alt a 1* fungal protein antigen, which has been associated with asthma exacerbations[30,31]. Administration of *Alt* Ex into the lungs rapidly induces ILC2 activation[32], and the short 4-day course precludes the involvement of the adaptive immune system. The precise quantitative phenotype that we planned to measure in the CC strains was ILC2 prevalence in the lung after 4 days of *Alt* Ex challenge. ILC2 prevalence was measured as the percentage of all live cells in the lung that were ILC2s, allowing for comparisons between strains that have differences in total lung cells (Fig. 1A).

We obtained 48 unique CC founder and recombinant strains and subjected each to our *Alt* Ex model. Whole lungs were harvested from the mice on day four and quantified by flow cytometry (Fig. 1B). ILC2s were identified according to the gating strategy defined in Supplementary Fig. 1. There existed a wide range of ILC2 prevalence among the strains of the CC, suggesting that QTL mapping would likely be successful. Strain differences persisted when quantifying ILC2s after challenge with total cell numbers in the lung (Supplementary Fig. 2A), and we observed comparable variability in baseline ILC2 prevalence and number in the CC mice without challenge (Supplementary Fig. 2B & C). Expansion ratios for each strain with naïve and challenged ILC2 quantification data are presented in Supplementary Table 1. The vast phenotypic differences among the tested strains confirm the broad genetic diversity within the CC panel and broadly highlight the importance of mouse strain background when considering any experimental model.

### QTL mapping with CC strains identifies a 3.43 Mb locus on mouse chromosome 7 that associates with ILC2 prevalence after aeroallergen challenge
We used the phenotypic data gathered in Fig. 1B, in conjunction with CC haplotype data from the Most Recent Common Ancestor 8-State haplotype files[33,34], to perform the QTL mapping analysis according to previously described methods[8]. The basics of the analysis involve grouping each of the recombinant strains based on founder diplotypes at a specific marker. Whether or not a difference in the quantitative phenotype exists between the groups is then determined by an unconstrained regression. Phenotypic differences based on founder diplotype at a marker suggest that the marker influences the quantitative phenotype of interest. This process is repeated across all markers and the resultant QTL map defines loci that regulate the phenotype, with statistical significance thresholds determined by permutation test.

QTL mapping returned a significant locus on mouse chromosome 7 that regulated ILC2 prevalence after aeroallergen challenge (Fig. 2A). This QTL spanned 3.43 megabases starting from position 30.538129 and contained 72 protein coding genes, fully detailed in Table 1. It was the only peak in our analysis to reach significance, and it exceeded the

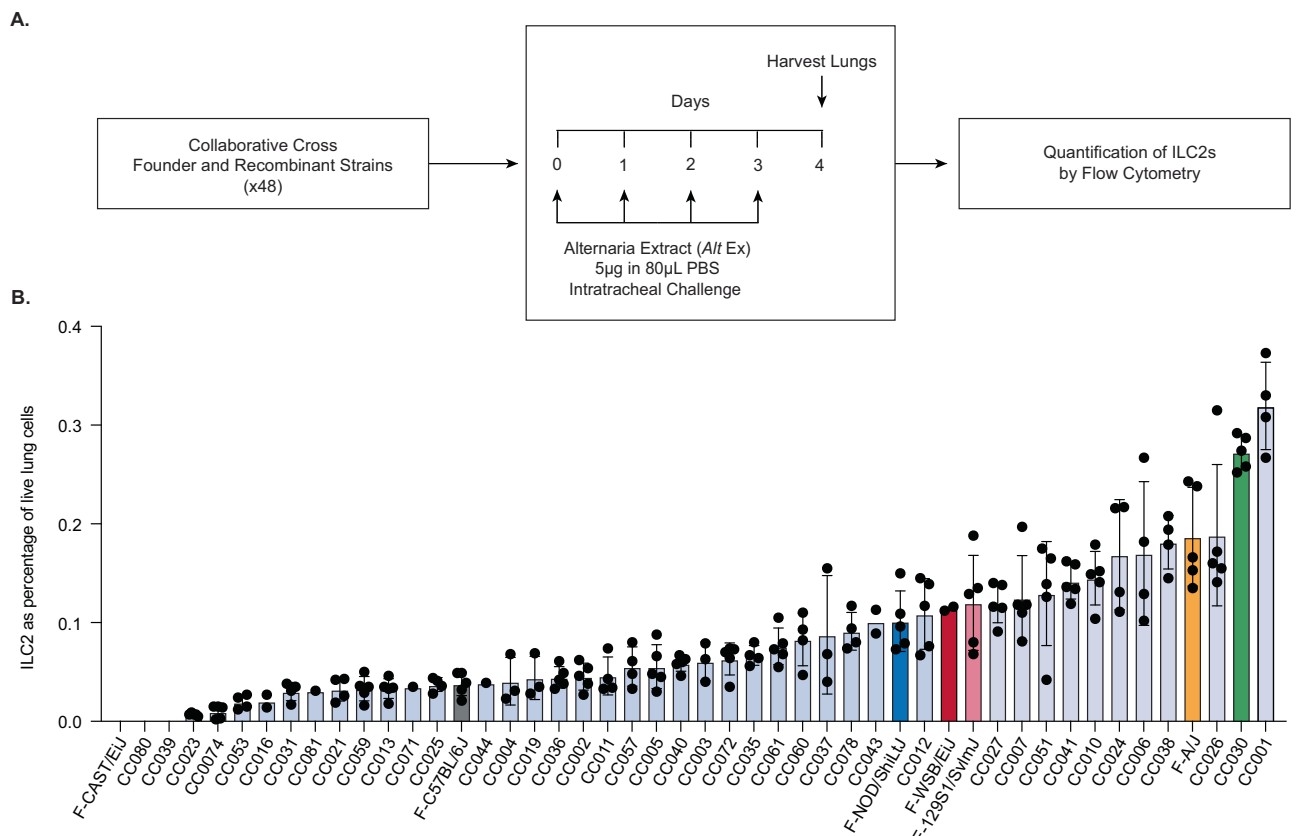

**Fig. 1 | Different strains of the CC have different ILC2 responses to aeroallergen challenge. A** Diagram of *Alternaria* extract challenge model. **B** ILC2s from different CC strains quantified as a percentage of live cells in the lung after challenge. Strains without bars indicate that all mice died during challenge. Colored bars represent founders and strains of interest. Average n for strains tested = 4.02, exact n for each strain is available in the Source Data file. Data is representative of one experiment.

95% permutation threshold. Henceforth, this locus will be referred to in this manuscript as the Ch.7 ILC2 Locus. The Ch.7 ILC2 Locus did not appear to contain any previously described regulators of ILC2 function, suggesting that the QTL had identified an uncharacterized target. Founder haplotype effect analysis reveals that the presence of the QTL is driven almost exclusively by the CAST/EiJ founder haplotype, meaning that recombinant strains inheriting this locus from the CAST/EiJ founder have elevated ILC2 numbers after aeroallergen challenge compared to recombinant strains that inherited the locus from all other founders (Fig. 2B). In our initial phenotyping study presented in Fig. 1B, all CAST/EiJ mice died during the *Alt* Ex challenge, preventing the quantification of ILC2s in this founder strain.

We sought to identify the specific gene or set of genes within the Ch.7 ILC2 Locus that is responsible for the QTL effect, and therefore, a regulator of ILC2 biology. Our following experimental approach had two aims: 1.) fully characterize the nature of the ILC2-related phenotype that the Ch.7 ILC2 Locus regulates, and 2.) narrow the list of candidate genes in the locus to identify and validate a specific ILC2 target. The best approach to tackle both aims was a direct comparison of two CC strains with opposite phenotypic extremes from Fig. 1B. We identified a strain with high ILC2 prevalence after challenge that inherited its locus from CAST/EiJ and a strain with low ILC2 prevalence after challenge that inherited its locus from a negative contributing founder strain. We hypothesized that: 1.) the differences between the ILC2s in these high and low strains more completely describe the phenotype governed by the QTL, as it is likely that the QTL's physiologic effects extend beyond simply the quantity of ILC2s after challenge, and 2.) a gene in the Ch.7 ILC2 Locus that influences ILC2 prevalence after challenge exhibits expression differences between a strain with very high ILC2 prevalence and a CAST/EiJ haplotype at the

Ch.7 ILC2 Locus and a strain with low ILC2 prevalence and a negatively contributing founder haplotype at the Ch.7 ILC2 Locus.

## CC030/GeniUnc mice have more numerous and transcriptionally distinct ILC2s at baseline compared to C57BL/6J

The two strains that we utilized for the high and low phenotypic comparison were the CC030/GeniUnc (henceforth CC030) recombinant strain and the C57BL/6J founder strain. We selected CC030 because it presented with the second highest ILC2 prevalence after challenge of the tested strains, and it possessed 100% CAST/EiJ lineage for the entirety of the QTL. The only strain with a higher prevalence was CC001/Unc, but this strain did not have complete haplotype homogeneity for the Ch.7 ILC2 Locus[34]. We selected C57BL/6J as our low phenotype reference strain because of its low ILC2 prevalence phenotype, locus haplotype homogeneity (inherent to the strain being a founder), and negative haplotype contribution (Fig. 2B).

We first sought to more completely characterize the phenotype that is regulated by the Ch.7 ILC2 Locus. Unchallenged CC030 mice had significantly higher ILC2 prevalence and total ILC2 numbers in the lung compared to unchallenged C57BL/6J mice (Fig. 3A–C), suggesting that the ILC2 prevalence phenotype regulated by the Ch.7 ILC2 Locus is present at baseline and is not a function of increased proliferation in response to aeroallergen challenge. To assess whether this elevation in ILC2 percentage in CC030 vs. C57BL/6J was: 1.) restricted to the lung and 2.) dependent on female sex, we quantified ILC2s in the lung, bone marrow, colon, and gonadal adipose tissue from female and male CC030 and C57BL/6J mice (Supplementary Fig. 3A). ILC2 prevalence and total numbers of ILC2s were elevated in the lungs of both male and female CC030 mice compared to their respective sexes of C57BL/6J mice. This pattern of CC030 mice exhibiting elevated ILC2s compared to C57BL/6J

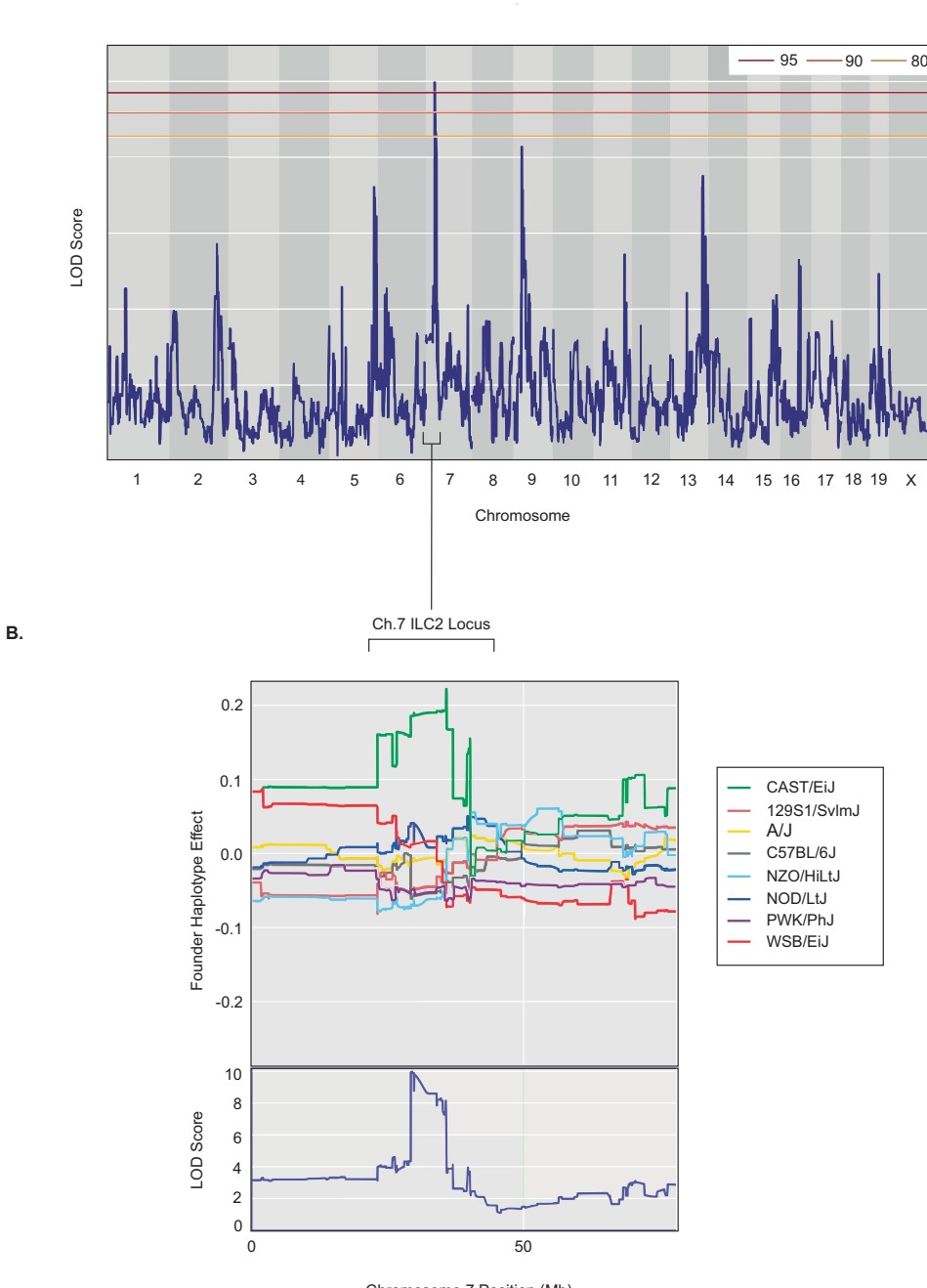

**Fig. 2 | QTL mapping with CC strains identifies a 3.43 Mb locus on mouse chromosome 7 that associates with ILC2 prevalence after aeroallergen challenge. A** QTL map of ILC2 prevalence after *Alt* Ex challenge. LOD = Logarithm of Odds. Horizontal lines represent significance thresholds determined by permutation testing (80%, 90%, and 95% from bottom to top). No peaks other than our QTL of interest reached even the lowest significance threshold. **B** The haplotype effect of each founder in generating the significant QTL described in (A), termed Ch.7 ILC2 Locus. The bottom panel represents a zoomed-in view of the Ch.7 ILC2 Locus presented in (**A**) detailing chromosomal position.

in a sex-independent manner was not observed in the other tissues, suggesting that the quantitative phenotype mapped with the Ch.7 ILC2 Locus is unique to the lung. Despite being more numerous, CC030 ILC2s exhibit less than half the expression of ST2, the receptor for ILC2's primary activating cytokine IL-33, compared to C57BL/6J ILC2s (Fig. 3D and E). The decrease of ST2 expression in the CC030 ILC2s suggests that they may be less capable of responding to IL-33 alarmin signals, implying decreased IL-33-mediated inflammatory potential.

Inspired by the differences in ST2 expression, we endeavored to more broadly characterize the differences between the ILC2s of these

strains. We performed bulk RNA-sequencing on naïve CC030 and C57BL/6J ILC2s sorted from the lungs via flow cytometry (Fig. 3F). Although these cells were isolated using identical ILC2 markers, they were drastically different by their gene expression profiles. This difference was appreciated in the quantity of differentially expressed genes (Fig. 3G), sample distance matrix (Fig. 3H), and principal component analysis (PCA) (Fig. 3I). The full list of differentially expressed genes is provided in Supplementary Data 1. We then performed gene set enrichment analysis (GSEA) with WikiPathways annotated gene sets to better understand the specific cellular pathways and processes that

**Table 1 | List of 72 protein coding genes within the Ch.7 ILC2 Locus**

| Start | Stop | Strand | Name | Gene ID | Description |
|---|---|---|---|---|---|
| 30.539129 | 30.552438 | – | Proser3 | MGI:2681861 | proline and serine rich 3 |
| 30.552178 | 30.555443 | + | Hspb6 | MGI:2685325 | heat shock protein, alpha-crystallin-related, B6 |
| 30.555441 | 30.559838 | – | Lin37 | MGI:1922910 | lin-37 homolog (C. elegans) |
| 30.561863 | 30.563627 | – | Psenen | MGI:1913590 | presenilin enhancer gamma secretase subunit |
| 30.563301 | 30.565365 | + | U2af1l4 | MGI:2678374 | U2 small nuclear RNA auxiliary factor 1-like 4 |
| 30.563344 | 30.56796 | + | Gm49396 | MGI:6121629 | predicted gene, 49396 |
| 30.565427 | 30.567962 | + | Igflr1 | MGI:3655979 | IGF-like family receptor 1 |
| 30.568855 | 30.588726 | – | Kmt2b | MGI:109565 | lysine (K)-specific methyltransferase 2B |
| 30.58968 | 30.60282 | – | Zbtb32 | MGI:1891838 | zinc finger and BTB domain containing 32 |
| 30.603092 | 30.612847 | – | Upk1a | MGI:98911 | uroplakin 1A |
| 30.616861 | 30.626151 | – | Cox6b1 | MGI:107460 | cytochrome c oxidase, subunit 6B1 |
| 30.633616 | 30.636509 | – | Etv2 | MGI:99253 | ets variant 2 |
| 30.640995 | 30.650317 | – | Rbm42 | MGI:1915285 | RNA binding motif protein 42 |
| 30.653708 | 30.664994 | – | Haus5 | MGI:1919159 | HAUS augmin-like complex, subunit 5 |
| 30.670722 | 30.671605 | – | Pmis2 | MGI:1922177 | PMIS2 transmembrane protein |
| 30.698886 | 30.710298 | + | 2200002J24Rik | MGI:1916397 | RIKEN cDNA 2200002J24 gene |
| 30.712209 | 30.725534 | + | Atp4a | MGI:88113 | ATPase, H + /K+ exchanging, gastric, alpha polypeptide |
| 30.727701 | 30.72954 | – | Tmem147 | MGI:1915011 | transmembrane protein 147 |
| 30.729775 | 30.744791 | – | Gapdhs | MGI:95653 | glyceraldehyde-3-phosphate dehydrogenase, spermatogenic |
| 30.748971 | 30.756134 | + | Sbsn | MGI:2446326 | suprabasin |
| 30.763724 | 30.781066 | + | Dmkn | MGI:1920962 | dermokine |
| 30.787896 | 30.791097 | + | Krtdap | MGI:1928282 | keratinocyte differentiation associated protein |
| 30.818031 | 30.823775 | – | Ffar2 | MGI:2441731 | free fatty acid receptor 2 |
| 30.85433 | 30.856178 | – | Ffar3 | MGI:2685324 | free fatty acid receptor 3 |
| 30.85739 | 30.861564 | – | Ffar1 | MGI:2684079 | free fatty acid receptor 1 |
| 30.865402 | 30.880342 | – | Cd22 | MGI:88322 | CD22 antigen |
| 30.899175 | 30.914921 | – | Mag | MGI:96912 | myelin-associated glycoprotein |
| 30.922372 | 30.924681 | – | Hamp2 | MGI:2153530 | hepcidin antimicrobial peptide 2 |
| 30.942368 | 30.944032 | – | Hamp | MGI:1933533 | hepcidin antimicrobial peptide |
| 30.945248 | 30.957367 | – | Usf2 | MGI:99961 | upstream transcription factor 2 |
| 30.95777 | 30.973469 | – | Lsr | MGI:1927471 | lipolysis stimulated lipoprotein receptor |
| 30.97379 | 30.989726 | + | Fam187b | MGI:1923665 | family with sequence similarity 187, member B |
| 31.032722 | 31.042481 | – | Fxyd5 | MGI:1201785 | FXYD domain-containing ion transport regulator 5 |
| 31.042513 | 31.051467 | – | Fxyd7 | MGI:1889006 | FXYD domain-containing ion transport regulator 7 |
| 31.051678 | 31.057199 | – | Fxyd1 | MGI:1889273 | FXYD domain-containing ion transport regulator 1 |
| 31.058825 | 31.070935 | + | Lgi4 | MGI:2180197 | leucine-rich repeat LGI family, member 4 |
| 31.068172 | 31.076704 | – | Fxyd3 | MGI:107497 | FXYD domain-containing ion transport regulator 3 |
| 31.098725 | 31.115326 | – | Hpn | MGI:1196620 | hepsin |
| 31.116524 | 31.127035 | – | Scn1b | MGI:98247 | sodium channel, voltage-gated, type I, beta |
| 31.130127 | 31.155936 | – | Gramd1a | MGI:105490 | GRAM domain containing 1A |
| 31.263736 | 31.265935 | + | Scgb2b1 | MGI:3782857 | secretoglobin, family 2B, member 1 |
| 31.290508 | 31.304977 | + | Scgb2b2 | MGI:3042579 | secretoglobin, family 2B, member 2 |
| 31.290519 | 31.291821 | – | Scgb1b2 | MGI:1930867 | secretoglobin, family 1B, member 2 |
| 31.359038 | 31.362072 | – | Scgb2b3 | MGI:3782547 | secretoglobin, family 2B, member 3 |
| 31.375592 | 31.376916 | + | Scgb1b3 | MGI:3644233 | secretoglobin, family 1B, member 3 |
| 31.617738 | 31.619526 | – | Scgb2b6 | MGI:5011792 | secretoglobin, family 2B, member 6 |
| 31.703779 | 31.705757 | – | Scgb2b7 | MGI:3782864 | secretoglobin, family 2B, member 7 |
| 31.712665 | 31.71398 | + | Scgb1b7 | MGI:3643480 | secretoglobin, family 1B, member 7 |
| 32.091996 | 32.093939 | – | Scgb2b10 | MGI:3643785 | secretoglobin, family 2B, member 10 |
| 32.100853 | 32.102169 | + | Scgb1b10 | MGI:3782569 | secretoglobin, family 1B, member 10 |
| 32.14624 | 32.148032 | – | Scgb2b30 | MGI:3646832 | secretoglobin, family 2B, member 30 |
| 32.209173 | 32.211145 | – | Scgb2b11 | MGI:3782867 | secretoglobin, family 2B, member 11 |
| 32.22284 | 32.224156 | + | Scgb1b11 | MGI:5581331 | secretoglobin, family 1B, member 11 |
| 32.325286 | 32.327245 | – | Scgb2b12 | MGI:3645177 | secretoglobin, family 2B, member 12 |

**Table 1 (continued) | List of 72 protein coding genes within the Ch.7 ILC2 Locus**

| Start | Stop | Strand | Name | Gene ID | Description |
|-------|------|--------|------|---------|-------------|
| 32.334182 | 32.335501 | + | Scgb1b12 | MGI:3645736 | secretoglobin, family 1B, member 12 |
| 32.441541 | 32.442844 | + | Scgb1b29 | MGI:3644231 | secretoglobin, family 1B, member 29 |
| 32.827683 | 32.829662 | – | Scgb2b15 | MGI:3644904 | secretoglobin, family 2B, member 15 |
| 32.836554 | 32.83787 | + | Scgb1b15 | MGI:3782584 | secretoglobin, family 1B, member 15 |
| 33.054582 | 33.056561 | – | Scgb2b17 | MGI:3647697 | secretoglobin, family 2B, member 17 |
| 33.063449 | 33.064765 | + | Scgb1b17 | MGI:3809666 | secretoglobin, family 1B, member 17 |
| 33.111669 | 33.113458 | – | Scgb2b33 | MGI:3643719 | secretoglobin, family 2B, member 33 |
| 33.171892 | 33.173864 | – | Scgb2b18 | MGI:3782872 | secretoglobin, family 2B, member 18 |
| 33.185568 | 33.186868 | + | Scgb1b18 | MGI:5578742 | secretoglobin, family 1B, member 18 |
| 33.278366 | 33.280341 | – | Scgb2b19 | MGI:3648809 | secretoglobin, family 2B, member 19 |
| 33.287291 | 33.288611 | + | Scgb1b19 | MGI:3646447 | secretoglobin, family 1B, member 19 |
| 33.364343 | 33.366322 | – | Scgb2b20 | MGI:3514009 | secretoglobin, family 2B, member 20 |
| 33.373234 | 33.374559 | + | Scgb1b20 | MGI:3779681 | secretoglobin, family 1B, member 20 |
| 33.518481 | 33.520458 | – | Scgb2b21 | MGI:3648283 | secretoglobin, family 2B, member 21 |
| 33.527376 | 33.528558 | + | Scgb1b21 | MGI:3782875 | secretoglobin, family 1B, member 21 |
| 33.73719 | 33.739312 | – | Scgb2b24 | MGI:2655741 | secretoglobin, family 2B, member 24 |
| 33.743799 | 33.745103 | + | Scgb1b24 | MGI:3649643 | secretoglobin, family 1B, member 24 |
| 33.942997 | 33.94504 | – | Scgb2b26 | MGI:87864 | secretoglobin, family 2B, member 26 |

Details regarding the genes contained within the Ch.7 ILC2 Locus that we mapped in Fig. 2. Start and stop positions indicate the base coordinates for each gene. Strand denotes the directionality of the template strand. Name and Gene ID represent Mouse Genome Informatics (MGI) symbol and code, respectively. The description column provides the MGI official full name.

were different between the ILC2s of these strains. A host of signaling pathways relevant to ILC2 activation and Type 2 inflammatory function were significantly enriched in the C57BL/6J ILC2s. Some of the most striking differences were in TSLP signaling, IL-7 signaling, IL-2 signaling, and the ST2 Signaling/IL-1 structural pathway (Fig. 3J). Although C57BL/6J had only one fourth of the number of upregulated genes compared to CC030, seemingly most of the traditional ILC2 inflammatory pathways were represented. To capture other differences between these cells in an unbiased manner, we performed hierarchical clustering of the top 30 Gene Ontology: Biological Processes pathways most enriched in C57BL/6J and CC030 ILC2s, ranked by normalized enrichment score (Fig. 3K). The most enriched gene sets in C57BL/6J ILC2s cluster together as pathways related to protein translation, mitochondrial complex assembly, RNA processing, ribosome biogenesis, and oxidative phosphorylation. This GSEA result emphasizes that C57BL/6J ILC2s are more transcriptionally, translationally, and metabolically active compared to CC030 ILC2s, suggesting their increased general readiness for activation and the propagation of Type 2 inflammation. In contrast, CC030 ILC2s are most transcriptionally enriched for pathways related to cell-cell interactions, cell membrane ion transport, and detection of chemical stimuli. Taken together, these data highlight the myriad of differences between ILC2s of these two strains (full GSEA lists in Supplementary Data 2-4). CC030 ILC2s have a larger population size at baseline, but their gene expression profiles suggest that they have a significant downregulation of canonical ILC2 inflammatory processes. Despite these differences, we assert that this population in CC030 should still be classified as ILC2s, as they are defined by GATA3+ and show a high percentage of ST2 +, despite a lower ST2 mean fluorescence intensity (MFI) compared to C57BL/6J ILC2s (Fig. 3A, D).

## CC030 mice have a greater number of ILC2s that are less proliferative and less inflammatory than C57BL/6J in vitro and in vivo

We established from our initial phenotyping experiment that CC030 have a very high prevalence of lung ILC2s after *Alt* Ex challenge compared to C57BL/6J. This large ILC2 population is present at baseline in CC030 mice, and our data suggest that CC030 ILC2s have reduced Type 2 inflammatory potential. We hypothesized that despite their large cell numbers, ILC2s in CC030 have abrogated Type 2 effector function compared to C57BL/6J. To test this hypothesis, we isolated ILC2s from naïve lungs of C57BL/6J and CC030 female mice by flow cytometry according to the gating strategy defined in Supplementary Fig. 4A, and we cultured them at equal numbers from both strains with IL-33, TSLP, and IL-2. After 4 days, the number of C57BL/6J ILC2s was significantly increased compared to CC030 ILC2s (Fig. 4A). We incubated ILC2s from both strains with CFSE in an identical culture system and harvested cells after 48 h to assess proliferative differences. C57BL/6J ILC2s showed a greater degree of CFSE dilution (Fig. 4B), indicating greater proliferation (Fig. 4C) and division (Fig. 4D) compared to CC030 ILC2s. These results confirmed that C57BL/6J ILC2s have an increased ability to proliferate in response to identical cytokine stimulation compared to CC030 ILC2s, as was suggested by greater ST2 expression on ILC2s in C57BL/6J. C57BL/6J ILC2s produced more Type 2 cytokines in culture than CC030 ILC2s. We detected significantly greater IL-13 in the supernatant (Fig. 4E) and observed greater IL-13 production on a per-cell basis by intracellular cytokine staining (Fig. 4F) in C57BL/6J ILC2s compared to CC030 ILC2s. These findings are in accordance with our proliferation data, showing that C57BL/6J ILC2s have greater intrinsic inflammatory capability than CC030 ILC2s.

The reduced proliferative capacity of the CC030 ILC2s compared to C57BL/6J prompted us to contemplate the mechanism by which the large ILC2 population exists in CC030 mice. A group previously described that NK cells deficient in Lysine-specific demethylase 6 A (UTX) had an elevated population size in the spleen but exhibited decreased cell fitness. UTX deficient NK cells had decreased apoptotic function, contributing to elevated NK cell numbers[35]. We hypothesized that the large ILC2 population size in CC030 could be attributed to decreased apoptotic activity, and we quantified the percentage of live C57BL/6J and CC030 ILC2s that were positive for activated caspases after 2 days of culture with IL-33, TSLP, and IL-2. CC030 ILC2s had a decreased percentage of active caspase positive cells compared to C57BL/6J ILC2s (Fig. 4G, H). To further compare apoptotic activity between the two strains, we performed surface staining with Annexin V and viability staining with propidium iodide (PI) to identify early

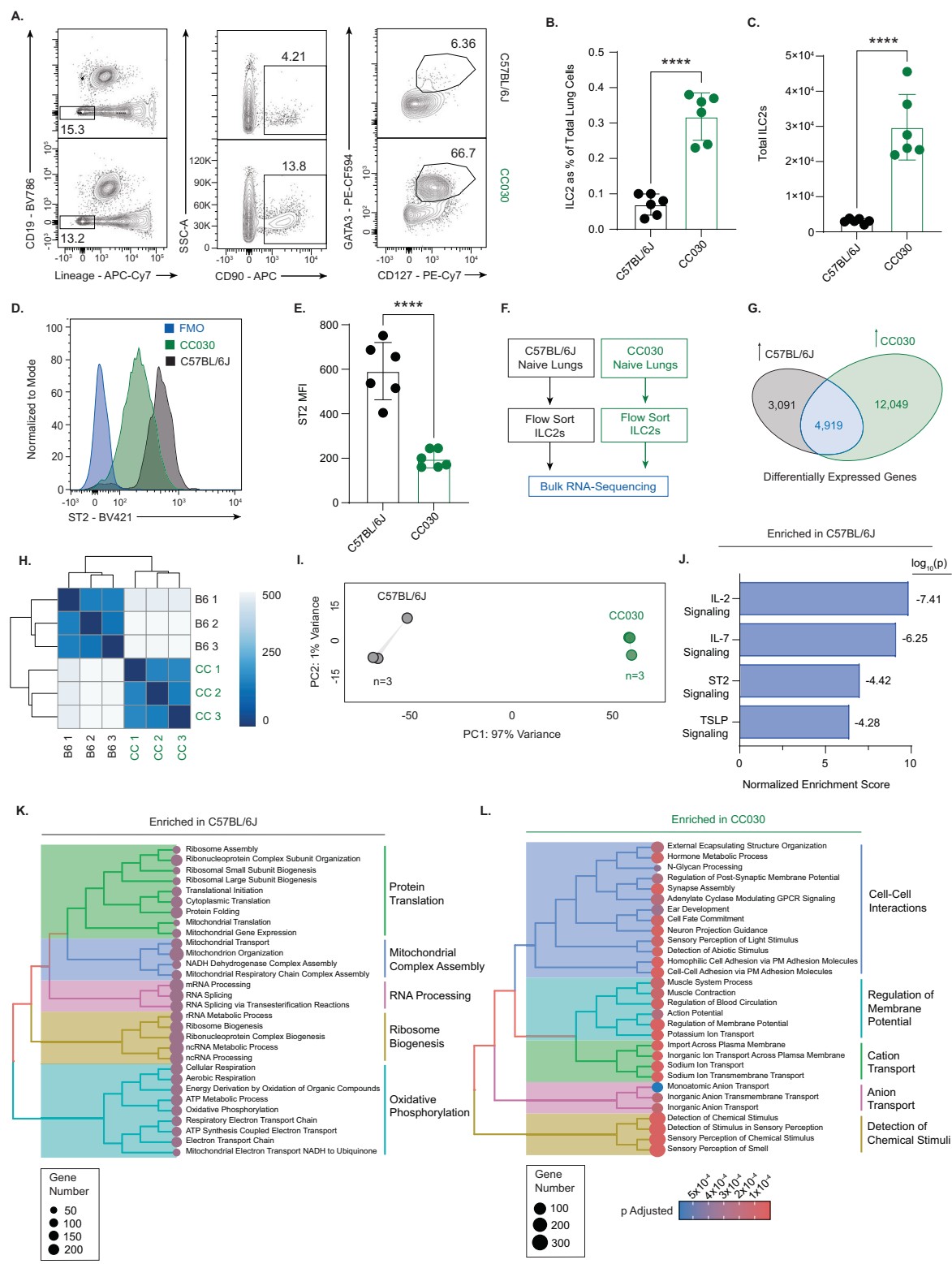

apoptotic (Annexin V + /PI-) and late apoptotic/dead (Annexin V + / PI + ) ILC2s (Fig. 4I). Interestingly, CC030 ILC2s had significantly more early apoptotic cells than C57BL/6J, but they had significantly fewer late apoptotic cells and showed an overall greater viability (Fig. 4J, K). This result may suggest that CC030 mice initiate apoptosis of ILC2s, but they become arrested in the early stages of this process and do not progress through programmed cell death as completely as C57BL/6J

mice. Corroborating these results, we identified multiple pro-apoptotic pathways enriched in C57BL/6J ILC2s compared to CC030 from our RNA-sequencing analysis between naïve CC030 and C57BL/6J ILC2s. (Fig. 4L). These results offer an explanation as to how CC030 ILC2s are less proliferative and less responsive to inflammatory cyto-kines compared to C57BL/6J, yet CC030 have a significantly larger population of ILC2s at baseline. A decreased ability to complete the

**Fig. 3 | CC030/GeniUnc mice have more numerous and transcriptionally distinct ILC2s at baseline compared to C57BL/6J. A** Gating strategy to define ILC2s in strain comparison studies (pre-gated on viable, singlet leukocytes). **B** ILC2s as percentage of live lung cells and as (**C**) total ILC2s in lungs of C57BL/6J and CC030 mice without challenge (n = 6 per strain, combined from 2 independent experiments) (Bar height = mean, error bars = SD). **D** Representative flow plots of ST2 expression on ILC2s from C57BL/6J and CC030 lungs at baseline. **E** Quantification of ST2 expression on ILC2s in C57BL/6J and CC030 lungs at baseline. MFI = mean fluorescence intensity, FMO = fluorescence minus one control (n = 6 per strain, combined from 2 independent experiments) (Bar height = mean, error bars = SD). **F** Diagram for comparison of C57BL/6J and CC030 ILC2s by bulk RNA-sequencing (n = 3 per strain). **G** Quantification of differentially expressed genes in ILC2 between C57BL/6J and CC030. **H** Sample distance matrix. 0 = perfect identity, 500 = maximum distance. B6 = C57BL/6J, CC = CC030. **I** Principal component analysis plot of ILC2s from C57BL/6J and CC030 naïve lungs. Group-wise convex hulls are included

in shading. **J** Bar plot representing annotated WikiPathways gene sets enriched in C57BL/6J compared to CC030. Gene sets were selected a priori as relevant to ILC2 signaling. **K.** Tree plot showing hierarchical clustering of the 30 Gene Ontology: Biological Processes (GO: BP) gene sets most enriched in C57BL/6J compared to CC030, ranked by normalized enrichment score. All adjusted p values for gene sets enriched in C57BL/6J are below $5 \times 10^{-9}$. **L** Tree plot showing hierarchical clustering of the 30 Gene Ontology: Biological Processes (GO: BP) gene sets most enriched in CC030 compared to C57BL/6J, ranked by normalized enrichment score. The color of the circle for each gene set denotes the p value according to the scale underneath the figure. Significance for (**B**), (**C**), and (**E**) was assessed by two-tailed unpaired Student's t-test. ****, p < 0.0001. Significance of gene set enrichment (**J–L**) was determined using a permutation-based one-sided test implemented in clusterProfiler (GSEA), with multiple testing correction by the Benjamini–Hochberg method (FDR).

process of apoptosis may compromise the overall inflammatory fitness of the ILC2 population, leading to an increase in cell number with diminished Type 2 effector function.

We then assessed if this same phenotypic distinction exists between the strains in vivo. We challenged CC030 and C57BL/6J mice in our 4-day *Alt* Ex challenge model (Fig. 1A), however, we increased the *Alt* Ex dosage to 8 µg from 5 µg to elicit a more robust Type 2 response, as our initial challenge did not produce detectable IL-5 and IL-13 in the lung homogenates of C57BL/6J mice. CC030 mice had significantly increased ILC2 prevalence compared to C57BL/6J after *Alt* Ex challenge (Fig. 4M), reconfirming our initial findings. We hypothesized that the higher number of ILC2s in CC030 was specific to ILC2s and not to innate lymphoid cells in general, so we compared ILC2s (Lin-, CD90 + , CD127 + , GATA3 + ) as a percentage of Lin-, CD90+ cells, which we term ILCs (Fig. 4N). Approximately 50% of the ILCs in the lung of CC030 mice were ILC2s, while ILC2s made up less than 10% of ILCs in C57BL/6J, suggesting that the higher percentage of ILC2s in CC030 mice compared to C57BL/6J mice is not due to a broad expansion of the innate lymphoid compartment. Despite having more than 15 times as many ILC2s, CC030 mice produced 5 times less IL-5 and 30 times less IL-13 compared to C57BL/6J mice by ELISA in the lung homogenate after *Alt* Ex challenge (Fig. 4O, P). CC030 mice had significantly decreased total eosinophils in the BAL after *Alt* Ex challenge compared to C57BL/6J mice, demonstrating a functional consequence of reduced IL-5 production (Fig. 4Q).

Intrigued by the diminished ILC2-driven Type 2 response in CC030 mice, we aimed to more broadly characterize the inflammatory profile of CC030 lungs at baseline and after challenge with different allergens. We quantified neutrophils, lymphocytes, eosinophils, and macrophages in the BAL and further quantified lymphocyte subtype by flow cytometry in the lung according to the gating strategy in Supplementary Fig. 5A without challenge, with papain challenge, and with *Alternaria* challenge. At baseline, CC030 mice have significantly more total ILC2s, CD4 + T cells, Tregs, BAL macrophages, and BAL neutrophils compared to C57BL/6J, but they have decreased CD8 + T cells and show no difference in total B cells, BAL eosinophils, or BAL lymphocytes (Supplementary Fig. 5B).

We then challenged C57BL/6J and CC030 mice with *Alt* Ex (4-days, 8 µg) and papain (3-days, 10 µg) to profile each strains immune response to different protease-containing allergens (Supplementary Fig. 5C). CC030 mice had significantly more ILC2s than C57BL/6J mice after papain challenge, mirroring our *Alt* Ex results. While *Alt* Ex challenge resulted in nearly significantly lower ILC2-expansion ratio and total IL-5, and significantly lower total IL-13 for CC030 compared to C57BL/6J, papain challenge did not reveal any differences in these outcomes. The paucity of IL-5 and IL-13 with papain challenge suggests an inability of papain to elicit a robust Type 2 innate response at this dose, which was chosen as the highest dose that did not cause hemorrhage in our model. CC030 mice also had a greater number of B

cells, CD8 + T cells, and CD4 + T cells with *Alt* Ex challenge but only a greater number of CD4 + T cells with papain. There was no significant difference in total Tregs with either allergen. In the BAL fluid, CC030 mice had significantly decreased eosinophilia but increased neutrophilia compared to C57BL/6J with *Alt* Ex challenge, with the neutrophilia alone being conserved in papain challenge. These results suggest that CC030 mice have a diminished innate Type 2 response to *Alternaria* that compensates with neutrophilia over eosinophilia, and some elements of this response are conserved in exposure to other protease containing antigens like papain.

## *Ffar3* is a gene candidate responsible for the QTL effect, thereby regulating ILC2 number and effector function

We leveraged the differences between CC030 and C57BL/6J mice to gain a better understanding of the phenotype that is regulated by the Ch.7 ILC2 Locus. Our next objective was to identify a specific targetable gene or subset of genes within the Ch.7 ILC2 Locus that could be responsible for the QTL effect. We developed a strategy to refine our list of 72 QTL genes from two hypotheses: 1.) a gene in the QTL that regulates ILC2 prevalence is differentially expressed between a strain with high ILC2 prevalence (CC030) and a strain with low ILC2 prevalence (C57BL/6J), and 2.) the gene in question is expressed in ILC2 and has an ILC2 intrinsic effect. Of the 72 protein coding genes, 50 were differentially expressed between ILC2s from CC030 and C57BL/6J mice. We were most interested in implicating genes that are relevant to ILC2 biology in humans, so we gated our gene list to only include those genes expressed in human ILC2s[36]. This yielded 12 human orthologs. The ultimate goal of this study was to uncover regulators of ILC2-mediated allergic inflammation that could be targeted for therapeutic benefit, and we therefore narrowed our selection to only those genes that are druggable targets[37]. We evaluated the 4 remaining genes to identify those with previously documented function on immune cells, which left only 2 genes: *Cd22 and Ffar3*. A diagram of this selection process is provided in Fig. 4R, and a detailed list of genes after each selection step is provided in Supplementary Fig. 6A. We then investigated the effects of both genes in regulating ILC2 prevalence and effector function. We used knockout mice and blocking antibodies to interrogate the role of CD22 on ILC2s, and while CD22 is expressed on ILC2s, we concluded from our experiments that it does not have an appreciable impact on ILC2 numbers or effector function. These experiments are provided in Supplementary Fig. 7.

With *Cd22* eliminated from our list, we turned our attention to free fatty acid receptor 3 (*Ffar3*). FFAR3 is a 7-transmembrane domain containing, lipid-binding Class A G-protein coupled receptor[38]. The receptor is expressed on pancreatic beta cells, epithelial cells of the gut, neuroendocrine tissues, and a host of other tissues throughout the body[39–41]. Microbiome-derived short-chain fatty acids (SCFAs) (acetate (C2), propionate (C3), butyrate (C4)) act as ligands for the receptor, with propionate and butyrate having increased binding

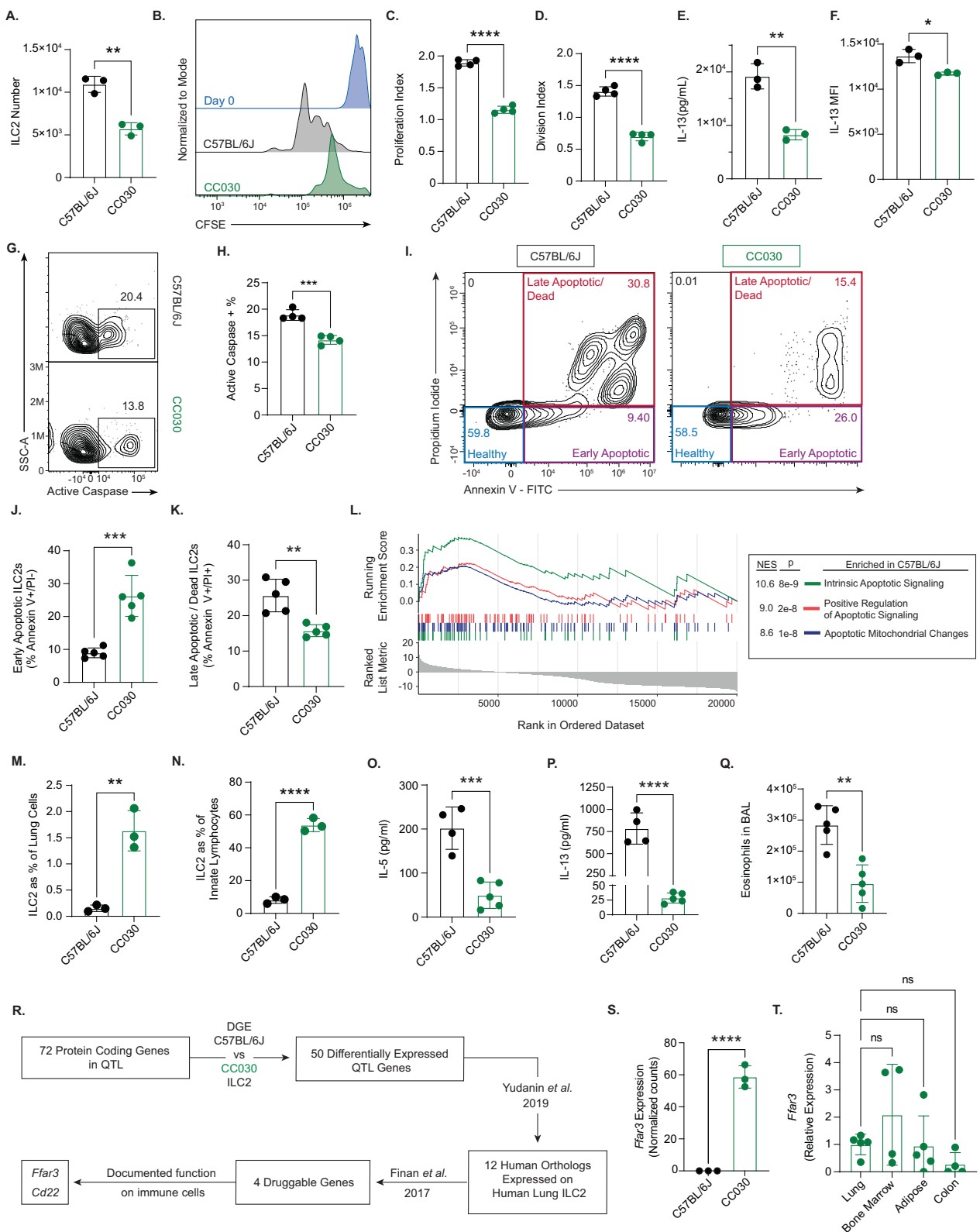

affinity compared to acetate. These same ligands activate the closely related free fatty acid receptor 2 (FFAR2), but this receptor shows increased affinity for acetate over propionate and butyrate[42,43]. While the FFAR3 receptor has been extensively studied in the context of diabetes and metabolism, more recent attention has been paid to the contribution of *Ffar3* to the regulation of inflammation[44–47]. SCFAs have a significant role in immune modification, through effects on epithelial barrier function[48,49], macrophages[50], B cells and plasma cells[51,52], helper T cell effector function[53], and regulatory T cell differentiation[54,55]. Most of the reported associations between SCFA signaling and immune regulation have implicated FFAR2 as the responsible receptor[56]. However, important studies have specifically linked the SCFA-FFAR3 axis to the regulation of Type 2 inflammation in asthma and allergic disease. Gut microbiome-derived SCFAs

**Fig. 4 | Strain-specific ILC2 phenotypes between CC030 and C57BL/6J implicate *Ffar3* as a candidate regulator. A** ILC2s from lungs of C57BL/6J and CC030 after 4 days of culture with IL-33, TSLP, and IL-2 (n = 3 per strain) ** p = 0.0016. **B** CFSE dilution between ILC2s after 48 h of culture with IL-33, TSLP, and IL-2. **C** Proliferation index and **D** division index after 48 h culture as in (B) (n = 4 per strain) **** p < 0.0001. **E** IL-13 in supernatant of ILC2s cultured per (A) (n = 3 per strain) ** p = 0.0018. **F** IL-13 expression by intracellular cytokine staining. MFI = mean fluorescence intensity (n = 3 per strain) * p = 0.011. **G** Representative flow plots of active caspase+ ILC2s, pre-gated on live cells (numbers above gates = %). **H** Quantification of active caspase+ ILC2s as in (G) (n = 4 per strain) *** p = 0.0004. **I** Representative flow plot of early/late apoptosis by Annexin V and propidium iodide (PI) staining on ILC2s 6 days post culture. **J** Early and late apoptotic **K** ILC2s (n = 5 per strain) *** p = 0.0004, ** p = 0.0019. **L** Apoptotic pathways enriched in naïve C57BL/6J vs CC030 ILC2s. **M** ILC2s as % of live lung cells and **N** as % of innate lymphocytes (Lineage⁻, CD90⁺) (n = 3 per strain) ** p = 0.0027, **** p < 0.0001. **O** IL-5 and **P** IL-13 in lung homogenate after 4 days of Alt Ex challenge (n = 4 C57BL/6J, 5 CC030) *** p = 0.0006, **** p < 0.0001. **Q** Total eosinophils in bronchoalveolar lavage (BAL) fluid after 4 days of Alt Ex challenge (n = 5 per strain, also in Supplementary Fig. 4 C) ** p = 0.0027. **R** Diagram implicating candidate genes in Ch.7 ILC2 locus. **S** Ffar3 expression by normalized counts from bulk RNA-seq (DESeq2 DGE analysis, Wald test (two-tailed) with Benjamini−Hochberg correction; **** p = 1.76 × 10⁻¹²). **T** qPCR for Ffar3 from RNA of flow-sorted naïve CC030 ILC2s from various tissues (n = 5 mice; one sample each from bone marrow, adipose, and colon had insufficient RNA). **A, C−F, H, J, K, M−Q, S, T**: bar height = mean, error bars = SD. **A, C−F, M−P, S−T**: each representative of two independent experiments, **H, J, K, Q, S, T**: each representative of one independent experiment. Significance for (**A**), (**C−F**), (**H**), (**J, K**), (**L−Q**), assessed by two-tailed unpaired Student's *t*-test; for (**T**) by one-way ANOVA with Tukey multiple comparisons (two-tailed).

downregulated T helper type 2 ($T_H2$) cell effector function in an FFAR3, not FFAR2, dependent manner[57]. Two groups reported that FFAR3 was highly expressed in $T_H2$ cells in eosinophilic esophagitis patients compared to healthy controls and described the receptor's critical role in regulating IL-5 and IL-13 responses[58,59]. Others have reported the effect of short chain fatty acids on influencing ILC2 biology, but they did not implicate FFAR3 as a regulating receptor of any SCFA-dependent phenomena in the lung[60–62]. One study in particular established the expression of *Ffar3* in pulmonary ILC2s in mice, but their data suggested that SCFAs modulated ILC2s primarily in a receptor-independent manner, likely through histone de-acetylase inhibition[63]. With known interaction between SCFAs and the immune system, and evidence supporting the role of FFAR3 in regulating adaptive Type 2 inflammation, we hypothesized that FFAR3 signaling influenced ILC2 function and aimed to clarify the role of this receptor on ILC2s.

We first sought to confirm the expression of *Ffar3* in naïve CC030 ILC2s, which is demonstrated by the normalized read counts from our RNA-sequencing data (Fig. 4S). We isolated ILC2s from the naïve lungs of five CC030 female mice by flow cytometry sorting and extracted RNA. We confirmed the presence of *Ffar3* transcript by q-PCR in line with previous reports (Fig. 4T), and we report its expression on ILC2s from a variety of tissue sources.

## FFAR3 signaling expands ILC2 numbers and reprograms ILC2s to an anti-inflammatory effector state in vitro

C57BL/6J mice did not express FFAR3 on their ILC2s in our RNA-sequencing experiment (Fig. 4S), and we were therefore concerned that available *Ffar3*KO strains on this background would have been suboptimal in investigating the effect of this receptor on ILC2s. As such, we were restricted to working with wild-type ILC2s and modulating FFAR3 directly in CC030 mice. To understand the effects of FFAR3 agonism on ILC2 activation and effector function, we cultured flow-sorted ILC2s from CC030 mice with the activating cytokines IL-33, TSLP, and IL-2. We added to the culture either AR420626, a highly specific FFAR3 agonist[64], or DMSO vehicle (Fig. 5A).

After 6 days of culture, AR42062 increased the number of ILC2s nearly 3-fold compared to vehicle controls (Fig. 5B). Despite this large increase in ILC2 numbers, AR420626 significantly decreased the ability of these ILC2 to produce IL-5 and IL-13, measured by the concentration of cytokines in the supernatant (Fig. 5C) and expression on a per-cell basis via intracellular cytokine staining (Fig. 5D, E). In addition to decreasing Type 2 cytokine production, FFAR3 agonism significantly decreased ST2 expression on ILC2s (Fig. 5F). AR420626-treated ILC2s had a greater than 3-fold decrease in the percentage of cells that were ST2+ (Fig. 5G), and the MFI of ST2 on positive cells was decreased by more than 60% (Fig. 5H). This decrease in ST2 MFI with FFAR3 agonism mirrors the observed decrease in ST2 expression in CC030 ILC2 compared to C57BL/6J. In contrast to decreased production of

inflammatory cytokines, AR42062-stimulated ILC2s experienced a ~12-fold increase in the secretion of the anti-inflammatory cytokine IL-10 compared to vehicle controls (Fig. 5I).

We hypothesized that the FFAR3 signaling-dependent increase in cell number would not be a function of proliferation, but of resistance to apoptosis, reflecting our observations in CC030 ILC2s (FFAR3 expression) compared to C57BL/6J (no FFAR3 expression). AR420626 treated ILC2s had a similar fraction of Ki-67+ cells compared to vehicle controls (Fig. 5J). While the difference was statistically significant, the small magnitude of change (~2%) suggests that it is unlikely to be biologically meaningful (Fig. 5K). Further supporting this observation was the lack of a significant difference in proliferation or division after 48 h of culture by CFSE dilution (Fig. 5L, M).

FFAR3 agonism increased the number of ILC2s but did not meaningfully increase proliferation. From this result, we hypothesized that the difference in ILC2 number was instead due to an FFAR3-signaling dependent decrease in apoptosis progression. The persistence of less fit cells that would be otherwise eliminated through programmed cell death could explain both the elevated ILC2 population size without proliferation difference and the decreased Type 2 effector function observed in AR420626-treated ILC2s. Similar to what was observed in CC030 vs. C57BL/6J mice, AR420626-treated ILC2s had a modestly higher percentage of cells in early apoptosis but a roughly 3-fold decrease in the percentage of cells advancing to late apoptosis and cell death (Fig. 5.N, P). This considerable increase in cell viability supports the idea that FFAR3 signaling is not expanding cell number through stimulating cell proliferation, but through the maintenance of a large cell population by the inhibition of programmed cell death.

These in vitro data show that the effect of FFAR3 agonism mirrors what was observed in the differences between CC030 and C57BL/6J, highlighting an increase in ILC2 cell number with a concomitant decrease in Type 2 effector function, and even revealing an increase in the ability of FFAR3 agonist-treated ILC2s to produce the anti-inflammatory cytokine IL-10. This effect is particularly interesting, as ILC2 cell number after activation and Type 2 cytokine secretion traditionally correlate[65–67]. ILC2s increasing cell number, decreasing Type 2 cytokine secretion, and increasing secretion of IL-10 is a response that, to the best of our knowledge, has not yet been categorized and possibly represents a unique ILC2 effector state.

We observed a lack of *Ffar3* expression in C57BL/6J ILC2s in our initial sequencing experiment, and we therefore expected C57BL/6J ILC2s to have diminished or no response to FFAR3 agonism. We cultured C57BL/6J ILC2s according to our model in Fig. 5A and did not observe a decrease in IL-5 or IL-13, a difference in ILC2 number, or an effect on apoptosis. (Supplementary Fig. 8A-G). Similarly, AR420626 treatment had no effect on C57BL/6J ILC2 proliferation, measured by CFSE dilution (Supplementary Fig. 8H-J). Thio et al. demonstrated expression of *Ffar3* on ILC2s from IL-33 pre-treated *Rag2⁻/⁻* mice, but they surprisingly did not observe any effect of AR420626 treatment

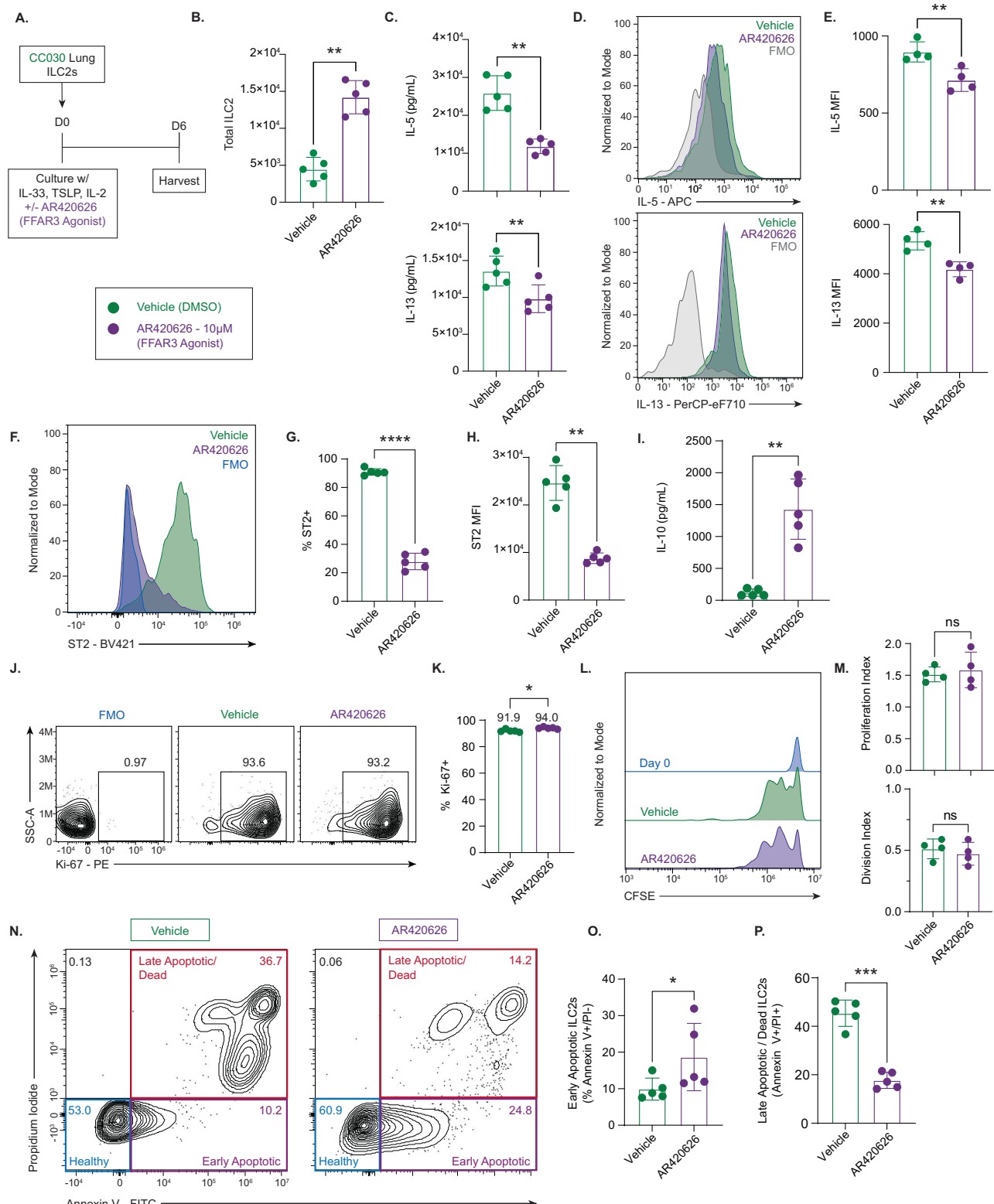

(10 μM) on cytokine production from ILC2s stimulated by IL-33 in vitro[63].

Our results suggested that differences in baseline FFAR3 signaling on ILC2s between C57BL/6J and CC030 mice were attributable to differences in receptor expression, but we aimed to investigate possible in vivo differences in SCFAs, the endogenous FFAR3 ligands, as well. We quantified acetate, propionate, and butyrate in the serum of C57BL/6J and CC030 mice at baseline via mass spectrometry, and CC030 mice had significantly lower serum concentrations of acetate compared to C57BL/6J, while there were no differences in propionate or butyrate (Supplementary Fig. 9). The lower concentration of FFAR3 ligands in the serum of CC030 mice suggested that the increased FFAR3 signaling on CC030 ILC2s is dependent on increased receptor expression, not SCFA concentration.

**Fig. 5 | FFAR3 signaling expands ILC2 numbers and reprograms ILC2s to an anti-inflammatory effector state in vitro. A** Schematic of 6-day ILC2 culture with IL-33 (10 ng/mL), TSLP (10 ng/mL), and IL-2 (100 U/mL). AR420626 concentration = 10 μM. All data shown are from Day 6 unless indicated. **B** Total ILC2 numbers after 6 days of culture (n = 5, paired biological replicates; representative of 3 independent experiments) ** p = 0.0029. **C** IL-5 (** p = 0.0014) and IL-13 (** p = 0.0084) in culture supernatant (n = 5, paired biological replicates; representative of 3 independent experiments). **D** Representative flow cytometry plots of IL-5 and IL-13 by intracellular cytokine staining. **E** Quantification of IL-5 (** p = 0.0015) and IL-13 (** p = 0.0034) expression by intracellular staining as MFI (mean fluorescence intensity) (n = 4, paired biological replicates; representative of 1 experiment for IL-5 and 2 for IL-13). **F** Representative flow cytometry of ST2 expression on cultured ILC2s (FMO = fluorescence minus one control). **G** Percentage of ST2+ ILC2s (n = 5, paired biological replicates; representative of 1 experiment) **** p < 0.0001. **H** Quantification of ST2 expression on ST2+ ILC2s by MFI (n = 5, paired biological replicates; representative of 1 experiment) ** p = 0.0012. **I** IL-10 in supernatant (n = 5, paired biological replicates; 3 samples in Vehicle group below detection limit; representative of 3 independent experiments) ** p = 0.0023. **J** Representative flow cytometry plot of Ki-67+ ILC2s. **K** Percentage of Ki-67+ ILC2s (n = 5, paired biological replicates; representative of 1 experiment) * = 0.044. (**L**) Representative CFSE dilution plot to assess proliferation after 48 h of culture. **M** Proliferation and division index quantification between groups (n = 4, paired biological replicates; representative of 2 independent experiments). **N** Representative flow cytometry of early/late apoptosis by Annexin V and propidium iodide (PI) staining after 6 days. Annexin V−/PI− = healthy, Annexin V+/PI− = early apoptotic, Annexin V+/PI+ = late apoptotic/dead. Numbers in quadrants = percentages. **O** Percentage of early apoptotic (Annexin V+/PI−) (* p = 0.038) and **P** late apoptotic/dead (Annexin V+/PI+) (*** p = 0.0002) cells (n = 5 per strain; representative of 1 experiment). **B**, **C**, **E**, **G**–**I**, **K**, **M**, **O**, **P** bar height = mean, error bars = SD, significance assessed by two-tailed paired Student's t-test. ns not significant.

## FFAR3 agonism elicits a unique IL-10-producing effector state in ILC2s

We were intrigued by the anti-inflammatory cytokine secretion profile that FFAR3 agonism imparted on ILC2s, and we sought to further characterize this effector state. We investigated whether this IL-10 producing phenotype was dependent the on the specific combination of stimulatory cytokines used to activate ILC2s in our culture system. A previous report from Seehus et al. demonstrated that IL-2 was necessary for IL-10 induction in ILC2s[68]. We cultured CC030 ILC2s with AR420626 or vehicle in the presence or absence of IL-2, and we observed that IL-2 was necessary for AR420626 to increase cell number (Fig. 6A), decrease IL-5 production (Fig. 6B), and induce IL-10 production (Fig. 6C). Given that FFAR3's anti-inflammatory effect is IL-2 dependent, we hypothesized that FFAR3 signaling may increase IL-2 signaling through upregulation of the high-affinity IL-2 receptor alpha (CD25). We confirmed our hypothesis by detecting greater CD25 expression by flow cytometry on ILC2s treated with AR420626 compared to vehicle controls (Fig. 6D, E). These data suggest that, like previously described IL-10-producing ILC2s, FFAR3 acts in an IL-2 dependent manner and may induce this state through an upregulation of CD25.

Given the concomitant production of IL-10 and decrease in Type 2 cytokines, it seemed possible that IL-10 was acting in an autocrine and or paracrine manner to inhibit IL-5 and IL-13 production in ILC2s. We used an IL-10 blocking antibody to determine if the reciprocal changes in anti-inflammatory and inflammatory cytokines were dependent or independent processes. IL-10 blockade had no effect on ILC2 number, IL-5 production, or IL-13 production (Fig. 6F–I). Our results corroborate a previous report that showed IL-10 signaling having no effect on IL-5 production by ILC2s[69].

To understand whether this anti-inflammatory state elicited by FFAR3 agonism was unique or similar to previously described ILC2 effector states, we compared AR420626 head-to-head with known inducers of IL-10 expression like retinoic acid (RA)[68,70–72]. AR420626 induced similar IL-10 expression to RA, but interestingly, RA significantly increased IL-5 secretion compared to vehicle, while AR420626 significantly decreased IL-5 secretion compared to vehicle (Fig. 6J, K). In fact, RA-stimulated ILC2s produced roughly 3-fold more IL-5 than those treated with the FFAR3 agonist. While a benchmark IL-10 inducer like RA increases IL-10 expression with concurrent increase in IL-5, AR420626 induces IL-10 while decreasing IL-5. This opposite regulation of Type 2 cytokines, which is independent of autocrine and or paracrine IL-10 signaling, suggests that the ILC2 effector state induced by FFAR3 signaling is unique in its anti-inflammatory nature; it increases anti-inflammatory cytokine production while decreasing pro-inflammatory cytokine production.

We compared AR420626-treated ILC2s with other IL-10-producing ILC2s in an unbiased manner to further assess the apparent distinctiveness of this effector state. We performed bulk-RNA-sequencing on AR420626 and vehicle treated ILC2s after 6 days of culture. We compared the transcriptional signatures of our FFAR3 agonist-treated ILC2s against those of other IL-10 + ILC2s from previously published sequencing experiments[68,73]. There are apparent transcriptional differences between AR4206262-treated ILC2s, IL-10 + ILC2s from Seehus et al., and IL-10 + ILC2s from Howard et al. To assess global similarity between the 3 sets of ILC2s, we performed PCA analysis on normalized read counts from each experiment using a batch correction method that retained variation associated with IL-10+ status (Fig. 6L). It is apparent from the PCA plot that the other two IL-10 + ILC2 effector states are much more similar to each other than they are to our FFAR3-dependent state, suggesting a uniqueness to our examined phenotype.

## FFAR3 reprograms ILC2s to an anti-inflammatory effector state by increasing EGFR expression

From our bulk RNA-sequencing data, we endeavored to gain further insight into the mechanism by which FFAR3 signaling reprogrammed ILC2s (Fig. 7A). Amongst all differentially expressed genes (Supplementary Data 5), *Egfr* had the greatest positive fold-change, and the second lowest p value when comparing AR420626-treated to vehicle-treated ILC2s (Fig. 7B). *Egfr* codes for the epidermal growth factor receptor (EGFR), a known regulator of cell growth that has been implicated in a host of neoplastic disorders[74]. A prior report established the expression of EGFR on regulatory T cells (Tregs), and it demonstrated that amphiregulin (an EGFR ligand) signaling through Treg-intrinsic EGFR was critical for enhancement of Treg suppressor function, including IL-10 production[75]. ILC2 have been reported to secrete amphiregulin[76,77], and thus, an autocrine/paracrine amphiregulin-EGFR signaling axis on our cultured ILC2s could explain the reprogramming of FFAR3-agonist treated ILC2s to an anti-inflammatory, IL-10 producing phenotype.

AR42062-treated cells had significantly greater *Egfr* expression compared to vehicle controls by RNA-sequencing (Fig. 7C), and we confirmed this finding by flow cytometry (Fig. 7D, E). We established causality between EGFR signaling and FFAR3 signaling through a repeat of our culture experiment with the EGFR inhibitor gefitinib[78]. We observed that AR42062 expanded ILC2 numbers compared to vehicle controls, but AR42062 with the addition of gefitinib reduced ILC2 numbers back to the level of vehicle controls (Fig. 7F). Similarly, while AR42062 treatment of ILC2s increased IL-10 secretion compared to vehicle controls, AR42062 and gefitinib reduced IL-10 secretion compared to AR42062 alone (Fig. 7G). EGFR blockade negated the AR42062-dependent decrease in IL-5 production, demonstrated by the increase in IL-5 expression by intracellular cytokine staining in gefitinib and AR42062-treated ILC2s compared to only AR42062-treated ILC2s (Fig. 7H, I). We confirmed that ILC2s secrete amphiregulin in our

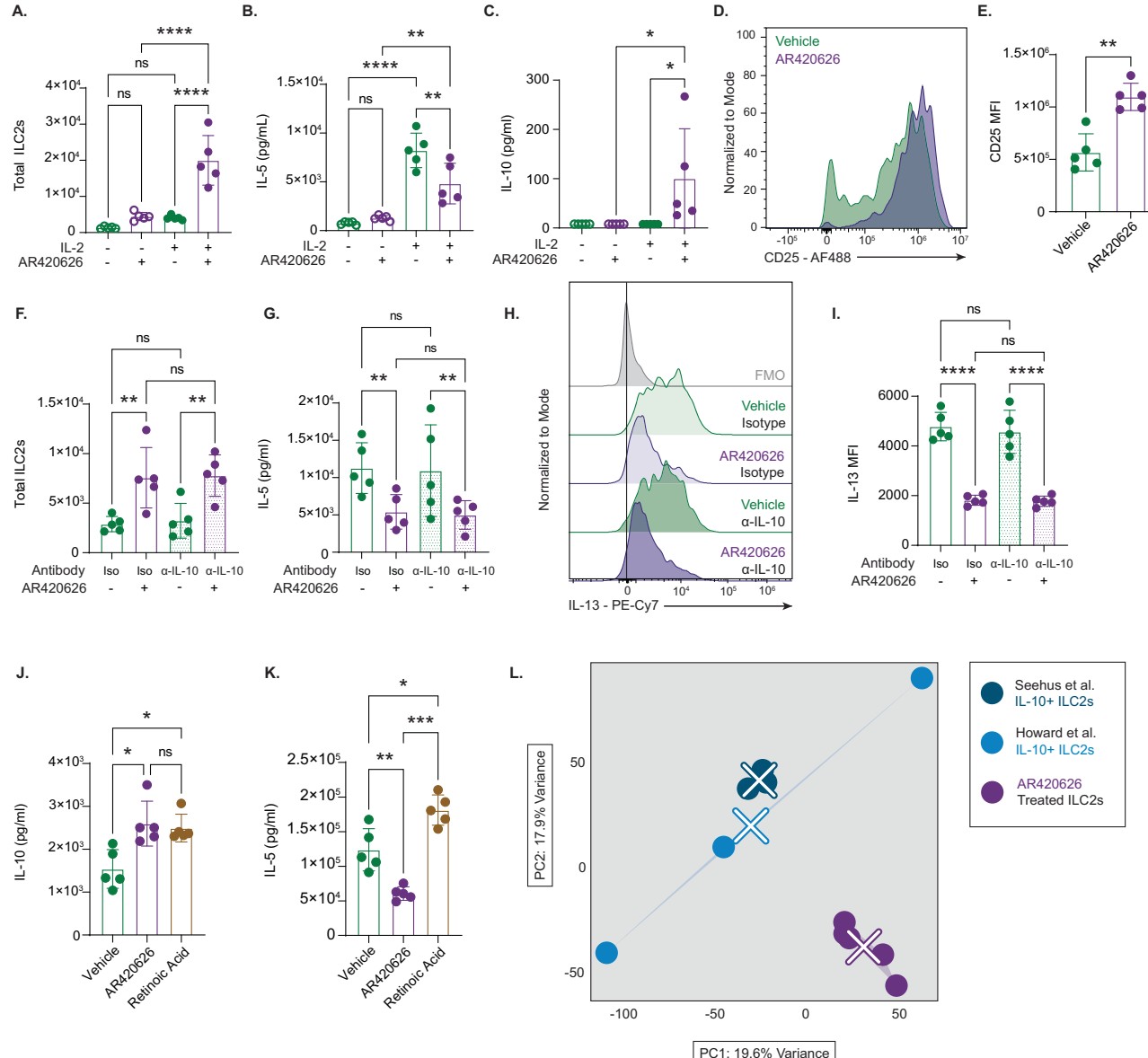

**Fig. 6 | FFAR3 agonism elicits a unique IL-10-producing effector state in ILC2s.**
**A** Total ILC2s after 6-day culture with IL-33 and TSLP ± IL-2 and ± AR420626. All cultures use CC030 ILC2s harvested at Day 6. **** p < 0.0001. **B** IL-5 concentration in the supernatant of ILC2s activated with IL-33 and TSLP ± IL-2 and ± AR420626, groups = A-D left to right. (A–C **** p < 0.0001), (B–D ** p = 0.0034), (C-D ** p = 0.0045). **C** IL-10 concentration in the supernatant of ILC2s activated as in (B); first three groups (from left) were below the detection limit * p = 0.026.
**D** Representative flow cytometry of CD25 expression on cultured ILC2s.
**E** Quantification of CD25 expression by median fluorescence intensity (MFI) ** p = 0.0067. **F** Total ILC2s cultured with IL-33, TSLP, and IL-2 ± AR420626 ± α–IL-10 antibody (A-B ** p = 0.0037) (C-D ** p = 0.0045). **G** IL-5 concentration in the supernatant of ILC2s cultured as in (F) (A-B ** p = 0.0080) (C-D ** p = 0.0074).
**H** Representative flow cytometry of IL-13 expression by intracellular cytokine staining in ILC2s with α–IL-10 antibody or isotype control ± AR420626 (FMO = fluorescence minus one control). **I** Quantification of IL-13 expression by MFI in ILC2s treated as in (H) **** p < 0.0001. **J** IL-10 concentration in supernatant of ILC2s activated with IL-33, TSLP, IL-2 and treated with DMSO(A), AR420626(B) (10 μM), or retinoic acid(C) (1 μM) (A-B * p = 0.013) (A–C * p = 0.020). **K** IL-5 concentration in supernatant of ILC2s cultured as in (J) (A B ** p = 0.0058), (A–C * p = 0.0242), (B-C *** p = 0.0003). **L** PCA of IL-10⁺ ILC2s from Howard et al., Seehus et al., and AR420626-treated ILC2s (see main text for references). Group-wise convex hulls indicate groups, with centroids marked by X. **A–C, E–G, I–K:** bar height = mean, error bars = SD, n = 5, paired biological replicates; each representative of one experiment. **A–C, F, G,** and **I** use two-way ANOVA with Sidak's multiple testing correction, **J, K** use one-way ANOVA with Tukey's multiple testing correction (two-tailed), and (**E**) uses a two-tailed paired Student's t-test. ns not significant.

culture system (Fig. 7J). Blockade of amphiregulin in AR420626-treated ILC2s limited IL-10 induction to a modest but statistically significant degree (Fig. 7K), suggesting that amphiregulin, and possibly other EGFR ligands, are signaling through EGFR in an autocrine and paracrine manner. Given that FFAR3 signaling upregulates EGFR on ILC2s, and that inhibition of EGFR signaling reversed many of the observed effects of FFAR3 signaling, we propose that FFAR3 is partially regulating ILC2s through an EGFR-dependent mechanism, seemingly

mirroring the effect of EGFR promoting increased IL-10 secretion in Tregs.

### FFAR3 signaling partially reprograms human ILC2s towards an anti-inflammatory effector state in vitro

Our gene selection process considered expression on human ILC2s to maintain the translational potential of any discovery, and so we aimed to validate the effect of FFAR3 signaling on human ILC2s directly.

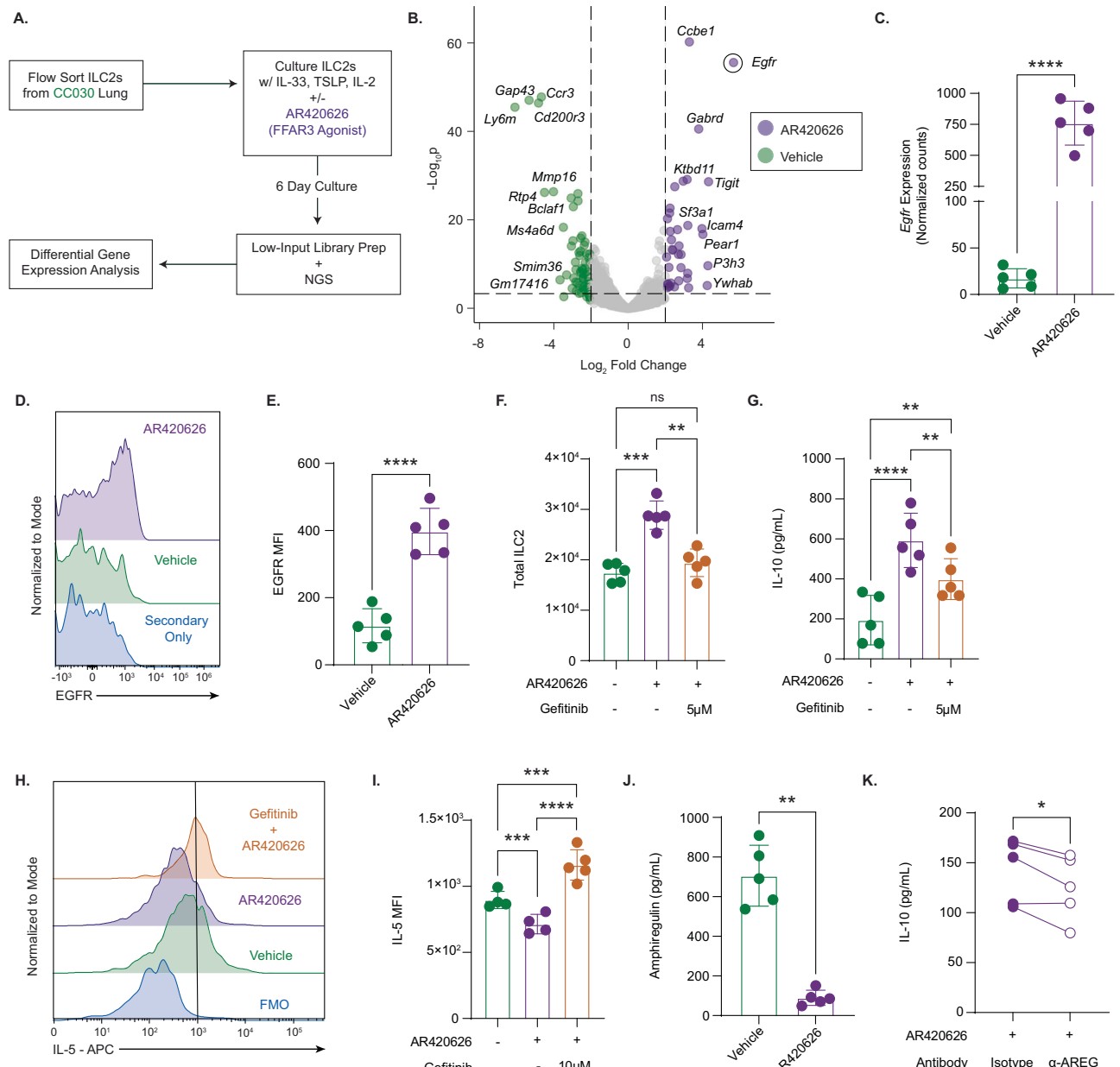

**Fig. 7 | FFAR3 reprograms ILC2s to an anti-inflammatory effector state by increasing EGFR expression. A** Schematic for vehicle- vs. AR420626 (10 µM)–treated ILC2s by bulk RNA-sequencing. **B** Volcano plot of differentially expressed genes from (**A**). Egfr encircled. Labels = 10 most positively and negatively differentially expressed genes by log2 fold change (p-value threshold = $5 \times 10^{-4}$; fold change threshold = 4; n = 5 paired biological replicates, representative of one experiment for (**A–C**). **C** Normalized EGFR counts from RNA-seq data in vehicle vs agonist-treated cells. Significance determined for (**B**, **C**) by DESeq2 DGE analysis (Wald test (two-tailed) with Benjamini–Hochberg correction); ***$p < 2 \times 10^{-52}$. (Bar height = mean, error bars = SD). **D** Representative flow plots of EGFR expression on ILC2s with and without FFAR3 agonist. Secondary-only control: no α-EGFR primary antibody. **E** EGFR expression by median fluorescence intensity (MFI) (n = 5 paired biological replicates; representative of one experiment) **** p < 0.0001. **F** Total ILC2s after 6-day culture with DMSO AR420626 (10 µM) and gefitinib (5 µM), groups = A–C left to right. (A-B *** p = 0.0005) (B-C ** p = 0.0018). **G** IL-10 levels in

the supernatant; two vehicle (AR420626-/gefitinib-) samples were below detection. (A-B **** p < 0.0001) (A-C ** p = 0.0070) (B-C ** p = 0.0097). (**F**, **G**): n = 5 paired biological replicates; representative of two independent experiments). **H** Representative intracellular cytokine staining of IL-5 (gefitinib = 10 µM). **I** Quantification of IL-5 MFI (n = 4 for Vehicle and AR420626 (1 sample each used as FMOs); n = 5 for AR420626 + gefitinib). IL-5 quantification for vehicle and AR420626 is shown in Fig. 5E (A-B *** p = 0.0007) (A-C *** p = 0.0002) (B-C **** p < 0.0001). **J** Amphiregulin in the supernatant of ILC2s with and without 10 µM AR420626 measured by ELISA (n = 5 paired biological replicates; representative of one experiment) ** p = 0.0012. **K** IL-10 in supernatant of ILC2s cultured with IL-33, TSLP, IL-2 + AR420626 + α-Amphiregulin antibody or isotype control (n = 5 paired biological replicates; representative of one experiment) * p = 0.033. **C**, **E–G**, **I**, **J** bar height = mean, error bars = SD). Significance for (**E**), (**J**), and (**K**) was assessed by two-tailed paired Student's t-test; for (**F**), (**G**), and (**I**) by two-way ANOVA with Tukey's multiple comparisons test. ns not significant.

Peripheral blood mononuclear cells (PBMCs) were isolated from healthy donors, and ILC2s were flow sorted as Lineage-, CD45 +, CD127 +, CRTH2+ cells (Fig. 8A). We cultured these human ILC2s with the activating cytokines IL-33, TSLP, and IL-2 with the FFAR3-specific agonist AR420626 at concentrations of 1, 5, and 10 µM or DMSO

vehicle control. AR420626 had no effect on cell number at 1 µM, but 5 µM and 10 µM significantly decreased ILC2s after 6 days of culture compared to vehicle controls (Fig. 8B). Neither of the 3 concentrations of AR420626 decreased ILC2 viability, suggesting that the drug's effects were not a function of toxicity (Fig. 8C). AR420626 treatment

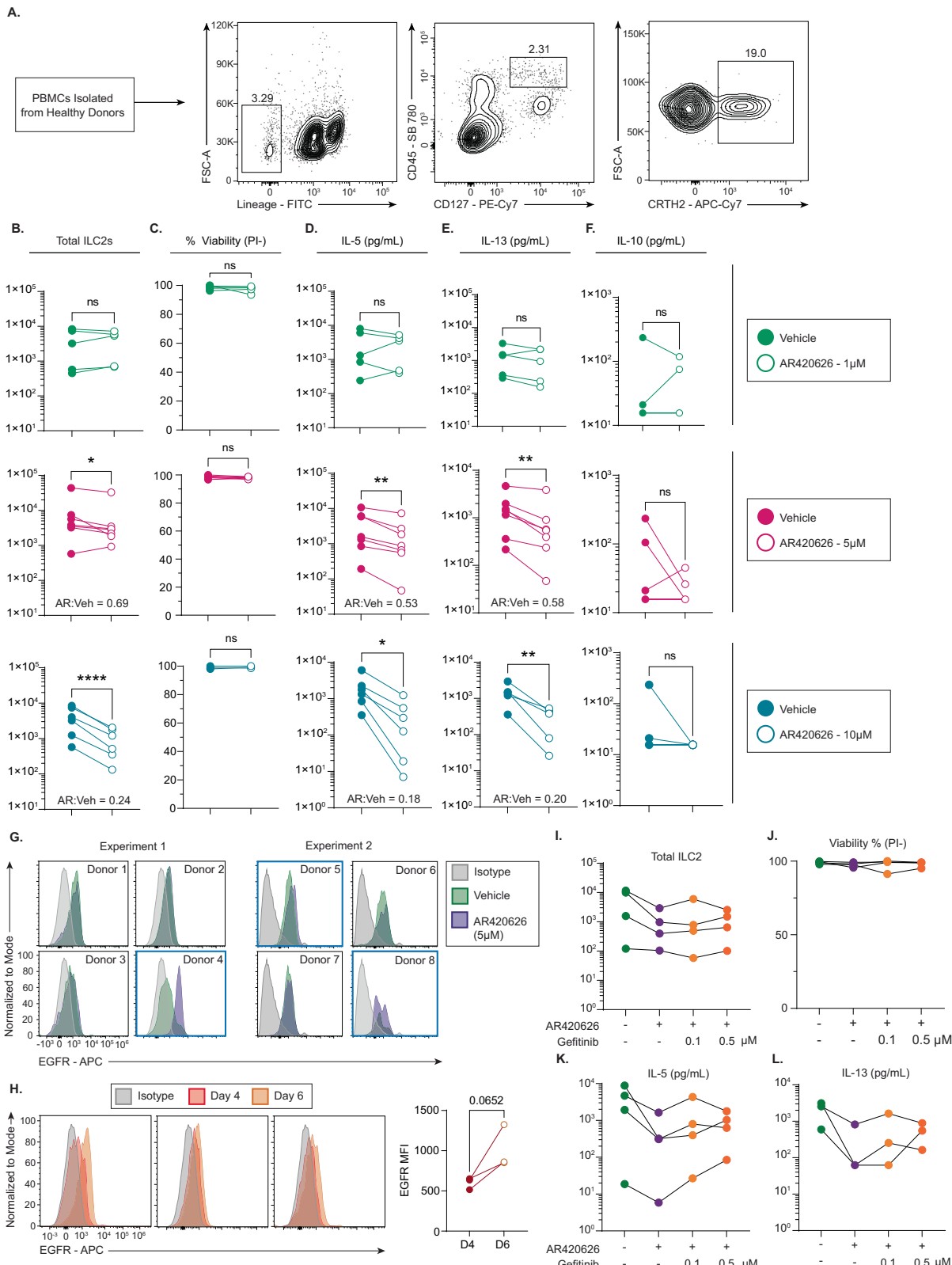

demonstrated a decrease in IL-5 and IL-13 production, with 10 μM decreasing both cytokines approximately 5-fold (Fig. 8D, E). Despite this marked decrease in Type 2 inflammatory cytokines, AR402626 did not induce IL-10 production at any concentration in these human ILC2s (Fig. 8F). At 5 μM, AR420626 had a more modest effect on decreasing cell number compared to decreasing IL-5 and IL-13 secretion,

suggesting a potentially greater effect on cytokine secretion than cell expansion.

We confirmed EGFR expression on human ILC2s, but we observed a significant degree of heterogeneity in the response of EGFR expression to FFAR3 signaling (Fig. 8G). Of 8 examined samples, 3 experienced increases in EGFR MFI of at least 10%, while all other remained

**Fig. 8 | FFAR3 signaling partially reprograms human ILC2s towards an anti-inflammatory effector state in vitro. A** Isolation of ILC2s from human PBMCs with representative gating for flow sorting (pre-gated on live, singlet lymphocytes). ILC2s from each donor evenly split between conditions (DMSO vehicle vs AR420626) and cultured with IL-33, TSLP, and IL-2 for 6 days. **B** Total ILC2s (n = 5 for 1 μM AR420626, n = 7 for 5 μM AR420626, n = 6 for 10 μM AR420626; same n for each endpoint in **B–F**). Values for AR:Veh = ratio of means between AR420626 and vehicle groups (5 μM: * p = 0.031, 10 μM: **** p < 0.0001). **C** Percent viability of ILC2s measured by propidium iodide (PI) negativity percentage. **D** Concentration of IL-5 (5 μM: ** p = 0.0018, 10 μM: * p = 0.031), **E** IL-13 (5 μM: ** p = 0.0033, 10 μM: ** p = 0.0080), and **F** IL-10 in the supernatant. Lowest values for IL-10 at each concentration = below the detection, multiple overlapping samples. **G** Representative flow plot of EGFR expression on human ILC2s with 5 μM AR420626 or vehicle.

**H** Flow plot of EGFR expression on human ILC2s with IL-33, TSLP, and IL-2 at Days 4 and 6. **I** Total cell number and **J** viability (PI–) of human ILC2s cultured with IL-33, TSLP, IL-2 ± AR420626 and gefitinib at different doses (n = 4; gefitinib treatment showed no significant effect by binomial test). **K** IL-5 and **L** IL-13 in supernatant of human ILC2s (n = 4 IL-5; n = 3 IL-13). All IL-13 values for one donor were below detection. Statistical testing for K and L assessed whether gefitinib reversed AR420626-mediated IL-5 and IL-13 suppression. IL-5 (4/4) and IL-13 (3/3) levels were higher with at least one gefitinib dose compared to AR alone. Binomial testing yielded p = 0.0625 (IL-5) and p = 0.125 (IL-13). Shapiro–Wilk tests assessed normality of paired differences in (**B–F, H**). Normality not rejected: ratio paired Student's t-test was used; rejected: Wilcoxon signed-rank test. Ratio two-tailed paired Student's t-test for all analyses in (**B–F, H**) except 8.B–5 μM, 8.C–1 μM, 8.D–10 μM, and 8.F–5 μM, which used Wilcoxon signed-rank test (two-tailed). ns not significant.

identical between AR420626 and vehicle treated ILC2s. Interestingly, there was a trend towards an increase in EGFR expression on cells at Day 6 of culture compared to Day 4 (Fig. 8H), suggesting that EGFR expression is induced through the process of ILC2 activation. Despite the variability in FFAR3-dependent changes in EGFR expression, the EGFR inhibitor gefitinib trended towards reversing the effects of AR420626 in limiting Type 2 cytokine production. While adding gefitinib at multiple concentrations to AR420626-treated ILC2s had no appreciable effect on cell number or viability, there was a near significant reversal of IL-5 production with gefitinib treatment (p = 0.063) (Fig. 8I–L), suggesting that EGFR blockade may undo the anti-inflammatory effects imparted on ILC2s by FFAR3 signaling.

Taken together, these results demonstrate that the anti-inflammatory effect of FFAR3 signaling is partially conserved between mice and humans. AR420626 treatment significantly decreased Type 2 effector function for ILC2s of both species, but the ability of AR420626 to increase IL-10 production was not conserved in the human ILC2s that we tested. We confirm that human ILC2s express EGFR as in mouse, and we suggest that FFAR3 signaling may act in a partially-EGFR dependent manner.

## Discussion

In this study, we present an unbiased approach leveraging mouse genetic diversity to identify a regulator of ILC2 biology, FFAR3, and we demonstate the ability of this receptor to reprogram ILC2s away from a Type 2 inflammatory state into an anti-inflammatory state in mouse and human. An immunomodulatory approach such as this allows alteration of ILC2 inflammatory function without the depletion of ILC2 populations.

These findings help to clarify an important signaling axis in the body that connects the microbiome to innate lymphoid cells. Previous groups have described effects of SCFAs on ILC2s in the gut and lung[60–62], and the report by Thio et al. demonstrated expression of *Ffar3* in ILC2s from IL-33 pre-treated *Rag2*–/– mice[63]. While they established the expression of the gene in ILC2s, they surprisingly did not observe any effect of AR420626 treatment (10 μM) on Type 2 cytokine production from ILC2s stimulated by IL-33 in vitro. There are some notable differences in the experimental approaches that may explain this discrepancy. Thio et al. used ILC2s that have been pre-expanded in vivo with IL-33 prior to isolation, altering their baseline phenotype. Additionally, they use only IL-33 to activate their ILC2s in culture, while our culture system used IL-33, TSLP, and IL-2. Our data suggest that the anti-inflammatory action of FFAR3 signaling is dependent on IL-2, and therefore, this effect was possibly not observed in previously published experiments.

We suggest that FFAR3 on ILC2s represents a potentially important linchpin in the gut-lung axis that could explain the association between intestinal microbiome dysbiosis and asthma[79–81]. This raises the question of what may be the physiological role of a signaling axis that makes ILC2 effector states responsive to microbe-derived SCFAs.

We speculate that this may be a way in which ILC2s, an integral part of the innate Type 2 inflammatory response that evolved to combat helminthic infections, may tune their inflammatory readiness based on the state of the microbiome. Many helminths, such as *Ascaris lumbricoides* have life cycles that involve the gut and the lung, and infections are known to perturb the microbiome and modulate SCFA levels[82,83]. One could imagine the SCFA-FFAR3 axis acting as an early warning system: in settings of microbiome eubiosis, high SCFAs levels shift ILC2s to an anti-inflammatory or maintenance effector state, but helminthic infection and subsequent microbiome dysbiosis may drop SCFA levels and shift pulmonary ILC2s to a pro-inflammatory state of high readiness where they are primed to respond in the event of systemic infection. Further, our results highlight the importance of investigation into the lung microbiome's contribution to immune modulation. While gut-derived SCFAs are present in the serum at millimolar concentrations that can activate FFAR3 and FFAR2 in distant tissues[84,85], SCFAs produced by the lung microbiome may allow for locally elevated concentrations in the subepithelial spaces of the lung that could more directly affect ILC2s[86,87].

Our work also sheds light onto the emerging ILC2 subset that is capable of IL-10 production. Previous reports have described IL-10 producing ILC2s that can be induced in the setting of allergic inflammation and have been associated with clinical response to immunotherapy[72,88]. IL-2, retinoic acid, and neuromedin U have been implicated as factors than can promote the generation of IL-10 producing ILC2s[68], but it is likely that the mechanisms that govern this Type 2 vs. anti-inflammatory ILC2 balance are incompletely understood. We assert that FFAR3 should be considered a potent inducer of IL-10 in ILC2s. Interestingly, this IL-10 production occurs while FFAR3 shifts ILC2s into a unique effector state where they significantly expand their cell numbers but reduce their capability of making Type 2 cytokines. A variety of mechanisms for the anti-inflammatory reprogramming of ILC2s have been proposed, including metabolic changes from fatty acid oxidation to glycolysis, KLRG1 expression, and autocrine IL-10 secretion[73,89,90]. We propose an additional mechanism, one dependent on EGFR upregulation, that might contribute to this unique IL-10 producing effector state. Our data regarding the amphiregulin-EGFR axis on these anti-inflammatory ILC2 may offer some explanation to the protective effect of amphiregulin signaling in several models[91–93].

We highlight the utility of the CC panel and using mouse lines that encompass a greater degree of genetic diversity. The CC provides an elegant and approachable resource to identifying loci that regulate complex phenotypes, which can in turn lead to the implication of specific genes in the physiology of that trait. There exists a myriad of advanced multi-omics approaches that are capable of unbiased screens, but when these resources are applied in mice, they risk missing out on important genetic discoveries due to the lack of diversity in the sparse number of inbred mouse strains that are commonly used. Many mouse studies have translational goals, and the

current pipeline of taking observations in limited inbred mouse lines and applying them to the entire population of genetically diverse humans could certainly be improved. Expanding preclinical research into more genetically diverse mouse populations can help to improve the generalizability of scientific findings before venturing into human work, and it can allow for unbiased target identification techniques that can pick up previously overlooked gene candidates. To our knowledge, few studies have expanded on QTL mapping in the CC with functional characterization of a specific gene target and validation in humans. Our workflow demonstrates the feasibility of this approach.

A limitation of this study is that we did not utilize gene knockout to directly interrogate the function of *Ffar3*. This limitation is inherent to our use of non-traditional strains, as knockouts and transgenic mice are not immediately available on recombinant CC lines. We attempted to generate an *Ffar3* knockout mouse on the CC030 background with a CRISPR/Cas9 based-gene editing approach, but with the recovery of only deceased pups, we concluded that global *Ffar3* deletion was embryonically lethal in the CC030 strain (Supplementary Data 6). Limitations also exist in our gene selection process. While we kept our approach unbiased for many steps, we ultimately selected gene candidates based on potential relevance and applicability, rather than which gene may truly have the greatest QTL effect. This could mean that there are other genes influencing ILC2 biology within the locus that we did not implicate. Another limitation is that there are notable differences in the response to FFAR3 signaling between ILC2s from our mouse and human experiments, especially relating to responses in cell number and IL-10 production. It is possible that this incomplete degree of phenotypic conservation can be explained by the different tissue sources of ILC2s for the experiments; our mouse experiments used lung ILC2s while our human experiments used ILC2s from the peripheral blood. Finally, we suggest that FFAR3 signaling acts through an EGFR dependent mechanism, and we show that FFAR3 signaling increases apoptosis resistance, but our data do not directly implicate EGFR signaling as a mediator of apoptosis resistance in ILC2s.

In summary, we have successfully utilized the genetic diversity of the CC panel to map a locus and identify a regulator of ILC2 function, FFAR3. We demonstrate that FFAR3 agonism reprograms ILC2 to an anti-inflammatory effector state in mouse that is partially conserved in humans. We believe that this study may serve as proof of concept for finding genetic regulators of disease phenotypes, and we hope to highlight that working with genetically diverse mice can increase the breadth of possible discoveries.

## Methods

All research presented in this manuscript complies with all relevant ethical regulations, approved by the Vanderbilt University Medical Center Institutional Animal Care and Use Committee (protocol number: M1800150-01) and the Vanderbilt University Medical Center Institutional Review Board (202162).

### Mice

All Collaborative Cross (CC) recombinant strains used in this manuscript (CC001/Unc, 002/Unc, 003/Unc, 004/TauUnc, 005/TauUnc, 006/TauUnc, 007/Unc, 010/GeniUnc, 011/Unc, 012/GeniUnc, 013/GeniUnc, 016/GeniUnc, 019/TauUnc, 021/Unc, 023/GeniUnc, 024/GeniUnc, 025/GeniUnc, 027/GeniUnc, 030/GeniUnc, 031/GeniUnc, 035/Unc, 036/Unc, 037/TauUnc, 038/GeniUnc, 039/Unc, 040/TauUnc, 041/TauUnc, 043/GeniUnc, 044/Unc, 051/TauUnc, 053/Unc, 057/Unc, 059/TauUnc, 060/Unc, 061/GeniUnc, 071/TauUnc, 072/TauUnc, 074/Unc, 078/TauUnc, 080/TauUnc, 081/Unc) were obtained from the UNC Systems Genetics Core Facility. For clarity, strain suffixes were omitted throughout the manuscript. CC founder strains 129S1/SvlmJ, A/J, C57BL/6J, CAST/EiJ, NOD/ShiLtJ, and WSB/EiJ mice were obtained from Jackson Laboratory. C57BL/6-*Cd22*^tm1Lam^ /J (CD22KO) mice were obtained from Jackson Laboratory. CC030 mice, C57BL/6J, and

CD22KO mice were bred and maintained in our mouse colony. All in vivo and in vitro mouse experiments utilized age-matched mice 8-16 weeks old. Mice used for experiments are female unless otherwise indicated. Animal sex was considered in the study design. Mice were housed in a temperature-controlled room at 22.2 °C on a half day light-dark cycle. Mice were fed a regular diet (PicoLab Laboratory Rodent Diet 5LOD) and tap water was available ad libitum. Mouse experiments were approved by the Institutional Animal Care and Use Committee (IACUC) at Vanderbilt University Medical Center and were conducted according to the guidelines for the Care and Use of Laboratory Animals prepared by the Institute of Laboratory Animal Resources, National Research Council.

### Aeroallergen challenge

Mice were anesthetized with intraperitoneal injection of a combination ketamine and xylazine solution. 5 min after anesthesia, mice were challenged with intratracheal administration of *Alternaria extract* (Greer) or papain (Millipore) dissolved in 80 μL of phosphate buffered saline (PBS). 5 μg of *Alternaria extract* was used for the initial phenotyping of CC strains presented in Fig. 1, but subsequent challenges employed a dosage of 8 μg to allow for the detection of inflammatory cytokines. 10 μg of papain were used, as this was the highest dose that did not elicit alveolar hemorrhage in a titration experiment. Intratracheal administration was performed with an IV catheter tip fixed on a 1 mL TB syringe (BD). Mice were allowed to recuperate from anesthesia on a heating pad before being returned to cages. Alt *Ex* challenge was carried out for 4 consecutive days, while papain challenge was carried out for 3 consecutive days.

### Isolation of cells from mouse tissues

All mice were euthanized with intraperitoneal injection of a lethal dosage of pentobarbital solution and allowed to expire. Mouse lungs were placed in a digestion solution (RPMI 1640, 5% FBS, 1 mg/mL Type IV Collagenase from *Clostridium histolyticum* (Sigma), and 20 μg/mL of DNase I) and minced with scissors. Lungs were digested for 35 min at 37 °C with rotation. Digestion was neutralized with EDTA, and lungs were subsequently ground through cell strainers to obtain a single cell suspension. Mouse bone marrow was obtained from the femurs and tibias of mice after euthanasia. Bones were cleared of muscle and connective tissue, and the ends of the long bones were cut with a scalpel. Bones were placed into 0.5 mL tubes with puncture holes in the bottom. These punctured tubes were fit into 1.5 mL tubes and centrifuged, allowing for the collection of the bone marrow effluent. Bone marrow cells were then passed through cell strainers. Gonadal adipose tissue was excised bilaterally from the mice after euthanasia. Adipose tissue was minced and suspended in digestion buffer (2 mg/mL Collagenase (Sigma C-6885) with 1% FBS in PBS). Digestion was carried out at 37 °C with rotation for 45 min. Digested adipose tissue was passed through a cell strainer and diluted with neutralization buffer (2 mM EDTA with 1% FBS in PBS). Samples were centrifuged, and the supernatant with adipocytes was removed. Colons were harvested from mice and washed with cold PBS. Colons were then opened longitudinally and cut into 0.5 cm pieces, which were subsequently incubated with RPMI containing Penicillin/Streptomycin (Pen/Strep), 5 mM EDTA, 20 mM HEPES, 5% FBS and 1 mM DTT for 40 min at 37 °C with shaking. Contents were then poured through cells strainers, with the flowthrough discarded. Remaining tissue pieces were shaken vigorously in cold RPMI with Pen/Strep, 5 mM EDTA, and 20 mM HEPES and passed through a cell strainer, again discarding flowthrough. Tissue pieces were minced in a beaker and incubated in RPMI Pen/Strep, 5 mM EDTA, and 20 mM HEPES, 0.1 mg/mL Liberase TL (Roche), 0.05% DNase I (Sigma D5025) with stirring for 30 min at 37 °C. Contents were pulled through a 10 mL syringe and poured through a cell strainer. Cells were subsequently washed with RPMI with Pen/Strep, 5 mM EDTA, and 20 mM HEPES, and 0.05% DNase. Cells were then isolated

with a 40%/90% Percoll gradient. Erythrocyte lysis was carried out for all samples with ammonium chloride-based RBC lysis solution (Tonbo) for 5 min before neutralization with cold PBS.

## Analytical flow cytometry for ILC2s and lymphocytes from mouse tissues

Viability staining was carried out in PBS with Live/Dead Aqua Fixable Viability dye for 25 min. Surface staining for single cell suspensions was carried out in FACS staining buffer (3% FBS in PBS). Prior to staining, cells were incubated with Fc block (BD) for 10 min. Surface staining for ILC2 was carried out with Lineage staining cocktail (α-CD3, α-CD4, α-FCεR1, Hematopoietic Lineage Labeling Cocktail, anti-mouse biotin (Miltenyi)), α-CD19, α-CD45, α-CD90, α-CD127, α-ST2, α-ICOS, α-CD25, α-CD22 for 20 min. Streptavidin conjugation to biotinylated antibodies was carried out for 15 min. Surface staining for lymphocytes was carried out with α-CD19, α-TCR β, α-CD4, α-CD8, and α-CD25. Fixation and permeabilization was carried out overnight with the FoxP3 Transcription Factor Staining Buffer Kit (eBioscience, cat: 00-5523-00). Intracellular staining for GATA3 and FoxP3 was carried out for 1 h in PermBuffer (eBioscience). Analysis was conducted in FACS staining buffer on a BD 5-Laser Fortessa or 4-laser Cytek Aurora. A full list of antibodies used in the study is provided in Supplementary Data 7.

## QTL mapping

We utilized 235 mice from 47 Collaborative Cross (CC) strains for quantitative trait locus (QTL) mapping. Genotype data, consisting of a probability matrix with 36 genotype calls (8 homozygous and 28 heterozygous) across 76,689 SNPs, were obtained from https://csbio.unc.edu/CCstatus/index.py?run=FounderProbs, which were then converted into an 8-state allele probability matrix. QTL mapping was performed at each SNP using the qtl2 package in R (version 4.2.1)[94]. A single-QTL genome scan was conducted by regressing the phenotypic trait of interest on the genotype probabilities at each SNP, and LOD (logarithm of the odds) scores were calculated. Genome-wide significance was evaluated empirically through 1000 permutations, and phenotypes with top candidate loci that exceeded a genome-wide significance threshold of 0.05 were identified as significant.

## Flow cytometry sorting of mouse ILC2

Single cell suspensions were obtained from the naïve lungs of mice according to the harvest procedure outlined in this methods section. For the flow sorting experiment described in Fig. 3, ILC2s were sorted separately as biological replicates from the whole lungs of 3 C57BL/6J female mice and 3 CC030 female mice. Single cell suspensions from lungs were enriched for ILC2 with the Lineage Cell Depletion kit, mouse (Miltenyi, cat: 130-110-470) and the CD25 MicroBead Kit, mouse (Miltenyi, cat: 130-091-072), utilizing LS columns (Miltenyi) for both separations. ILC2 for this experiment were identified by surface staining as Lymphocytes (FSC-A vs. SSC-A), singlets by pulse gating, Live/Dead Aqua-, Lineage- (α-CD3, α-CD4, α-FCεR1, Hematopoietic Lineage Labeling Cocktail, anti-mouse biotin (Miltenyi), CD90 +, CD127 +, CD25 +, ICOS +. Sorting was performed on a BD FACS Aria III run on the slowest setting with a 100 μM nozzle. Due to equipment availability changes, ILC2 sorted for assessment of *Ffar3* expression (Fig. 4) and culture (Figs. 4–7) were sorted on a BD FACS Symphony S6 and defined as Lymphocytes (FSC-A vs. SSC-A), singlets by pulse gating, Live/Dead Aqua-, Lineage- (α-CD3, α-CD4, α-FCεR1, Hematopoietic Lineage Labeling Cocktail, anti-mouse biotin (Miltenyi), CD90 +, CD127 +, ST2 +. Cells were again sorted on the lowest setting with a 100 μM nozzle.

## Low-input library preparation and next generation sequencing

ILC2 were sorted into Single Cell Lysis Buffer (Takara), and subsequent library preparation and sequencing was performed at the Vanderbilt Technologies for Advanced Genomics (VANTAGE) core lab. The SMART-Seq v4 Ultra Low Input RNA Kit (Takara, cat: 634888) was used to isolate RNA. cDNA was generated from extracted RNA through reverse transcription. cDNA was fragmented, blunt-ended, and adenylated for adaptor ligation and subsequently PCR amplified. Amplified cDNA libraries were quality controlled by bioanalyzer. The libraries were sequenced using the NovaSeq 6000 with 150 bp paired end reads targeting 50 M reads per sample. RTA (version 2.4.11; Illumina) was used for base calling and analysis was completed using MultiQC v1.7.

## RNA-sequencing data analysis

We conducted RNA-Seq analysis on both C57BL/6J and CC030 mice. Sequencing data quality was initially assessed using FastQC[95], followed by read trimming with fastp[96]. For the C57BL/6J samples, reads were aligned to the mm10 reference genome using STAR[97] with default parameters. For the CC030 samples, reads were aligned to the mm9 reference pseudogenome using STAR with default parameters, after which lapels[98] was used to convert the alignments to mm10 reference coordinates. Gene expression counts were subsequently quantified across all samples using HTSeq[99]. Differential gene expression analysis was performed on the resultant read count matrices using the DESeq2 R package[100]. For the bulk-RNA sequencing experiment presented in Figs. 6, 7, paired differential gene expression analysis was conducted, as biological replicates were cultured separately for both conditions (vehicle vs. agonist). Normalized counts of genes, sample distance matrices, and principal component analyses were conducted with DESeq2, and results were visualized with ggplot2[101] and pheatmap[102]. Differentially expressed genes were visualized in volcano plots with EnhancedVolcano[103]. Mouse genes were converted to human orthologs with biomaRt[104,105]. ClusterProfiler was used to perform gene-set enrichment analysis using Gene Ontology: Biological Processes and WikiPathways curated gene sets and to generate the resultant tree plot[106]. A significant enrichment for the IL-1R signaling pathway is represented as ST2 signaling, as no specific ST2 Wikipathways gene set exists, but the IL-1R signaling gene set describes identical signaling to ST2 downstream of the cell surface receptor. Transcriptomic comparison in Fig. 6 compared the normalized count matrices for ILC2 samples from 3 experiments: IL-10 + ILC2s from Seehus et al. (2017, GSE81882), IL-10 + ILC2s from Howard et al. (2020, GSE158983), and AR420626-treated ILC2s from this manuscript (GSE288176). For the PCA plot in Fig. 6, values in the normalized count matrices were log2-transformed to stabilize variance. We corrected for inter-experiment batch effects with ComBat from the sva R package[107]. A design matrix was specified to retain variation associated with IL-10+ vs. IL-10- status (condition) while adjusting for experiment (batch). The ComBat corrected expression matrix was then subset to IL-10+ samples. PCA analysis included group-wise convex hulls and centroids to highlight sample clustering by experiment.

## CFSE labelling for proliferation quantification

ILC2 were resuspended in 0.1% FBS in PBS immediately after flow cytometry sorting. CFSE was diluted to the cell suspension for a final concentration of 1.5 μM and allowed to incubate at room temperature for 8 min. An equal volume of pre-warmed FBS was added, and the cells were incubated at 37 °C for 10 min to allow for the efflux of unbound CFSE. Cells were washed with 2%FBS in PBS prior to culturing. A small sample of CFSE-labelled cells was saved and immediately fixed with the FoxP3 Transcription Factor Staining Buffer Kit (eBioscience, cat: 00-5523-00) for the Day 0 control.

## ILC2 culture

All ILC2 culture experiments were carried out in RMPI 1640 with 10% FBS, Penicillin/Streptomycin, HEPES, Sodium Pyruvate, and β-mercaptoethanol. Cells were plated in 96-well non-tissue culture treated U-bottom plates (Corning) in a total volume of 250 μL per

plate. For mouse experiments, ILC2 were cultured at a density of 3000 cells per well. Due to low numbers, human PBMC-derived ILC2 were cultured at varying cell densities, with cells equally split from each donor between conditions. ILC2 in all experiments were stimulated immediately upon plating with IL-33 (10 ng/mL, mouse (PeproTech), human (PeproTech)), TSLP (10 ng/mL, mouse (PeproTech), human (PeproTech)), and IL-2 (100 U/mL, human (NIH), used for both mouse and human cells). Experiments involving AR420626 (Sigma) and Gefitinib (Sigma) and retinoic acid (Sigma) utilized DMSO vehicle controls at appropriate concentrations. The concentration of AR420626 in culture was 10 μM unless otherwise specified. Gefitinib was used at concentrations of 5 μM and 10 μM for mouse ILC2 culture and 0.1 μM and 0.5 μM for human ILC2 culture. Retinoic acid was used at a concentration of 1 μM. α-IL-10 antibody (R&D) or isotype control were added at 10 μM, and α-amphiregulin or isotype control was added at 20 μM. Cells were harvested after 48 h for proliferation assessment via CFSE, after 4 days for strain comparison between C57BL/6J and CC030, and 6 days for CC030 drug-treatment and human ILC2 experiments.

### Active caspase staining
ILC2 were harvested from culture and immediately placed in 1 mL of pre-warmed culture media without cytokines. BD Pharmigen Yellow-Green Caspase Probe was added for a final dilution of 1:100,000. Cells were incubated at 37 °C for 45 min protected from light. Cells were washed with media and centrifuged, then resuspended in pre-warmed media and allowed to incubate for an additional 15 min at 37 °C to allow for the outward diffusion of unbound probe.

### Apoptosis detection by annexin V and propidium iodide (PI) staining
Apoptosis detection by Annexin V and PI was carried out with the BD Pharmingen FITC Annexin V Apoptosis Detection Kit I (cat: 556547). ILC2s were harvested from culture at Day 6 and washed in room temperature PBS. ILC2s were resuspended in 100 μL 1x Binding Buffer with Annexin V-FITC (1:500) and PI (1:400) and incubated for 15 min. After incubation, 400 μL of 1x Binding Buffer was added and cells were analyzed by flow cytometry.

### Flow cytometry analysis of cultured ILC2
All ILC2s harvested from culture were stained for viability, and human ILC2s underwent surface staining for EGFR (antibody) for 15 min prior to fixation. Mouse ILC2s were surface stained with ST2 and CD25. Cells were fixed with the FoxP3 Transcription Factor Staining Buffer Kit (eBioscience, cat: 00-5523-00). Intracellular cytokine staining and Ki-67 staining was carried out for 45 min, and cells were not restimulated prior to intracellular cytokine staining. Mouse EGFR staining was performed by incubating cells with concentration of the unconjugated α-EGFR antibody (Invitrogen) for 30 min. Cells were washed and subsequently incubated with a 1:1,000,000 dilution of the secondary antibody for an additional 30 min. 123count eBeads (Invitrogen) were utilized to enumerate ILC2. Cultured cells were analyzed on a 4-laser Cytek Aurora spectral flow cytometer.

### Cytokine detection by ELISA
Cell supernatants were harvested, and IL-5, IL-13, IL-10, and amphiregulin were detected by DuoSet ELISA development systems kits (R&D, cat: IL-5: DY405(Mouse) DY205(Human), DY413(Mouse) DY213(Human), DY417(Mouse) DY217B(Human), DY989(Mouse)) for ILC2 culture. Cytokine detection in mouse lungs was carried out on lung homogenate. Briefly, the left lobe of the lung was isolated from euthanized mice and homogenized in bead homogenizer tubes with silicone beads and Dulbecco's Minimal Essential Media. Homogenization was carried out in a Bead Beater machine, and supernatants of lung homogenate were collected. IL-5 and IL-13 were detected in

mouse lung homogenates by Quantikine (R&D) ELISA kits (cat: M5000, M1300CB). For detected values under the lowest point on the standard curve, one half of the lowest point on the curve was assigned for that sample.

### Cell counts and differential with bronchoalveolar lavage (BAL)
Mice were euthanized and tracheostomy was performed with scissors. A blunt needle in a flexible rubber adaptor was inserted into the trachea and 800 μL of sterile PBS was injected into the lungs with a 1 mL TB syringe. Fluid was withdrawn and cell counts were performed on the sample. 100 μL of the sample were spun onto a glass slide with 20 μL FBS in a CytoSpin machine. Slides were stained with hematoxylin and eosin, and cell differentials were performed by lab members blinded to the experimental groups.

### Ffar3 detection by qRT-PCR
The single cells from the whole lungs of naïve female mice were isolated as biological replicates, and lineage negative cells were enriched with the Lineage Cell Depletion kit, mouse (Miltenyi, cat: 130-110-470). ILC2 were isolated with the flow cytometry sorting strategy outlined in this methods section and were sorted directly into lysis buffer. The CellAmp Whole Transcriptome Amplification kit (Takara, cat: 3734) and TaKaRa Ex Taq Hot Start Version (Takara, cat: RR006A) were used for RNA-isolation and cDNA preparation. Quantitative real-time PCR was carried out with TaqMan Gene Expression assays: *Gapdh* (Mm99999915) and *Ffar3* (Mm02621638_s1) on a QuantStudio 5 Real-Time PCR System (Applied Biosystems) instrument for 45 cycles. Relative gene expression was quantified by calculating $2^{-\Delta\Delta CT}$. ΔCT was calculated from the *Gapdh* housekeeping gene run with each sample.

### α-CD22 antibody treatment
C57BL/6J and CC030 mice were injected intraperitoneally with 250 μg of the α-CD22 blocking and depleting antibody Cy34.1 (BioXCell) or IgG1 Isotype control (BioXCell) in 100 μL of PBS. Mouse lungs were harvested according to the protocol outlined in this methods section 8 days after antibody or isotype control injection.

### Mass spectrometry-based quantification of short-chain fatty acids
Short-chain fatty acids (SCFAs) in serum were quantified by LC-HRMS following derivatization with dansyl hydrazine and EDC. Samples were spiked with deuterated internal standards, extracted, derivatized, and analyzed on a Thermo Q Exactive HF Orbitrap mass spectrometer with Vanquish Horizon HPLC. Quantification was performed using targeted selected ion monitoring (t-SIM) with a mass tolerance of ±5 ppm and calibration against external standards. Chromatographic separation was achieved using a BEH C18 column with a 15-minute gradient and ammonium acetate/acetic acid buffer system.

### Human ILC2 isolation
Phlebotomy was performed on healthy, non-asthmatic participants to recover 50 mL of blood from each donor. Human primary cells used in this study were obtained as de-identified samples under an IRB-approved protocol managed by collaborating investigators. Sex and/or gender information was not provided to the experimenters and therefore could not be incorporated into study design or analysis. As a result, sex-disaggregated analyses were not performed. Informed consent was obtained from all participants. Additional details about participant recruitment and exclusion are available in the reporting summary. PBMCs were isolated using SepMate tubes with Lymphoprep (StemCell). Red blood cells lysis was carried out with ammonium chloride-based RBC Lysis Buffer (Tonbo). Lineage negative cells were enriched with the Lineage Cell Depletion Kit, human (Miltenyi, 130-092-211) and stained for viability with the LIVE/DEAD Fixable Aqua Dead Cell Stain Kit (Invitrogen, cat: L34957) in PBS. Surface staining for

ILC2 was carried out for 20 min in 3% FBS with anti-human lineage cocktail (Biolegend), α-CD45, α-CD127, and α-CRTH2. ILC2 were sorted on a BD FACS Symphony S6 cell sorter as Lineage-, CD45 + , CD127 + , CRTH2 + .

## Statistical analysis

All statistical analysis outside of QTL mapping and transcriptomic analysis was carried out in GraphPad Prism 10 software. Comparisons of two groups with assumed normal distributions were evaluated with a Student's t-test. Comparisons of 3 or more groups with assumed normal distributions and two independent variables and were assessed with a two-way analysis of variance (ANOVA) with Tukey or Sidak's multiple comparisons testing. Comparisons of 3 or more groups with assumed normal distributions and one independent variable and were assessed with a one-way analysis of variance (ANOVA) with Tukey or Sidak's multiple comparisons testing. All measurements represent independent biological replicates unless otherwise indicated in figure legends; paired donor experiments are described as such. Exact sample sizes (n) for each experimental condition are provided in the corresponding figure legends. Non-transcriptomic data did not include covariates in statistical analyses. Data are presented as mean ± standard deviation (SD) unless otherwise specified. P values are reported in the figure legends where applicable. Effect sizes were not calculated. For human ILC2 experiments (Fig. 8B–F) with large variability between donors, the distribution of paired differences was assessed for normality using the Shapiro–Wilk test. When the differences were approximately normally distributed, significance was evaluated with a two-tailed paired Student's t-test. When the Shapiro–Wilk test indicated non-normality, significance was instead evaluated with the Wilcoxon signed-rank test, a nonparametric test that evaluates whether the median difference between paired samples differs from zero while accounting for both the magnitude and direction of changes across donors. For Wilcoxon tests with small sample sizes (n ≤ 7), p-values reflect the exact distribution of rank sums. All tests were two-sided, and p < 0.05 was considered statistically significant. To evaluate consistent directional effects across donors in small-sample experiments (Fig. 8I–L), a one-sided binomial sign test was used to assess whether the number of donors showing an increase in IL-5 or IL-13 following co-treatment with AR420626 and gefitinib (compared to AR420626 alone) exceeded what would be expected by chance. For each donor, the maximum value observed across the two gefitinib concentrations was compared to the AR420626-alone condition. The test was performed with a null probability of 0.5 (i.e., 50% chance of increase), and significance was interpreted at α = 0.05.

## Reporting summary

Further information on research design is available in the Nature Portfolio Reporting Summary linked to this article.

# Data availability

The raw and processed high-throughput RNA-sequencing data generated in this study have been deposited in the NCBI Gene Expression Omnibus (GEO) under the accession code GSE288176. Human ILC2 RNA-sequencing data from Yudanin et al. used in this study are available in GEO under the accession code GSE126107. IL-10 + ILC2 RNA-sequencing data from Seehus et al. and Howard et al. used in this study are available in GEO under the accession codes GSE81882 and GSE158983, respectively. The Supplementary Figs. and tables generated in this study can be found in the Supplementary Information file. All differential gene expression analysis and gene set enrichment analysis datasets generated in this study can be accessed in the Supplementary Data files. All other data are available in the article and its Supplementary files or from the corresponding author upon request. Source data are provided with this paper.

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

## Acknowledgements

National Institutes of Health grant R01 AI124456 (RSP), National Institutes of Health grant R01 AI145265 (RSP), National Institutes of Health grant U19AI095227 (RSP), National Institutes of Health grant R01AI111820 (RSP), National Institutes of Health grant R21AI145397 (RSP), National Institutes of Health grant F30AI176712 (MR), United States Department of Veterans Affairs Biomedical Laboratory Research and Development Service grant 101BX004299 (RSP). We would like to thank the Vanderbilt Technologies for Advanced Genomics (VANTAGE) Core, the Vanderbilt Genome Editing Resource (VGER) Core, and the Vanderbilt Flow Cytometry Shared Resource Core. We would additionally like to thank Dr. Vivian Siegel, PhD for her help with reviewing our manuscript.

## Author contributions

M.R. designed all experiments, performed almost all experiments, analyzed the data, created the figures, wrote the manuscript, and helped secure funding for the project. S.T. helped design experiments, helped perform experiments, provided scientific expertise, and edited the manuscript. Y.H. helped perform QTL mapping and RNA-sequencing data alignment, provided scientific expertise, and edited the manuscript. M.J.C.T., L.M., J.F.G., C.D., W.Z., and D.B. helped perform experiments. C.M.W. helped perform experiments and provided flow cytometry expertise. M.A. helped perform experiments. M.T.S. helped conceptualize the work, helped in data analysis, and edited the manuscript. C.M.T. and J.L. helped perform experiments. J.J. helped with experiments, edited the manuscript, and provided scientific expertise in transcriptomic data analysis. A.P.P. helped with experiments. M.I.P., J.Y.C., S.N.K. helped with experiments and lead procurement of human samples. M.W.C. performed all mass spectrometry experiments. A.E.N., F.Y., and J.AG. .helped conceptualize and perform experiments. D.R.M. and R.M.L. helped house, maintain, transport and design experiments related to Collaborative Cross mice. D.P.C. and D.C.N. helped conceptualize experiments, provided scientific expertise, and edited the manuscript. F.Z. provided expert guidance on QTL mapping and sequencing experiments as well as helping to secure funding for the project. R.S.P. oversaw all experiments, guided the project, provided scientific expertise, helped to secure funding, and edited the manuscript.

## Competing interests

The authors declare no competing interests.
