## [Transparent Peer Review file · Nature Communications]

Genetic diversity of Collaborative Cross mice reveals FFAR3 as a target for ILC2 anti-inflammatory reprogramming

Corresponding Author: Dr R. Stokes Peebles

Version 0:

Reviewer comments:

Reviewer #1

(Remarks to the Author)

Rusznak *et al.* demonstrate in this manuscript that FFAR3 is a factor to expand lung ILC2 by aeroallergens by comparison CC030 and C57BL/6J strain mice showing high and low expansions and that FFAR3 does this by upregulating EGFR. Many interpretations in this article are unreasonable and the present data are not enough to draw these interpretations.

Major

1. In Figure 1 and Figure 2, the authors have drawn interpretations about the mouse strains and the chromosome loci. However, they did it only based on the ILC2 numbers expanded by *Alternaria* extract. If they want to compare the ability to expand ILC2, they should count the basal numbers of ILC2 and calculate the ratios of ILC2 expansion.
2. We cannot exclude the possibility that the acquired immunity interferes ILC2 expansion in *Alternaria* extract-treated mice. If the authors want to identify the ability of ILC2 to expand by itself, they should use Rag1/2-deficient mice to exclude involvement of the acquired immunity.
3. There is no data that FFAR3 is responsible for the different numbers of ILC2 among mouse strains. To draw such conclusion, the authors have to stimulate or block FFAR3 in vivo using genetic or pharmacological methods leading to upregulation or downregulation of the ILC2 numbers, respectively. Moreover, the authors should examine expression of FFAR3 in lung ILC2 of each mouse strain and show that it is parallel with expanded numbers of ILC2 shown in Figure 1.
4. Higher number of ILC2 in CC030 compared to C57BL/6J is detected only in lungs. However, there is no explanation about it. Expression or behaviors of FFAR3 in other organs are different from those of lungs in CC030 mice?
5. The authors have drawn an interpretation that CC030 mice have decreased apoptotic activity based on the result that active caspase positive cells were decreased in CC030 mice compared to C57BL/6J mice. It is unreasonable and not enough. The authors have to estimate and compare apoptotic activities in both lung ILC2 from CC030 mice and C57BL/6J mice.
6. The authors have drawn conclusion that CC030 mice have an ability to expand ILC2 by decreasing apoptotic activity. In contrast, they show that FFAR3 expand ILC2 by upregulating EGFR. The action of EGF is cell proliferation, not inhibition of apoptosis. These two interpretations are contradictory.
7. The human data in Figure are just negative. These do not support the mouse data.

Minor

8. If the used dose of *Alternaria* extract for CAST/EiJ mice in Figure 1 is lethal, the authors should use a lower, not lethal, dose.
9. In Figure 3K, there is no comparison with CC030.

10. If C57BL/6J mice do not express FFAR3, the authors should use another mouse strain showing poorly expanded ILC2 like C57BL/6J mice.

11. In Figure 6, the authors should show cell fraction data of alive and apoptosis cells.

Reviewer #2

(Remarks to the Author)

In the manuscript by M. Rusznak et al., the authors have undertaken an elegant approach to identify the free-fatty acid receptor 3 (Ffar3/GPR41) as an important regulator of ILC2 cell behaviour.

Via quantitative trait locus mapping to the collaborative Cross (CC) mouse panel, the authors identified a locus on Ch7, which was associated with ILC2 abundance following *Alternaria* extract challenge. Subsequent experiments pinpointed the free-fatty acid receptor 3 (FFAR3) as the key mediator in reprogramming ILC2s. Finally, the authors propose that the FFAR3 mediated effects occurred via an EGFR-dependent mechanism, and that FFAR3 may represent a potential therapeutic target for Type 2 inflammation in humans.

The manuscript is well written and the figures clearly presented. Although, the data is generally supportive of the authors' conclusions, a few open points remain.

Fig1a, shows ILC2s from different collaborative Cross strains quantified as a percentage of live cells in the lung post *Alternaria* challenge. Considering that this model involves both eosinophil (EOS) and lymphocyte recruitment, which could significantly affect the relative percentages, please provide the ILC2 data as the total number of ILC2s in the lung. Furthermore, what were the baseline ILC2 levels in naïve mice across the different strains? Finally, if available, please include data on EOS, neutrophil and lymphocyte numbers post challenge. This would be important to distinguish ILC2-specific changes from general immune cell recruitment.

Do CC030 mice have any other response to *Alternaria* challenge e.g. neutrophil counts, or is the entire immune response to *Alternaria* repressed?

Please speculate on the reasons behind the lower expression of ST2 in CC030 ILC2, is there any relationship to the Ch.7 ILC2 Locus?

Based on the authors hypothesis concerning the importance of FFAR3, and consequently EGFR signalling in ILC2 cells, it would be expected that C57BL/6J ILC2 would not respond to AR420626, as they do not express Ffar3. To support this hypothesis, please provide data demonstrating the response (or lack thereof) of C57BL/6J ILC2s to AR420626 for key readouts, such as type 2 cytokine levels (e.g., IL-5 and IL-13), cell numbers, proliferation, and apoptosis. This data will be crucial in confirming the role of FFAR3 in mediating ILC2 responses through EGFR signaling.

Additionally, please confirm the absence of Ffar3 expression in C57BL/6J ILC2 by qPCR and ideally flow cytometry in Fig 5DE.

It is unclear why the authors presented the data in in Fig 8 "as normalised to vehicle", when all other figures give absolute values for all graphs. Please present this figure as absolute values.

The connection and data presentation between Fig 8F and Extended Data Figure 6 is currently confusing. It appears that Donor 4 From (Extended Data Figure 6) is also presented in Fig 8F the main manuscript, and was chosen for the main figure as it was the one donor from four that exhibited a similar increase in EGFR receptor expression in response to AR420626 as observed in mouse ILC2. For transparency, please include in the main figure a summary graph showing the difference between EGFR levels between Vehicle and AR420626 after 6 days of culture.

Please show whether the AR420626 effect in human ILC2 is dependent on EGFR signalling.

In contrast to mouse ILC2, which increased total numbers in response to AR420626, human ILC2 responded with a decrease. Please discuss the reasons behind this difference.

In the 2018 paper by Thio et al. (DOI: 10.1016/j.jaci.2018.02.032), the authors detail the role of short-chain fatty acids (SCFAs) in allergic airway inflammation and the induction of GPR41/FFAR3 in ILC2s by IL-33. This conflicts with the statement in the current study regarding the lack of knowledge on FFAR3's role in regulating ILC2 biology in the lung. Please revise the associated statements to accurately reflect the prior knowledge presented by Thio et al.

Furthermore, the implications of Thio et al.'s findings should be thoroughly discussed in the context of the current study. Notably, the previous study did not observe a reduction in IL-13 and IL-5 levels when ILC2s were treated with the FFAR3 agonist AR420626. This raises important questions about the regulatory mechanisms of FFAR3 in ILC2 function and its potential differential effects on cytokine production.

Minor

From the results, it is slightly unclear whether the Ch.7 ILC2 Locus was the only locus that exceeded the significance threshold. Please clarify in the text.

Please clarify whether strains CC026 and F-A/J also possessed the Ch.7 ILC2 Locus.

Fig 7B please label the top10 up and down genes and not just Egfr

Please change "dose" to "concentration", when talking about in vitro experiments.

The citation for Ref 35 is incomplete.

In Extended Data Figure 3, there is no concentration-dependent manner observed, please correct the phrasing or include

additional concentrations.

Reviewer #3

(Remarks to the Author)

In the manuscript "Genetic diversity of Collaborative Cross mice reveals FFAR3 as a target for ILC2 anti-inflammatory reprogramming," Rusznak et al. propose a novel role for FFAR3 in regulating ILC2 responses. By examining genetically diverse mouse strains, they identify a locus that influences ILC2 abundance in the lung following allergen exposure. Despite increased ILC2 numbers, these cells show lower activation and diminished type 2 cytokine production. The study further highlights FFAR3 as crucial for inducing a less inflammatory ILC2 state with enhanced IL-10 production. While these findings are promising and the manuscript is broadly well-constructed, there are several critical issues and several points that require further investigation to increase robustness of study:

1. All experiments rely on one allergen (*Alternaria alternata*). Testing additional allergens (e.g., papain or HDM) would strengthen manuscript and clarify whether this phenotype is general or unique to *A. alternata*.
2. Because ILC2s in CC030 mice appear less responsive to IL-33, experiments should be conducted under diverse cytokine conditions, possibly excluding IL-33 or adding other stimuli to confirm whether the observed response (lower proliferation and cytokine production) depends on specific culture conditions.
3. The study often reports small sample sizes (e.g., n=3) from what appear to be single experiments (not independent experiments), raising concerns about statistical power. More independent experiments with larger n are needed, particularly for critical conclusions drawn from subtle cell-number differences.
4. Different mouse strains can have distinct microbiomes that influence ILC2 phenotypes, especially since FFAR3 responds to microbiome-derived fatty acids (FFAs such as acetate, butyrate, propionate act as ligands for this receptor). The authors should detail backcrossing or husbandry methods used to control for these variables.
5. Additional data on how FFAR3 agonism alters ILC2 phenotype—particularly regarding pathways related to IL-10—would strengthen the study. Discrepancies between Figures 6 and 7 in IL-10 expression levels were also unclear.
6. The claim that FFAR3 signaling maintains cell populations by inhibiting programmed cell death requires more mechanistic support, as the reported changes in caspase expression and viability appear modest.
7. For the human ILC2 experiments, raw cell numbers and viability assessments at the endpoint should accompany fold-change data. Moreover, it remains unclear whether IL-10 induction, observed in mice, extends to human ILC2s. This is central/key for the overall stated significance of FFAR3/possible therapeutic interventions proposed in discussion etc.
8. To strengthen novelty, the authors could compare the transcriptional signature of CC030 ILC2s to known ILC210 profiles, including those in studies that link IL-10 production with altered metabolism. For example :
 - Seehus CR, Kadavallore A, Torre B, Yeckes AR, Wang Y, Tang J, Kaye J. Alternative activation generates IL-10 producing type 2 innate lymphoid cells. *Nat Commun.* 2017 Dec 1;8(1):1900. doi: 10.1038/s41467-017-02023-z. PMID: 29196657; PMCID: PMC5711851.
 - Howard E, Lewis G, Galle-Treger L, Hurrell BP, Helou DG, Shafiei-Jahani P, Painter JD, Muench GA, Soroosh P, Akbari O. IL-10 production by ILC2s requires Blimp-1 and cMaf, modulates cellular metabolism, and ameliorates airway hyperreactivity. *J Allergy Clin Immunol.* 2021 Apr;147(4):1281-1295.e5. doi: 10.1016/j.jaci.2020.08.024. Epub 2020 Sep 6. PMID: 32905799.
9. In vivo validation of targeting FFAR3 therapeutically would provide proof-of-principle of the authors' suggestion that targeting FFAR3 may mitigate allergic inflammation. Otherwise discussion needs to be tempered.
10. With IL-10 production highlighted, testing other known IL-10-inducing factors would confirm FFAR3's role in driving this phenotype. Do they also upregulate FFAR3? How do they compare in inducing IL-10 head-to-head with FFAR3 agonist?
11. The potential co-expression of amphiregulin also remains an intriguing avenue for CRISPR or siRNA-based mechanistic studies. Is amphiregulin and IL-10 co-expressed? Does knocking down amphiregulin decrease IL-10 production?

Collectively, these points suggest that deeper mechanistic and translational data will strengthen the manuscript. Addressing these concerns will provide a more robust foundation for the authors' conclusion that FFAR3 is a key regulator of anti-inflammatory reprogramming in ILC2s.

Version 1:

Reviewer comments:

Reviewer #1

(Remarks to the Author)

The content has been significantly improved in the revised version. It can be easily read and the importance of FFAR3 in the anti-inflammatory reprogramming of ILC2 has been clearly described.

I want the authors to correct one point.

The potential physiological role of functional regulation of ILC2 by FFAR3 has not been mentioned in the present version. I assume that anti-inflammatory regulation by the short-chain fatty acid/FFAR3 system may be equipped to dampen inflammation and/or that microbes may secrete FFAR3 ligands to escape from the host defense system via FFAR3. The authors should mention the possibility quoting appropriate reference(s).

Reviewer #2

(Remarks to the Author)

The authors have responded positively to the prior reviews and have addressed my concerns satisfactorily.

Minor comments

Please add the %variance to the x and y axis labels in Fig 6L

Reviewer #3

(Remarks to the Author)

I greatly appreciate the effort the research team put forward to address prior concerns. The revisions have been thoughtful and fully address prior comments. I believe the manuscript is now suitable for publication.

I have only two minor requests:

1. Please include Response to Reviewer Figure 4, either within the Methods section or as a Supplemental Figure, and add one or two sentences in the main text describing these data, as they are important to document.
2. I would suggest tempering the heading "FFAR3 signaling reprograms human ILC2s towards an anti-inflammatory effector state in vitro", to "FFAR3 signaling partially reprograms human ILC2s towards an anti-inflammatory effector state in vitro", to not overstate the conclusion.

Response to Reviewers
Rusznak *et al.*

We greatly appreciate the opportunity to have had our manuscript reviewed at *Nature Communications* and want to thank the reviewers and editor for their very insightful and helpful comments. We have performed the many additional experiments requested and believe that the manuscript is much stronger as a result of the reviewers' suggestions. We are grateful for the opportunity to submit this revised manuscript which we hope will now be of sufficient impact to warrant publication in *Nature Communications*. Please let us know if there are further questions that we can answer.

Reviewer 1

Rusznak *et al.* demonstrate in this manuscript that FFAR3 is a factor to expand lung ILC2 by aeroallergens by comparison CC030 and C57BL/6J strain mice showing high and low expansions and that FFAR3 does this by upregulating EGFR. Many interpretations in this article are unreasonable and the present data are not enough to draw these interpretations.

We appreciate the reviewer's concern for some of the interpretations that we have drawn from the data, and we are grateful for the opportunity to revise the manuscript to address these concerns. We have significantly expanded on our original data, and we have made corrections to many of our interpretations highlighted below. We hope that our responses may offer some clarification on the claims and rationale presented in the manuscript.

Major Comments

1.) In Figure 1 and Figure 2, the authors have drawn interpretations about the mouse strains and the chromosome loci. However, they did it only based on the ILC2 numbers expanded by *Alternaria* extract. If they want to compare the ability to expand ILC2, they should count the basal numbers of ILC2 and calculate the ratios of ILC2 expansion.

R1C1: The reviewer raises an important point regarding the quantitative phenotype that we are mapping in the experiment. The reviewer is correct that mapping ILC2 expansion would require a ratio of ILC2 prevalence after aeroallergen challenge over ILC2 prevalence at baseline for every strain. This phenotype would be very useful in describing the ability of ILC2s to expand their numbers in response to aeroallergen challenge; however, the phenotype that we are describing in this manuscript is the prevalence of ILC2s in the setting of challenge, not their degree of expansion. For this reason, we believe that our quantitative phenotype of choice is valid for the conclusions that we have drawn. ILC2 prevalence after challenge is a quantitative readout that is a function of both ILC2s at baseline and expansion in response to challenge. Our rationale for using this phenotype was that we could perform our mapping on a single value from a single batch of experiments that reflects the total degree of an immune cell's prevalence in the setting of challenge. We recognize that the high ILC2 prevalence phenotype that we

describe in CC030s is one that is also present at baseline, and we further explore the steady-state differences in ILC2s between CC030 and C57BL/6J in **Figure 3**.

We agree that the expansion ratio is a critically important quantitative phenotype, and while it is not the one that we mapped, it would be very helpful to present. We now provide the ILC2 expansion ratios for the 42 strains for which we were able to obtain baseline and post-challenge ILC2 phenotype data. Due to strain availability for the second batch, we could not calculate expansion ratios for all 48 strains. We have added these new data in **Supplementary Table 1**, and we have added discussion of these data to the results section with the following text:

“Strain differences persisted when quantifying ILC2s after challenge with total cell numbers in the lung (Supplementary Figure 2.A), and we confirmed strain variability in ILC2 prevalence and number in the CC mice at baseline (Supplementary Figure 2.B&C). Expansion ratios for each strain with naïve and challenged ILC2 quantification data are presented in **Supplementary Table 1**.”

2.) We cannot exclude the possibility that the acquired immunity interferes ILC2 expansion in *Alternaria* extract-treated mice. If the authors want to identify the ability of ILC2 to expand by itself, they should use *Rag1/2*-deficient mice to exclude involvement of the acquired immunity.

R1C2: The reviewer highlights an important limitation of using Collaborative Cross strains to assess innate immune phenotypes. The gold standard mouse model for attributing a phenotype to the innate immune response is the *Rag1/Rag2*^{-/-} mouse, where contribution from the adaptive immune system can be confidently excluded due to the lack of T and B cells. Given that the experiments presented in this manuscript relied on using 48 strains of Collaborative Cross mice to perform QTL mapping, it would have been unfeasible to generate *Rag1/Rag2*^{-/-} on every one of the 48 different strain backgrounds. Further, introducing BALB/c or C57BL/6 genes from the *Rag1/Rag2*^{-/-} mice that were crossed with the Collaborative Cross strains would have introduced a great deal of genetic heterogeneity into the experiment which would have made the experiment impossible as it would have required many generations of breeding to replicate the Collaborative Cross genotypes that we did study.

3.) There is no data that FFAR3 is responsible for the different numbers of ILC2 among mouse strains. To draw such conclusion, the authors have to stimulate or block FFAR3 in vivo using genetic or pharmacological methods leading to upregulation or downregulation of the ILC2 numbers, respectively. Moreover, the authors should examine expression of FFAR3 in lung ILC2 of each mouse strain and show that it is parallel with expanded numbers of ILC2 shown in Figure 1.

R1C3: The most direct way to demonstrate *Ffar3*'s causal role in the high-abundance ILC2 phenotype in CC030 mice would have been to knockout the gene on the CC030 background. We attempted to create a global *Ffar3*^{-/-} mouse with a CRISPR/Cas9 approach. Despite effective embryo recovery, editing, and transfer into 5 pseudo-pregnant females, only 3 deceased pups were recovered, all of which were positive for the targeted *Ffar3* deletion. These results suggest that global deletion of this gene might be embryonically lethal on the CC030 background. A full report of this gene deletion attempt is provided in **Supplementary Data 5**.

We thank the reviewer for their suggestion that *Ffar3* expression should be evaluated in all strains to correlate expression with ILC2 abundance. There is a wide range of ILC2 abundance between the CC strains that we phenotyped for our study, however, our results do not suggest that the entirety of this variation can be attributed to the *Ch.7 ILC2 Locus* or *Ffar3*. While the *Ch.7 ILC2 Locus* was the only QTL that reached significance, the presence of other peaks suggests that its effect size is not enough to explain the entirety of the variation observed between the strains. A hypothetical QTL is present at a locus when the phenotype of interest is affected by different genotypes at the locus (i.e. founder haplotype). At a genomic locus that does not regulate the phenotype, that locus may be inherited from any of the founders and it will not affect said phenotype. At a locus that does regulate the phenotype (QTL), one or more of the founder genotypes will have either a statistically significant positive or negative contribution to that trait.

Our founder haplotype effect analysis from **Figure 2.B** describes how the different founder haplotypes contribute to the presence of a QTL for ILC2 abundance at the *Ch.7 ILC2 Locus*. The CAST/EiJ founder haplotype has a strongly positive QTL effect, while all other founder haplotypes have very weak positive or negative contributions. This QTL is driven by the CAST/EiJ founder haplotype versus other founder haplotypes that behave similarly, meaning that the CAST/EiJ haplotype is responsible for the majority of the variation in ILC2 abundance that can be attributed to this QTL. This result can be interpreted as: inheriting the *Ch.7 ILC2 Locus* from CAST/EiJ will increase a founder strain's ILC2 abundance compared to inheriting that locus from any other founder strain. As most of the 48 strains that we phenotype do not have the CAST/EiJ haplotype at the QTL, the locus is not contributing meaningfully to their ILC2 abundance. Yet, there is a large phenotypic range even amongst the strains with other haplotypes, suggesting that there are other sources of phenotypic variability outside of locus, which is to be expected given the large number of genes that are known to regulate ILC2 behavior.

In summary, it would be accurate to say that the CAST/EiJ founder haplotype at the *Ch.7 ILC2 Locus* positively contributes to ILC2 abundance, and we suggest that the most relevant consequence of inheriting the CAST/EiJ haplotype of the *Ch.7 ILC2 Locus* is higher *Ffar3* expression. This would mean that higher *Ffar3* expression positively contributes to ILC2 abundance, but a majority of the variation between the strains in ILC2 abundance is not attributable to our QTL. This is demonstrated by the fact that the 7 other founder haplotypes had minimal positive or negative contribution to the QTL effect in **Figure 2.B**, meaning that for the CC strains with the other 7 haplotypes at the *Ch.7 ILC2 Locus*, the specific haplotype that they inherit (WSB/EiJ vs. NOD/ShiLtJ etc.), does not make much of a difference with respect to the QTL. However, there is still substantial variation between these other strains with different haplotypes at the locus. Therefore, while we assert that *Ffar3* regulates ILC2 abundance, the other CC strains with different founder haplotypes at the *Ch.7 ILC2 Locus* would not necessarily have their ILC2 abundance correlating with their *Ffar3* expression, as the variation in these strains is explained by other genetic factors.

4.) Higher number of ILC2 in CC030 compared to C57BL/6J is detected only in lungs. However, there is no explanation about it. Expression or behaviors of FFAR3 in other organs are different from those of lungs in CC030 mice?

R1C4: We appreciate the reviewer inquiring about what may be driving the elevated numbers of ILC2s in the lung compared to other tissues. To assess whether *Ffar3* expression differences may be responsible for the observed difference in ILC2 abundance, we isolated ILC2s from lung, bone marrow, adipose tissue, and colon by flow cytometry and isolated RNA. We then created cDNA libraries from these samples for analysis via qPCR with *Gapdh* as a housekeeping gene.

We have added these data in **Figure 4.T** and discuss them in the results section. There were no significant differences in *Ffar3* expression between ILC2s from any of the 4 tissues, although there may be a trend towards lower expression in colonic ILC2s compared to lung ILC2s. We may conclude from these data that *Ffar3* expression alone is not able to account for the differences in these tissues. Despite comparable levels of gene expression, local differences in ligand concentration (SCFAs) may significantly influence the degree of FFAR3 signaling that is occurring at baseline, which may still contribute to the observed differences in ILC2 abundance.

5.) The authors have drawn an interpretation that CC030 mice have decreased apoptotic activity based on the result that active caspase positive cells were decreased in CC030 mice compared to C57BL/6J mice. It is unreasonable and not enough. The authors have to estimate and compare apoptotic activities in both lung ILC2 from CC030 mice and C57BL/6J mice.

R1C5: We appreciate the reviewer's concern that the current data regarding strain differences in apoptosis lack mechanistic depth. To address this concern, we have employed an Annexin V and propidium iodide-based apoptosis detection assay to more accurately detect and quantify apoptosis in our ILC2 culture. Using this method, we can identify 3 distinct ILC2 populations. We term Annexin V- and PI- cells as healthy ILC2s, Annexin V+ and PI- cells as early apoptotic ILC2s, and Annexin V+ and PI+ cells as late apoptotic/dead ILC2s. With this method, we have another way of quantifying apoptosis and differentiating between early and late phases of programmed cell death. We have added these experiments in **Figure 4 I-K** and have discussed them in the results section.

We conclude from our new data in Figure 4 that CC030 ILC2s indeed have greater viability during cell culture and resist progressing through apoptosis to cell death. CC030 mice have significantly higher overall viability with a decreased fraction of late apoptotic/dead cells compared to C57BL/6J, but they also have an increased percentage of early apoptotic cells compared to C57BL/6J. A possible explanation for this pattern of more early apoptotic cells and fewer late apoptotic cells is that apoptosis may become arrested in its early stages in CC030 ILC2s. While these programmed cell death processes are initiated to enough of a degree that phosphatidyl serine begins to lose its asymmetry and it can be detected with Annexin V staining, it is possible that this apoptotic process does not progress further towards complete cell death, or apoptotic change in these early stages may be reversed. In either situation, you would have a comparatively large population of less fit or hypofunctional ILC2s that are less capable of

carrying out Type 2 effector function and may assume an alternative, less inflammatory effector state.

6.) The authors have drawn conclusion that CC030 mice have an ability to expand ILC2 by decreasing apoptotic activity. In contrast, they show that FFAR3 expand ILC2 by upregulating EGFR. The action of EGF is cell proliferation, not inhibition of apoptosis. These two interpretations are contradictory.

R1C6: The reviewer raises an important concern regarding the somewhat unorthodox manner in which EGFR is behaving in our system. We agree with the reviewer that EGFR signaling on many cell types, especially with EGF as the ligand, contributes to cell proliferation. There is evidence, however, that amphiregulin signaling through EGFR on lymphocytes does not have a proliferative effect and instead affects other cellular functions like cytokine secretion. Zais *et al.* show in their 2018 manuscript that treatment of EGFR+ Tregs with 100ng/mL of amphiregulin (an EGFR ligand) significantly enhanced their suppressive function, but it had no effect on the total number of ILC2s.¹ This published data is provided below:

[editorial note: third party material redacted]

We suggest in our manuscript that amphiregulin acting through EGFR on ILC2s is contributing to primarily non-proliferative cell changes. Florentin *et al.* published data which demonstrate that amphiregulin signaling through EGFR in endothelial cells decreases apoptosis and cell death.² Surprisingly, endothelial cells with cell-specific deletion of *Egfr* experienced a greater degree of proliferation compared to WT controls, suggesting that the amphiregulin-EGFR signaling axis may exert a broader range of downstream effects than previously appreciated. Similarly, the authors demonstrate that amphiregulin decreases expression of the pro-apoptotic factor BCL2-associated agonist of cell death (BAD) in a concentration dependent manner, offering some evidence as to how amphiregulin may be suppressing apoptosis.

While there is published evidence to support the plausibility of our proposed function of amphiregulin and EGFR in our system, we concur with the reviewer that we do not directly implicate EGFR signaling in apoptosis resistance with our current data. We have adjusted the language in the manuscript to acknowledge this limitation:

“Finally, we suggest that FFAR3 signaling acts through an EGFR dependent mechanism, and we show that FFAR3 signaling increases apoptosis resistance, but our data do not directly implicate EGFR signaling as a mediator of apoptosis resistance in ILC2s.”

7.) The human data in Figure are just negative. These do not support the mouse data.

R1C7: We thank the reviewer for the concern about the human data not fully reflecting the mouse data. There are important elements of the murine ILC2 response to FFAR3 signaling that are conserved in humans, but there are some notable differences as well. An important consideration is that the ILC2s used for the mouse studies were isolated from the lung, while the human ILC2s were isolated from PBMCs. We have added data in **Figure 8.B-L** along with textual descriptions to the results section to address these concerns.

Most importantly, we see that AR420626 treatment decreases Type 2 cytokine production in human ILC2s like in mouse. There is a large degree of heterogeneity, but we also observe an increase in EGFR expression in ILC2s from a subset of donors. Additionally, we see that inhibition of EGFR with gefitinib trended towards a reversal of the AR420626-mediated decrease in IL-5 and IL-13. Collectively, these suggest that FFAR3 signaling is acting to decrease the Type 2 cytokine producing capability of these ILC2s, and there is data to suggest that this might be occurring through an EGFR-mediated mechanism.

Differences between the murine and human phenotypes are that FFAR3 agonism did not expand the number of human ILC2s, it did not affect viability, and it did not induce IL-10 production. AR420626 did not significantly affect ILC2 number at 1 μ M or 5 μ M, but it reduced ILC2 numbers at 10 μ M. Notably, AR420626 at 5 μ M had no significant effect on cell number, but it significantly reduced Type 2 cytokines, suggesting that there is a negative effect on IL-5 and IL-13 production out of proportion to the effect on cell number. Our mouse findings showed that FFAR3 agonism increases ILC2 viability, but our human cultures reveal consistently high degrees of viability in both treatment groups. It is reassuring that while a decrease in ILC2 number is observed with AR420626 treatment, there is no decrease in viability even at the highest doses, suggesting against non-specific drug toxicity. Finally, we were unable to reproducibly induce IL-10 production in human ILC2s with AR420626 treatment. From the new mechanistic experiments that we present in **Figure 6**, we assert that the effect of FFAR3 agonism on Type 2 cytokines and IL-10 are uncoupled from one another, meaning that they are separate processes and the decrease in IL-5 and IL-13 is not dependent on IL-10. It is possible that FFAR3 signaling reduces Type 2 cytokines in both species, but its effect on IL-10 is not conserved. It should be emphasized that the human ILC2s were derived from PBMCs, meaning that there are many differences between the human and mouse ILC2s in our study that are attributable to their different tissues of origin. It is possible that the induction of IL-10 in ILC2s with FFAR3 signaling is a phenotype that is specific to ILC2s imprinted by the pulmonary environment, and we were unable to capture this conservation because we were using ILC2s from a different human tissue.

We have addressed this limitation of our study with the following added text in the discussion:

“Another limitation is that there are notable differences in the response to FFAR3 signaling between ILC2s from our mouse and human experiments, especially relating to responses in cell

number and IL-10 production. It is possible that this incomplete degree of phenotypic conservation can be explained by the different tissue sources of ILC2s for the experiments; our mouse experiments used lung ILC2s while our human experiments used ILC2s from the peripheral blood. Additionally, some of our mechanistic data supports the notion that the downstream responses of FFAR3 signaling in ILC2s (decreased Type 2 cytokines, increased cell number through apoptosis resistance, and increased IL-10) are uncoupled from one another, and different arms of this response may be variably conserved across species and ILC2 tissue types.”

Minor Comments

8.) If the used dose of *Alternaria* extract for CAST/EiJ mice in Figure 1 is lethal, the authors should use a lower, not lethal, dose.

R1C8: We agree with the reviewer that having data for the CAST/EiJ founder strain in the initial phenotyping experiment would have been helpful for drawing conclusions, and it was unfortunate that they did not survive our 4-day challenge protocol. We obtained further batches of CAST/EiJ mice to characterize their ILC2 responses in a modified *Alt* Ex challenge model that would not cause excessive mortality. We modified the quantity of *Alt* Ex per intratracheal challenge through multiple trial experiments, but CAST/EiJ mice continued to die in our 4-day model. Given that mice tended to die immediately after challenge, we concluded that the mortality was likely due to an acute complication of the challenge, and decreasing challenge frequency may improve survival. We restricted our challenges to 8µg of *Alt* Ex on Day 0 and Day 2, harvesting the mice on Day 4, and we found that this dose was able to induce detectable cytokine production without significant mortality.

Because our initial *Alt* Ex challenge experiment for **Figure 1.A** was completed with the 4-day model, these results could not be incorporated into the analysis. However, it is interesting to informally compare the ILC2 abundance in CAST/EiJ mice after a reduced-intensity challenge alongside C57BL/6J and CC030 ILC2 abundances from **Figure 1.A**. The CAST/EiJ founder strain has similar ILC2 abundance to CC030 mice and significantly increased ILC2 abundance compared to C57BL/6J. These data are presented below in **Response to Reviewers Figure 1.A**. While statistical significance is evaluated between groups, we caution direct comparison due to the different challenge models used.

Response to Reviewers Figure 1. (A) Comparison of ILC2 abundance between C57BL/6J, CC030, and CAST/EiJ females after *A/t* Ex challenge. C57BL/6J and CC030 groups were challenged with 5µg of *A/t* Ex daily for 4 days, while CAST/EiJ mice were challenged for every other day for a total of 2 challenges. Significance was assessed with a one-way ANOVA with Tukey's multiple comparison test. n=5 for C57BL/6J and CC030, n=4 for CAST/EiJ. Result is representative of one experiment. ns = not significant. ** = p<0.005.

9.) In Figure 3K, there is no comparison with CC030.

R1C9: We thank the reviewer for highlighting an aspect of our gene-set enrichment analysis that could have been clearer in the text. The gene sets in the tree plot in **Figure 3.K** represent cell signaling pathways that were positively enriched in C57BL/6J ILC2s compared to CC030, meaning that the comparison to CC030 is inherent to the analysis.

We have included another tree plot in **Figure 3.K** that shows gene sets which were positively enriched in CC030 compared to C57BL/6J. It is interesting to note that the pathways most upregulated in CC030 ILC2s center around cell-to-cell interactions and membrane-based ion transport. This could suggest that CC030 ILC2s are engaging in some sort of homotypic interactions that may be affecting their collective phenotype.

We now have the following text to clarify the two analyses:

“To capture other differences between these cells in an unbiased manner, we performed hierarchical clustering of the top 30 Gene Ontology: Biological Processes pathways most enriched in C57BL/6J and CC030 ILC2s, ranked by normalized enrichment score (Figure 3.K). The most enriched gene sets in C57BL/6J ILC2s cluster together as pathways related to protein translation, mitochondrial complex assembly, RNA processing, ribosome biogenesis, and oxidative phosphorylation. This GSEA result emphasizes that C57BL/6J ILC2s are more transcriptionally, translationally, and metabolically active compared to CC030 ILC2s, suggesting

their increased general readiness for activation and the propagation of Type 2 inflammation. In contrast, CC030 ILC2s are most transcriptionally enriched for pathways related to cell-cell interactions, cell membrane ion transport, and detection of chemical stimuli.”

10.) If C57BL/6J mice do not express FFAR3, the authors should use another mouse strain showing poorly expanded ILC2 like C57BL/6J mice.

R1C10: C57BL/6J mice were chosen as a reference strain due to the C57BL/6J founder haplotype's negative QTL effect (**Figure 2.B**) at the *Ch.7 ILC2 Locus*. By picking the founder strain rather than a recombinant strain, we can be certain that the entirety of the genome is derived from the C57BL/6J founder. We found in our initial experiments that C57BL/6J mice did not express *Ffar3* in their ILC2s, but it would have been difficult to know which of the low ILC2 abundance strains express *Ffar3* on their ILC2s and which ones don't. As discussed in the response to comment #3, it is not guaranteed that *Ffar3* expression would correlate with ILC2 abundance for strains at the low end of the spectrum.

11.) In Figure 6, the authors should show cell fraction data of alive and apoptosis cells.

R1C11: We have included apoptosis data for **Figure 5.N-P** with the Annexin V/PI staining protocol that allows us to differentiate between early apoptotic (live) and late apoptotic (dead) cells.

Reviewer 2

In the manuscript by M. Rusznak et al., the authors have undertaken an elegant approach to identify the free-fatty acid receptor 3 (*Ffar3*/GPR41) as an important regulator of ILC2 cell behaviour.

Via quantitative trait locus mapping to the collaborative Cross (CC) mouse panel, the authors identified a locus on Ch7, which was associated with ILC2 abundance following *Alternaria* extract challenge. Subsequent experiments pinpointed the free-fatty acid receptor 3 (FFAR3) as the key mediator in reprogramming ILC2s. Finally, the authors propose that the FFAR3 mediated effects occurred via an EGFR-dependent mechanism, and that FFAR3 may represent a potential therapeutic target for Type 2 inflammation in humans.

The manuscript is well written and the figures clearly presented. Although, the data is generally supportive of the authors' conclusions, a few open points remain.

We appreciate the reviewer's enthusiasm for the work presented in the manuscript, and we are thankful for the opportunity to address the reviewer's concerns with additional experiments and discussion. The reviewer's recommendations have substantially increased the robustness and impact of our study.

Major Comments

1.) Fig1a, shows ILC2s from different Collaborative Cross strains quantified as a percentage of live cells in the lung post *Alternaria* challenge. Considering that this model involves both eosinophil (EOS) and lymphocyte recruitment, which could significantly affect the relative

percentages, please provide the ILC2 data as the total number of ILC2s in the lung. Furthermore, what were the baseline ILC2 levels in naïve mice across the different strains?

R2C1: We appreciate the reviewer's insightful comments regarding our use of ILC2 abundance as our quantitative phenotype and some of the limitations of this choice. It is an important point that these ILC2 percentages may be increased or decreased based on changes in the relative abundance of other immune cells after *Alternaria* challenge. To address this concern, we have provided the total numbers of ILC2s for each of the strains presented in **Figure 1.A**. These total ILC2 counts are derived from the same experiment, and they are presented in the new **Supplementary Figure 2.A**. While CC030 mice remain the second highest strain when ranked by total ILC2s, there are noticeable differences in the rank of other strains throughout the range. Because of these differences, it is important to emphasize that the quantitative phenotype to which we associated the *Ch.7 ILC2 Locus* was ILC2 abundance (percentage) after *Alt Ex* challenge. While we go on to demonstrate a broader effect of the QTL and *Ffar3*, the mapping portion of this study is specifically relevant to abundance calculated by percentage.

We further thank the reviewer for inquiring about the ILC2s present at baseline throughout all of our strains. A limitation of our mapping study is that it only takes into account ILC2 abundance after challenge. This single phenotype is useful for comparing the post-activation numbers of ILC2s, but quantifying ILC2s in naïve mice adds important insight into differences between their baseline states. This is especially important as our data in **Figure 3** suggest that many of the phenotypic differences between C57BL/6J and CC030 ILC2s are present at baseline.

We have provided the baseline ILC2 percentages and total numbers for 52 strains of the Collaborative Cross in **Supplementary Figure 1.B&C**. Due to the challenged and naïve experiments being performed at separate times, the availability of strains for acquisition from the UNC Systems Genetics Core was different, and so we could not obtain naïve ILC2 data for the exact same selection of strains presented in **Figure 1.A**. Nonetheless, there is a robust range of differences between strains in both ILC2 abundance and total number at baseline. It is interesting to note that CC030 mice again have the second highest ILC2 number, but they are nearly identical to the highest strain. These results support our conclusions from **Figure 3** that the high ILC2 abundance present in CC030 is largely driven by a phenotype that is present at baseline.

2.) Finally, if available, please include data on EOS, neutrophil and lymphocyte numbers post challenge. This would be important to distinguish ILC2-specific changes from general immune cell recruitment.

R2C2: We agree with the reviewer that it would be valuable to include data on the different types of immune cells that are induced by *Alt Ex* challenge, as these likely have significant variation between the tested CC strains. Regrettably, we did not quantify these other cell types in our challenge experiment from **Figure 1**. We have repeated our *Alt Ex* challenge experiment to more comprehensively profile the differences in immune response between CC030 mice and C57BL/6J to short-course allergen challenge. These data address the reviewer's third comment and are discussed in detail below in section R2C3.

3.) Do CC030 mice have any other response to *Alternaria* challenge e.g. neutrophil counts, or is the entire immune response to *Alternaria* repressed?

R2C3: This is an important question posed by the reviewer regarding the global differences in the immune response of CC030 to allergen challenge. We demonstrate that CC030 mice have broadly diminished innate Type 2 inflammatory response, evidenced by their decreased levels of Type 2 cytokines in vivo, decreased eosinophilia, and diminished Type 2 effector function in cultured ILC2s compared to C57BL/6J. While the Type 2 response is suppressed, it is possible that these mice compensate by driving other types of inflammation. We investigated the immune profiles of CC030 and C57BL/6J mice at baseline and in response to both *Alternaria* extract challenge as well as papain. These new data are provided in **Supplementary Figure 5**.

In the naïve state, we confirmed the increased number of ILC2s in CC030 compared to C57BL/6J, and we further examined differences in lymphocyte subsets. There was no significant difference in total B cells between the two strains, but CC030 mice did have more total CD4⁺ T cells and specifically Tregs, while C57BL/6J had more CD8⁺ T cells. In the BAL fluid, CC030 mice had more total macrophages at baseline compared to C57BL/6J. While C57BL/6J had virtually no cells other than macrophages present on the differential, CC030 mice had detectable neutrophils in the BAL without any allergen challenge (**Supplementary Figure 5.B**).

Allergen challenge revealed a broadly reduced Type 2 inflammatory response with *Alternaria* for CC030, but this difference was less appreciable with papain (**Supplementary Figure 5.C**). Interestingly, there were no differences between strains in papain-induced inflammation except in that CC030 mice had a greater number of total CD4⁺ cells and neutrophils compared to C57BL/6J. With *Alternaria*, however, CC030 mice exhibited a greater number of total B cells, CD8⁺ T cells, CD4⁺ T cells, and neutrophils. CC030 mice had significantly fewer eosinophils, which aligns with the strain's lower ILC2 expansion ratio, IL-5, and IL-13.

It is clear from this experiment that the innate Type 2 response in CC030 mice is suppressed, but it does not appear that the entire immune response to *Alternaria* is inhibited. Allergen-challenged CC030 mice favor a profile of increased lymphocytes and neutrophils, suggesting that this may be a compensatory response in a strain-specific phenotype where Type 2 immunity is blunted. Just as the increased ILC2 phenotype is present in the CC030 mice at baseline, the neutrophilic predominance is similarly present in the absence of allergen challenge.

4.) Please speculate on the reasons behind the lower expression of ST2 in CC030 ILC2, is there any relationship to the Ch.7 ILC2 Locus?

R2C4: We are grateful to the reviewer for bringing attention to the relationship between ST2 and the *Ch.7 ILC2 Locus*. As the reviewer mentioned, CC030 mice have significantly lower expression of ST2 at baseline compared to C57BL/6J. Our assertion throughout the manuscript is that much of the CC030 phenotype, especially related to diminished Type 2 inflammatory potential, can be attributed to the *Ch.7 ILC2 Locus*, and specifically, *Ffar3*. On the reviewer's recommendation, we assessed if ST2 expression was regulated by the locus through agonism of FFAR3. This new data is provided in **Figure 5.F-H**.

We were encouraged to find that FFAR3 agonism with AR420626 in cultured CC030 ILC2s drastically decreased ST2 expression. The total percentage of ILC2s that were ST2+ decreased more than 3-fold, and the MFI of ST2 on ST2+ positive ILC2s also decreased more than 2.5-fold. It was intriguing that ST2 expression, another feature of ILC2 inflammatory reactivity, was inhibited with FFAR3 modulation. This may explain the decrease in IL-5 and IL-13 observed with AR420626 treated ILC2s, and it is another way in which FFAR3 agonism mimics the observed phenotype in CC030 mice. The precise mechanism of how FFAR3 signaling regulates ST2 expression would be an interesting avenue for further investigation.

5.) Based on the authors hypothesis concerning the importance of FFAR3, and consequently EGFR signalling in ILC2 cells, it would be expected that C57BL/6J ILC2 would not respond to AR420626, as they do not express *Ffar3*. To support this hypothesis, please provide data demonstrating the response (or lack thereof) of C57BL/6J ILC2s to AR420626 for key readouts, such as type 2 cytokine levels (e.g., IL-5 and IL-13), cell numbers, proliferation, and apoptosis. This data will be crucial in confirming the role of FFAR3 in mediating ILC2 responses through EGFR signaling. Additionally, please confirm the absence of *Ffar3* expression in C57BL/6J ILC2 by qPCR and ideally flow cytometry in Fig 5DE.

R2C5: The reviewer highlights an important aspect of the manuscript's claim regarding the strain differences in ILC2s that contributed to our selection of *Ffar3* as our primary gene candidate. In our initial RNA-sequencing experiment, we observed that C57BL/6J ILC2s, which had very low numbers in the lung at baseline in the $1-3 \times 10^3$ range, did not express *Ffar3*. Given that C57BL/6J ILC2s did not express *Ffar3* in the naïve state, we hypothesized that they would not respond to AR420626 treatment, and we performed new experiments in **Supplementary Figure 8** to test this hypothesis. We isolated ILC2s from pooled C57BL/6J mice (lungs of four mice were combined for one biological replicate), and we cultured them with IL-33, TSLP, and IL-2 +/- 10 μ M AR420626 in paired biological replicates.

There was no significant difference between vehicle and AR420626 treated C57BL/6J ILC2s in IL-5 or IL-13 in the supernatant, total ILC2s, total live ILC2s, early apoptotic percentage, or late apoptotic/dead percentage (**Figure 8.A-G**). In a separate experiment, there was no difference in proliferation between vehicle and AR420626 treated ILC2s 48 hours after culture in CFSE dilution (**Figure 8.H-J**). The cytokine and cell number results support the conclusion that C57BL/6J ILC2s do not express *Ffar3* at baseline, and they are therefore unresponsive to FFAR3 signaling. We would not expect the ILC2s to respond with changes in proliferation even if FFAR3 signaling was occurring, given that our CC030 culture data showed no difference in CFSE dilution with FFAR3 agonism.

We have added the following text to the manuscript to describe these findings: "We observed a lack of *Ffar3* expression in C57BL/6J ILC2s in our initial sequencing experiment, and we therefore expected C57BL/6J ILC2s to have diminished or no response to FFAR3 agonism. We cultured C57BL/6J ILC2s according to our model in **Figure 5.A** and did not observe a decrease in IL-5 or IL-13, a difference in ILC2 number, or an effect on apoptosis. (**Supplementary Figure 8.A-G**). Similarly, AR420626 treatment had no effect on C57BL/6J ILC2 proliferation, measured by CFSE dilution (**Supplementary Figure 8.H-J**)."

We were unable to perform qPCR on these ILC2s, as the low cell yields allowed only for cell culture, and we felt that the functional assays were more important to address the reviewer's question. To support our claim that C57BL/6J ILC2s do not express *Ffar3* in the naïve state, we have compiled data from 4 other published RNA-sequencing data sets that isolated RNA from C57BL/6J ILC2s. We were careful to ensure that these data were collected on ILC2s that were not pre-expanded in vivo, cultured, or activated in any way. We have presented the normalized read counts for *Ffar3* from Yang *et al.* (GSE125584), Langlais *et al.* (GSE180075), Ricardo-Gonzales *et al.* (GSE117568), and Huang *et al.* (GSE237815) alongside our data in **Response to Reviewers Figure 2**. *Ffar2* expression, which we found to be present in our experiment, has been included as a reference for each of the other datasets. The other experiments found zero (Yang *et al.* & Langlais *et al.*) or very minimal (Ricardo-Gonzales *et al.* (0, 0, 0, 3.3, 10.9) & Huang (0, 0.07) *Ffar3* expression on C57BL/6J ILC2s.

A.

Response to Reviewers Figure 2. Normalized read counts of *Ffar2* and *Ffar3* in RNA from pre-published data of non-stimulated, non-cultured C57BL/6J lung ILC2s isolated by flow cytometry. Read counts were accessed and downloaded from the NCBI Gene Expression Omnibus. Studies were included where bulk RNA-sequencing was performed on ILC2s isolated from the lungs of C57BL/6J mice without any in vivo expansion (e.g. IL-33 administration, allergen challenge) or culture (e.g. in vitro stimulation with cytokines). *Ffar2* normalized read counts are provided as a reference.

Finally, we were unable to reliably detect FFAR3 protein with flow cytometry. Unfortunately, there are no monoclonal antibodies raised against murine FFAR3 that have been appropriately validated in flow cytometry to our knowledge. We attempted staining FFAR3 with a polyclonal antibody (ThermoFisher Cat: FFAR3-FITC) conjugated to FITC previously optimized for immunofluorescence, and a primary human antibody (ThermoFisher Clone: 1D10B7, Cat: 66811-1-IG). We were unable to detect any positive signal in a heterogeneous lung tissue sample with the polyclonal antibody, and we could not optimize our human antibody, as our available secondary antibodies bound non-specifically to the other antibodies that we used for surface staining to identify ILC2s.

6.) It is unclear why the authors presented the data in in Fig 8 “as normalised to vehicle”, when all other figures give absolute values for all graphs. Please present this figure as absolute values.

R2C6: We thank the reviewer for pointing out that our initial mode of presentation for these data was unclear. We had initially chosen this format given the large degree of variability in expansion and cytokine production of human ILC2s between subjects. We noted that the magnitude of the effect of AR420626 was quite consistent between individual donors, but the individual variability made the data difficult to interpret.

We have added to the number of replicates (donors) for our AR420626 treatment experiments in **Figure 8.B-F**, and we now present all of these data with absolute values on log scales to more clearly show the data while preserving the natural variability that exists between human samples.

7.) The connection and data presentation between Fig 8F and Extended Data Figure 6 is currently confusing. It appears that Donor 4 From (Extended Data Figure 6) is also presented in Fig 8F the main manuscript, and was chosen for the main figure as it was the one donor from four that exhibited a similar increase in EGFR receptor expression in response to AR420626 as observed in mouse ILC2. For transparency, please include in the main figure a summary graph showing the difference between EGFR levels between Vehicle and AR420626 after 6 days of culture.

R2C7: We appreciate the reviewer’s careful reading, and we recognize that our previous presentation was not clear. The donor highlighted by the reviewer was in fact selected to be presented in the main **Figure 8** as the example of a donor who responded to AR420626 by increasing EGFR expression. We agree that it would be clearer and more transparent to present EGFR expression for all donors in the main figure, highlighting that it was a heterogeneous response.

We have updated **Figure 8.G** to show EGFR expression for every donor for which it was measured. We defined a meaningful upregulation of EGFR as an increase in MFI of at least 10%. Of the 8 donors examined in 2 experimental batches, 3 donors’ ILC2s responded to AR420626 by upregulating EGFR to varying degrees. These data highlight the heterogeneity of the response of human ILC2s and suggest that FFAR3-dependent upregulation of EGFR is conserved in humans, although with significant subject-to-subject variation.

8.) Please show whether the AR420626 effect in human ILC2 is dependent on EGFR signalling.

R2C8: We thank the reviewer for their helpful suggestion to clarify whether FFAR3 signaling acts through EGFR on human ILC2s like in mouse. While there is donor-specific variability in whether or not FFAR3 signaling modifies EGFR expression, the ILC2s of all tested donors expressed EGFR (**Figure 8.G**). We also show that there is a trend towards ILC2s increasing EGFR expression with time spent in culture with activating cytokines like IL-33, TSLP and IL-2 (**Figure 8.H**).

We performed a new experiment in **Figure 8.I-L** where we culture human ILC2s with IL-33, TSLP, and IL-2 with DMSO vehicle, AR420626 alone, or AR420626 + varying concentrations of the EGFR inhibitor gefitinib (0.1 μ M or 0.5 μ M). EGFR inhibition via gefitinib did not reverse the AR420626-mediated decrease in ILC2 number, and it did not have an effect on ILC2 viability (**Figure 8.I&J**). Gefitinib treatment did, however, trend towards reversing the FFAR3-dependent decreases in IL-5 and IL-13. In each of the donors (n=4 for IL-5, n=3 for IL-13 because one donor had all conditions under limit of detection for IL-13 ELISA), the AR420626 + gefitinib condition had higher IL-5 and IL-13 than the AR420626 alone condition. The gefitinib concentration at which the maximal increase in IL-5 and IL-13 occurred varied between the donors, but gefitinib increased IL-5 an average of 2-fold and IL-13 an average of 1.88-fold.

To evaluate whether the observed increases in IL-5 and IL-13 with gefitinib were consistent across donors, we performed a binomial sign test comparing each donor's IL-5 or IL-13 level in the AR420626-alone condition to the highest value observed with either concentration of gefitinib. This approach asks whether the number of donors showing an increase with gefitinib exceeds what would be expected by chance (i.e., 50% probability). All tested donors exhibited higher IL-5 (4/4) and IL-13 (3/3) levels with at least one dose of gefitinib compared to AR420626 alone. Binomial testing yielded $p = 0.0625$ for IL-5 and $p = 0.125$ for IL-13, indicating a reproducible trend toward reversal of AR420626-mediated suppression despite limited sample size.

9.) In contrast to mouse ILC2, which increased total numbers in response to AR420626, human ILC2 responded with a decrease. Please discuss the reasons behind this difference.

R2C9: The thank the reviewer for this important observation, and it prompts discussion about the degree of conservation of our phenotype between mouse and human ILC2s. The most straightforward explanation for the opposite direction of the response in ILC2 number between our mouse and human experiments is the fact that they use ILC2s taken from different tissues within their respective organisms. Our mouse experiments were all performed using ILC2s isolated from the lung, while our human experiments used ILC2s isolated from the peripheral blood. It is a well-documented phenomenon that ILC2s exhibit a significant degree of tissue heterogeneity and their phenotypes are substantially molded by the environments in which they reside.³ While human lung ILC2s perhaps from deceased tissue donors would have allowed for the most appropriate inter-species comparison, this was not possible for our study due to practical and technical considerations.

It would appear that FFAR3's negative effect on Type 2 cytokine production is the most robustly preserved element of this phenotype, as it remains present in ILC2s across species and tissues of origin. It is likely that the three major effects of FFAR3 signaling that we observe in CC030 ILC2s (decreased Type 2 cytokines, increased cell number through apoptosis resistance, and increased IL-10) are independent downstream consequences of FFAR3 signaling. Our new **Figure 6.F-I** shows that the downregulation of IL-5 and IL-13 is not a result of increased autocrine/paracrine IL-10 signaling, and our human data shows Type 2 cytokine effects and variable EGFR responses without increasing cell number. Furthermore, our data from the C57BL/6J mice in Experiment 2 (**Supplementary Figure 8**), discussed in R2C5, shows an increase in cell number via apoptosis suppression but no Type 2 cytokine response. We suggest

that the different aspects of the FFAR3-mediated response in ILC2s may be uncoupled from one another, and the different responses are variably conserved between ILC2s of different genotypes, species, and tissues. It is reassuring, however, that the most therapeutically beneficial aspect of FFAR3-signaling, decreasing IL-5 and IL-13 production, is present in mouse and humans across multiple ILC2 tissue types.

We have added discussion around this topic in the manuscript in the discussion which reads:

“Another limitation is that there are notable differences in the response to FFAR3 signaling between ILC2s from our mouse and human experiments, especially relating to responses in cell number and IL-10 production. It is possible that this incomplete degree of phenotypic conservation can be explained by the different tissue sources of ILC2s for the experiments; our mouse experiments used lung ILC2s while our human experiments used ILC2s from the peripheral blood. Additionally, some of our mechanistic data supports the notion that the downstream responses of FFAR3 signaling in ILC2s (decreased Type 2 cytokines, increased cell number through apoptosis resistance, and increased IL-10) are uncoupled from one another, and different arms of this response may be variably conserved across species and ILC2 tissue types.”

10.) In the 2018 paper by Thio et al. (DOI: [10.1016/j.jaci.2018.02.032](https://doi.org/10.1016/j.jaci.2018.02.032)), the authors detail the role of short-chain fatty acids (SCFAs) in allergic airway inflammation and the induction of GPR41/FFAR3 in ILC2s by IL-33. This conflicts with the statement in the current study regarding the lack of knowledge on FFAR3's role in regulating ILC2 biology in the lung. Please revise the associated statements to accurately reflect the prior knowledge presented by Thio et al.

R2C10: We sincerely apologize for the oversight regarding the previously published data demonstrating *Ffar3* expression in ILC2s by Thio *et al.* We have amended the manuscript to reflect this prior knowledge with the following text:

“One study in particular established the expression of FFAR3 on pulmonary ILC2s in mice, but their data suggested that SCFAs modulated ILC2s primary in a receptor-independent manner, likely through histone de-acetylase inhibition.⁴”

11.) Furthermore, the implications of Thio et al.'s findings should be thoroughly discussed in the context of the current study. Notably, the previous study did not observe a reduction in IL-13 and IL-5 levels when ILC2s were treated with the FFAR3 agonist AR420626. This raises important questions about the regulatory mechanisms of FFAR3 in ILC2 function and its potential differential effects on cytokine production.

R2C11: We agree with the reviewer regarding the importance of discussing our findings in the context of this previous report that examined the response of FFAR3 expressing ILC2s to AR420626 at the same concentration that we used in our paper. Thio *et al.* demonstrate in their 2018 *JACI* paper that ILC2s taken from mice on the BALB/c background express *Ffar3*.⁴ Importantly, these mice were pre-expanded with in vivo-administered IL-33 prior to isolation, and upon stimulation with IL-33 in culture, they upregulated *Ffar3* more than 4-fold after 6 hours. These data suggest that ILC2 activation may upregulate *Ffar3* expression. These data from Figure 6 in their manuscript are presented below:

[editorial note: third party material redacted]

It is very intriguing, albeit somewhat perplexing, that they report an ILC2 experiment where they detect *Ffar3* transcript in their ILC2s but do not observe a response to AR420626 in IL-5 or IL-13 in culture. They did investigate differences in proliferation by Ki-67 positivity which showed no effect of AR420626, and this aligns with our data in both CC030 and C57BL/6J ILC2s. Notably, they did not report cell number or apoptosis data, and it would be interesting to know if these readouts were affected in their culture model. All of this is contrasted with CC030 ILC2s that reproducibly show cell number increases, apoptotic resistance, decreased IL-5 and IL-13, and IL-10 induction in response to treatment with the FFAR3 agonist AR420626.

It is interesting to consider what might be contributing to the observed differences in response to AR420626 between ILC2s that express *Ffar3*. It is important to recognize that Thio *et al.*'s and our experimental setups use different mouse strain backgrounds with ILC2s in different baseline states pre-culture: Thio *et al.* use ILC2s from BALB/c that have been pre-expanded in vivo with IL-33 and we use ILC2s from CC030 mice in a naïve, unstimulated state (**Figures 4-7**). It is possible that the *Ffar3* expression in the ILC2s from Thio *et al.* is transient and low (only relative expression to ILC2s cultured without IL-33 was reported), resulting in only minimal signaling taking place. Another notable difference is that their culture system only uses IL-33 to stimulate ILC2s. From our data in **Figure 6.B**, it seems that IL-2 signaling is necessary for AR420626 to decrease IL-5 secretion. We have added the following text to the manuscript to present some of this discussion:

“...and the report by Thio *et al.* demonstrated expression of *Ffar3* in ILC2s from IL-33 pre-treated *Rag2^{-/-}* mice.⁴ While they established the expression of the gene in ILC2s, they surprisingly did not observe any effect of AR420626 treatment (10 μ M) on Type 2 cytokine production from ILC2s stimulated by IL-33 in vitro. There are some notable differences in the experimental approaches that may explain this discrepancy. Thio *et al.* use ILC2s that have been pre-expanded in vivo with IL-33 prior to isolation, altering their baseline phenotype. Additionally, they use only IL-33 to activate their ILC2s in culture, while our culture system used IL-33, TSLP, and IL-2. Our data suggest that the anti-inflammatory action of FFAR3 signaling is dependent on IL-2, and therefore, this effect was possibly not observed in previously published experiments.”

Minor Comments

12.) From the results, it is slightly unclear whether the Ch.7 ILC2 Locus was the only locus that exceeded the significance threshold. Please clarify in the text.

R2C12: We appreciate the reviewer’s concern relating to the lack of clarity about which loci achieved significance in the QTL mapping experiment. The *Ch.7 ILC2 Locus* was the only locus to reach any of the permutation thresholds (80%, 90%, or 95%), and it exceeded the 95% permutation threshold.

We have clarified this point in the manuscript with the following text:

“It was the only peak in our analysis to reach significance, and it exceeded the 95% permutation threshold.”

13.) Please clarify whether strains CC026 and F-A/J also possessed the Ch.7 ILC2 Locus.

R2C13: The reviewer requests an important clarification. The *Ch.7 ILC2 Locus* is technically a genomic interval on mouse chromosome 7 from position 30.538129 to 33.94594, so every mouse strain possesses this locus. The difference between the strains is the founder haplotype that the strain has at the *Ch.7 ILC2 Locus*. Put simply, while every strain has the locus, the founder haplotype at the locus is what makes the difference. Our QTL mapping and subsequent founder haplotype analysis (**Figure 2**) revealed that it was the CAST/EiJ founder haplotype that was primarily responsible for the significant QTL at the *Ch.7 ILC2 Locus*, i.e. strains that inherited the *Ch.7 ILC2 Locus* chromosomal interval from the CAST/EiJ founder had significantly higher ILC2 abundance.

The implication of the reviewer’s request is very important. CC026 and F-A/J have the locus, like all strains, but they have different founder haplotypes at the locus. CC026 has inherited the entirety of the *Ch.7 ILC2 Locus* from the A/J founder strain. This information can be obtained from the Locus Probabilities tool available on the UNC Systems Genetics Core website, at <https://csbio.unc.edu/CCstatus/index.py?run=locus>. This tool will detail the founder haplotype probabilities for all CC recombinant strains for any given gene or chromosomal interval. An example of CC026 is provided below, showing that there is a 100% chance that the entirety of the *Ch.7 ILC2 Locus* (based on chromosomal interval) was inherited from A/J:

CC026/GeniUnc AU8026	100.00	0.00	0.00	0.00	0.00	0.00	0.00	0.00	0.00
AU8026f201	100.00	0.00	0.00	0.00	0.00	0.00	0.00	0.00	0.00
AU8026f206	100.00	0.00	0.00	0.00	0.00	0.00	0.00	0.00	0.00
AU8026f208	100.00	0.00	0.00	0.00	0.00	0.00	0.00	0.00	0.00
AU8026m202	100.00	0.00	0.00	0.00	0.00	0.00	0.00	0.00	0.00
AU8026m203	100.00	0.00	0.00	0.00	0.00	0.00	0.00	0.00	0.00
AU8026m204	100.00	0.00	0.00	0.00	0.00	0.00	0.00	0.00	0.00

Each of the 8 columns represent the different founder haplotypes, with the first column (yellow) representing F-A/J. The 100.00 denotes that the entirety of the locus was certainly inherited from A/J, while other strains may have variable percentages if different portions of the locus were inherited from more than one founder.

The A/J strain, which had the 4th highest ILC2 abundance, just after CC026, must definitionally have 100% A/J founder haplotype at the *Ch.7 ILC2 Locus* because it is a founder strain, so the entire genome is A/J founder haplotype. One might be tempted to conclude that two strains with the A/J founder haplotype at the *Ch.7 ILC2 Locus* having high ILC2s might suggest that the A/J haplotype is driving the QTL. However, by the analysis in Figure 2.B, the A/J haplotype has a remarkably neutral contribution to the QTL at the *Ch.7 ILC2 Locus*, visualized by the yellow line being very close to 0 over the range of the locus:

The likely explanation is that the F-A/J haplotype does contribute to high ILC2 abundance, but not at the *Ch.7 ILC2 Locus*. It is very possible that other intervals of the CC026 genome that were inherited from A/J could be contributing to other peaks on the QTL map from **Figure 2.A** that did not quite reach significance. This highlights an important point regarding this analysis. Not all of the variation that exists in ILC2 abundance between CC strains can be explained by the *Ch.7 ILC2 Locus*. ILC2 abundance is a complex trait that is regulated by many different genes that are outside of the *Ch.7 ILC2 Locus*, and some of the other CC strains with high ILC2 abundance might have this phenotype because of other loci positively contributing to this trait.

14.) Fig 7B please label the top10 up and down genes and not just Egfr.

R2C14: We have changed the new **Figure 8.A** to show the 10 most upregulated and downregulated genes in AR420626 treated ILC2s compared to vehicle controls.

15.) Please change “dose” to “concentration”, when talking about in vitro experiments.

R2C15: We apologize for the oversight regarding terminology. We now use “concentration” throughout the manuscript.

16.) The citation for Ref 35 is incomplete.

R2C16: We appreciate the reviewer’s attention to detail regarding the incomplete citation. That citation was a truncated version of the DOQTL R package citation that was described in Gatti *et al.*’s 2014 paper, which is citation #8. We have consulted standard referencing guidelines, which describe that the most appropriate method for citation of the package is to cite the original paper that published the package. Therefore, we now only provide the reference for the 2014 paper.

17.) In Extended Data Figure 3, there is no concentration-dependent manner observed, please correct the phrasing or include additional concentrations.

R2C17: We concur with the reviewer that the provided data does not demonstrate concentration dependent response. We have removed the data.

Reviewer 3

In the manuscript “Genetic diversity of Collaborative Cross mice reveals FFAR3 as a target for ILC2 anti-inflammatory reprogramming,” Rusznak *et al.* propose a novel role for FFAR3 in regulating ILC2 responses. By examining genetically diverse mouse strains, they identify a locus that influences ILC2 abundance in the lung following allergen exposure. Despite increased ILC2 numbers, these cells show lower activation and diminished type 2 cytokine production. The study further highlights FFAR3 as crucial for inducing a less inflammatory ILC2 state with enhanced IL-10 production. While these findings are promising and the manuscript is broadly well-constructed, there are several critical issues and several points that require further investigation to increase robustness of study:

We are encouraged by the reviewer’s optimism for the potential impact of our findings. We thank the reviewer for identifying important issues and opportunities to strengthen the

manuscript. We have performed the experiments requested by the reviewer, and we are confident that this additional work has strengthened the rigor and breadth of our findings.

1.) All experiments rely on one allergen (*Alternaria alternata*). Testing additional allergens (e.g., papain or HDM) would strengthen manuscript and clarify whether this phenotype is general or unique to *A. alternata*.

R3C1: We agree with the reviewer's suggestion for an opportunity to strengthen the manuscript and broaden the applicability of our in vivo findings. To assess whether the CC030 phenotype of diminished Type 2 innate inflammation was broadly present or restricted to *Alternaria* extract challenge, we performed parallel experiments with *Alt* Ex challenge and papain challenge.

We selected 10µg administered over 3 consecutive days, as this was the highest dosage of papain that did not induce alveolar hemorrhage in our C57BL/6J mice. We performed a new challenge experiment with both antigens, as well as unchallenged mice to establish baseline immune profiles, in **Supplementary Figure 5**. Our *Alt* Ex groups for this new experiment largely validated our previous results that CC030 mice have significantly diminished ILC2-mediated Type 2 responses to short-course allergen challenge. A more comprehensive evaluation of the immune response to *Alternaria* revealed that while CC030 respond with diminished eosinophilia, they have an enhanced neutrophilic response compared to C57BL/6J.

Overall, the immune response to papain challenge was diminished compared to *Alternaria* in both C57BL/6J and CC030, especially with regards to Type 2 cytokine production. However, papain challenge still revealed a greater number of total ILC2s and CD4+ T cells in CC030 compared to C57BL/6J. Notably, the increased neutrophilic response of CC030 mice to *Alternaria* was also seen with papain, evidenced by the increased number of total neutrophils in the BAL after papain challenge.

Taken together, some elements of the CC030 response (increased ILC2s, increased CD4+ cells, and increased neutrophilia) are also present with papain challenge. The response to papain was significantly milder in both strains, and so differences in minimal cytokine expression could not be appreciated. While it is possible that the CC030 phenotype is generalizable across other allergens that can elicit innate allergic inflammation, it is most robust with *Alternaria* challenge.

We have added the following text to the manuscript regarding these results:

"We then challenged C57BL/6J and CC030 mice with *Alt* Ex (4-days, 8µg) and papain (3-days, 10µg) to profile each strain's immune response to different protease-containing allergens. C57BL/6J mice had significantly more ILC2s than CC030 mice after papain challenge, mirroring our *Alt* Ex results. While *Alt* Ex challenge resulted in nearly significant decreases in ILC2-expansion ratio and total IL-5, and significant decreases in total IL-13 for CC030 compared to C57BL/6J, papain challenge did not reveal any differences in these outcomes. The paucity of IL-5 and IL-13 with papain challenge would suggest an inability of papain to elicit a robust Type 2 innate response at this dose, which was the highest dose that did not cause alveolar hemorrhage in our model. CC030 mice also had a greater number of B cells, CD8+ T cells, and CD4+ T cells with *Alt* Ex challenge and a greater number of CD4+ T cells only with papain.

There was no significant difference in total Tregs with either allergen. In the BAL fluid, CC030 mice had significantly decreased eosinophilia but increased neutrophilia compared to C57BL/6J with *Alt* Ex challenge, with the neutrophilia alone being conserved in papain challenge. These results suggest that CC030 mice have a diminished innate Type 2 response to *Alternaria* that compensates with neutrophilia over eosinophilia, and some elements of this response are conserved in exposure to other protease containing antigens like papain (**Supplementary Figure 5.C**).”

2.) Because ILC2s in CC030 mice appear less responsive to IL-33, experiments should be conducted under diverse cytokine conditions, possibly excluding IL-33 or adding other stimuli to confirm whether the observed response (lower proliferation and cytokine production) depends on specific culture conditions.

R3C2: We are grateful to the reviewer for suggesting this line of inquiry, as it led us to uncover more about the possible mechanisms that explain the anti-inflammatory effect of FFAR3 signaling in ILC2s. We were similarly curious as to whether this anti-inflammatory, IL-10 producing phenotype was dependent on the presence of specific cytokines. Previous reports have suggested that IL-10 production in ILC2s was dependent on, or at least greatly enhanced by, IL-2 signaling.^{5,6}

To assess whether our anti-inflammatory ILC2 phenotype was dependent on IL-2, we performed an in vitro culture experiment with different cytokine conditions examining AR420626 responsiveness in the presence or absence of IL-2. These new experiments are presented in **Figure 6.A-E**. We were intrigued to find that the anti-inflammatory effect of FFAR3 signaling on ILC2s was dependent on IL-2. AR420626-mediated increases in ILC2 number, decreases in IL-5, and production of IL-10 only occurred in cytokine conditions containing IL-2 (**Figure 6.A-C**). Without IL-2 (only IL-33 + TSLP), vehicle and AR420626 had no difference in total IL-5 production, and the AR420626 group even trended towards a marginal increase in production of the inflammatory cytokine. The IL-10 induction was seemingly the most dependent on IL-2, as there was virtually no IL-10 production outside of the IL-2 + AR420626 group in this experiment.

Intrigued by the apparent dependence on IL-2 of our FFAR3-mediated phenotype, we evaluated whether FFAR3 signaling may somehow affect IL-2 responsiveness in ILC2s. We hypothesized that AR420626 signaling would upregulate CD25 expression on ILC2s, thereby increasing IL-2 signaling. We confirmed our hypothesis by quantifying CD25 expression on ILC2s cultured with IL-33, TSLP, and IL-2 and we found that AR420626 treatment significantly increased CD25 expression roughly 2-fold (**Figure 6.D-E**).

We have added the following text to the manuscript to describe these experiments:

“We were intrigued by the anti-inflammatory cytokine secretion profile that FFAR3 agonism imparted on ILC2s, and we sought to further characterize this effector state. We investigated whether this IL-10 producing state was dependent the stimulatory cytokines used to activate ILC2s in our culture system. A previous report from Seehus *et al.* demonstrated that IL-2 was necessary for IL-10 induction in ILC2s.⁷ We cultured CC030 ILC2s with AR420626 or vehicle in the presence or absence of IL-2, and IL-2 was necessary for AR420626 to increase cell number (**Figure 6.A**), decrease IL-5 production (**Figure 6.B**), and induce IL-10 production (**Figure 6.C**).

Given that FFAR3's anti-inflammatory effect is IL-2 dependent, we hypothesized that FFAR3 signaling may increase IL-2 signaling through upregulation of the high-affinity IL-2 receptor alpha (CD25). We confirmed our hypothesis by observing greater CD25 expression by flow cytometry on ILC2 in the presence of IL-33, TSLP, and IL-2 treated with AR420626 compared to vehicle controls (**Figure 6.D-E**). These data suggest that, like previously described IL-10-producing ILC2s, FFAR3 acts in an IL-2 dependent manner and may induce this state through an upregulation of CD25."

3.) The study often reports small sample sizes (e.g., n=3) from what appear to be single experiments (not independent experiments), raising concerns about statistical power. More independent experiments with larger n are needed, particularly for critical conclusions drawn from subtle cell-number differences.

R3C3: We appreciate the reviewer's concern regarding some experiments with small samples sizes and lack of clarity on the number of independent experiments that were performed to represent each. We have updated the manuscript with repeated independent experiments for those that had small samples sizes, and we have increased the number of biological replicates to improve power and confidence in our results. We have also made sure to clearly and definitively state in the figure legends the number of biological replicates for each experiment with the number of independent experiments that were performed, including whether the data are combined from independent experiments or one experiment representing the results of multiple independent experiments.

Upon improving the power and rigor, the original **Figure 4.O** had heterogeneous results that decreased our confidence in meaningful differences. This experiment was not essential to the story, and in the interest of rigor and reproducibility, we removed it from our manuscript.

"Differences in ILC2 responses in the *Alt* Ex model might have been influenced by strain differences in alarmin cytokine release and airway anatomy, so we employed an intra-peritoneal IL-33 challenge model to identically stimulate CC030 and C57BL/6J ILC2s in vivo. Direct IL-33 challenge grew the number of ILC2s in C57BL/6J but did not in CC030 (**Figure 4.O**)."

4.) Different mouse strains can have distinct microbiomes that influence ILC2 phenotypes, especially since FFAR3 responds to microbiome-derived fatty acids (FFAs such as acetate, butyrate, propionate act as ligands for this receptor). The authors should detail backcrossing or husbandry methods used to control for these variables.

R3C4: The reviewer highlights an important point regarding differences in microbiomes between mice. Much of our study centers around strain differences FFAR3, but the reviewer is correct that the endogenous ligands of FFAR3, short chain fatty acids (SCFAs), are an integral part of the signaling axis in vivo. A unique element of the multiple SCFA-FFAR signaling axes is that the ligands are not produced by the host, but rather by the microbiomes which colonize the host. Therefore, the composition of the microbiome is a critical determining factor for FFAR signaling within an animal.

It is important to consider that ultimately the QTL mapping study that we performed revealed the *Ch.7 ILC2 Locus*, and thereby *Ffar3*, due to genetic differences in the receptor between the

strains. So, while levels of SCFAs may vary between the strains and influence FFAR signaling, these differences would not contribute to the formation of a QTL at a locus containing the genes for FFAR receptors. The fact that a significant QTL was present in the locus containing *Ffar3* suggests that it is genetic differences in the receptor (either expression or other genetic mechanisms) that are principally responsible for the phenotype.

Nonetheless, in comparing the ILC2 phenotypes of two strains, it is important to consider the contribution of possible SCFA differences secondary to variable microbiomes. We did not perform any backcrossing in our facility, as fully inbred strains were obtained directly from UNC. We ensured that mice used in the study had at least two weeks to acclimate before the initiation of any experiment. All mice were kept on the same rack within the same room of our housing facility, fed the same diet from the same communal source and had access to the same water delivery system on the rack.

Despite these measures to ensure a consistent environment and limit possible microbiome variation, there are genetic factors inherent to the strains that may contribute to microbiome differences. For this reason, we elected to quantify SCFAs in the serum to test for differences between C57BL/6J and CC030 mice. We quantified acetate, propionate, and butyrate via mass spectrometry.

Briefly, short-chain fatty acids (SCFAs) in serum were quantified by LC-HRMS following derivatization with dansyl hydrazine and EDC. Samples were spiked with deuterated internal standards, extracted, derivatized, and analyzed on a Thermo Q Exactive HF Orbitrap mass spectrometer with Vanquish Horizon HPLC. Quantification was performed using targeted selected ion monitoring (t-SIM) with a mass tolerance of ± 5 ppm and calibration against external standards. Chromatographic separation was achieved using a BEH C18 column with a 15-minute gradient and ammonium acetate/acetic acid buffer system.

Response to Reviewers Figure 4. (A) Quantification of acetate, propionate, and butyrate in the serum of naïve C57BL/6J and CC030 female mice. Statistical significance was assessed with an un-paired Student's t-test. ns = not significant, ****= $p < 0.00005$.

It was clear from our experiment that there exist substantial differences in the SCFA profiles of C57BL/6J and CC030 mice. In both strains, acetate makes up the overwhelming majority of the SCFAs, and C57BL/6J have a more than 4-fold higher concentration of acetate in their blood compared to CC030 mice. This is particularly interesting, as our study suggests that elevated FFAR3 signaling in CC030 ILC2s at baseline contributes to their unique anti-inflammatory phenotype, which is re-created with FFAR3 agonists. Clearly, the increase in FFAR3 signaling in CC030 mice cannot be explained by an overabundance of the ligand, as SCFA levels are substantially lower compared to C57BL/6J. Seemingly, the CAST/EiJ haplotype at the *Ch. 7 ILC2 Locus* in CC030 mice confers such an increase in FFAR3 signaling that CC030 ILC2s experience substantially more of it despite significantly lower exposure to the ligands.

5.) Additional data on how FFAR3 agonism alters ILC2 phenotype—particularly regarding pathways related to IL-10—would strengthen the study. Discrepancies between Figures 6 and 7 in IL-10 expression levels were also unclear.

R3C5: We thank the reviewer for this suggestion in further investigating the FFAR3-induced phenotype in relation to IL-10. The added studies performed at the reviewer's request to address this point have strengthened the manuscript.

Given that FFAR3 signaling both induces IL-10 production and decreases IL-5 and IL-13, we hypothesized that the anti-inflammatory IL-10 may be acting in an autocrine and or paracrine manner to reduce IL-5 and IL-13 expression. We tested this hypothesis by blocking IL-10 signaling with the use of an α -IL-10 antibody, and this new data is presented in **Figure 6.F-I**. The α -IL-10 antibody had no effect on the total number of ILC2s, IL-5 production, or IL-13 production. We gathered from these data that FFAR3's effect on decreasing Type 2 cytokine production and increasing IL-10 production were two independent processes, and autocrine/paracrine IL-10 signaling could not explain the altered phenotype.

We were encouraged to find that our results had been independently validated by Miyamoto *et al.* in their 2019 paper. The authors similarly observe a decrease in IL-5 expression with their IL-10-producing ILC2 phenotype. However, when they block IL-10 with an antibody or add IL-10 to the culture, there is no effect on IL-5, suggesting that an ILC2's production of IL-5 is independent of IL-10 signaling. Their data is presented below:

[editorial note: third party material redacted]

We further investigated the IL-10-producing phenotype elicited by AR420626 treatment through comparison with other IL-10-inducing agents (retinoic acid) and through analysis of transcriptomic similarity to other, pre-published IL-10 signatures. These points will be comprehensively addressed in R3C8 and R3C10.

The reviewer raises an important point regarding the differences between figures in the relative difference in IL-10 production between vehicle and AR420626 treated ILC2s and total IL-10 produced. There are a few factors that may contribute to difference in IL-10 results between **Figure 5.I** (previously Figure 6) and **Figure 7.F**. Firstly, we have observed a degree of variation in the quantity of IL-10 produced in our cultures between experimental batches. While the ultimate result is always the same (AR420626-treated ILC2s produce more IL-10 than vehicle), the magnitude of this difference has been variable. Additionally, we believe that the magnitude of the difference is affected by the concentration of DMSO that is in the culture system. We have noticed with both of our intended experiments for **Figure 7** with gefitinib, and our experiment in **Figure 6** with retinoic acid (to be discussed in detail in R3C10), higher quantities of DMSO vehicle in the culture due to the presence of other compounds seems to diminish this difference. We again reiterate that the difference is reproducibly present, but the magnitude is subject to inter-experiment variation.

6.) The claim that FFAR3 signaling maintains cell populations by inhibiting programmed cell death requires more mechanistic support, as the reported changes in caspase expression and viability appear modest.

R3C6: We acknowledge the reviewer's concern about the need for more mechanistic support regarding our claim of increased ILC2 number via apoptosis resistance rather than increased cell proliferation. We have provided a new experiment in **Figure 5.N-P** using an Annexin V / propidium iodide (PI) based method to more effectively detect and quantify apoptosis in its various stages. Using this method, we can identify 3 distinct ILC2 populations. We term Annexin V- and PI- cells as healthy ILC2s, Annexin V+ and PI- cells as early apoptotic ILC2s, and Annexin V+ and PI+ cells as late apoptotic/dead ILC2s.

The data in **Figure 5.N-P** show that AR420626 treatment increases the percentage of ILC2s that are undergoing early apoptosis, but it drastically decreases the proportion of cells that progress through the early stage of apoptosis into late apoptosis and cell death. The increase in viability imparted by FFAR3 signaling as measured with this method is striking. The reproducible lack of meaningful difference in proliferation by CFSE dilution and Ki-67+ positivity, paired with the significant decrease in late apoptotic cells, suggests that the difference in cell number induced by FFAR3 signaling can be attributed to survival differences rather than differences in cell division.

This interpretation is further supported by the specific patterns of staining within the Annexin V+/PI+ quadrant. Annexin V binds phosphatidyl serine that loses membrane asymmetry during apoptosis and begins to appear on the cell membrane surface. PI positivity indicates membrane permeability that suggests more advanced apoptosis and cellular decay. It is particularly intriguing that the Annexin V+/PI+ late apoptotic/dead cell quadrant is heterogeneous and has 3

distinct populations within it. These populations can be appreciated in the top right (double positive) quadrant in the flow cytometry plot from **Figure 5.N** reproduced below:

The difference in Annexin V+/PI+ cells between vehicle and AR420626 ILC2s occurs mostly in the weakly PI+ population in the top right quadrant (indicated with the black rectangle). Given that PI positivity indicates membrane compromise, these Annexin V+, weakly PI+ positive cells likely represent cells that are experiencing an intermediate degree of cell membrane compromise compared to the distinctly more PI+, likely necrotic, cells above them. AR420626 reduces cells in this initial stage of membrane compromise, supporting the notion that AR420626 treatment is reducing progression to late apoptosis, not just generally preventing cell death.

7.) For the human ILC2 experiments, raw cell numbers and viability assessments at the endpoint should accompany fold-change data. Moreover, it remains unclear whether IL-10 induction, observed in mice, extends to human ILC2s. This is central/key for the overall stated significance of FFAR3/possible therapeutic interventions proposed in discussion etc.

R3C7: We agree with the reviewer about the importance of showing viability data for each of the human ILC2 experiments to demonstrate that effects on cell number and cytokines are not merely due to non-specific drug toxicity. We now provide the total numbers of ILC2s in each experiment as well as viability data in **Figure 8.B&C**. AR420626 did not affect viability in any of our human ILC2 experiments.

We measured IL-10 production in our human ILC2 cultures and unfortunately found that AR420626 had no effect at the 3 concentrations that we tested (**Figure 8.F**). While the ability of FFAR3 signaling to decrease Type 2 cytokines is present in both mouse and human ILC2s, our data would suggest that the receptor's effect on IL-10 production is not conserved. A possible explanation for the differences in the AR420626-induced phenotype is that our mouse and human experiments use ILC2s taken from different tissues within their respective organisms.

Our mouse experiments were all performed using ILC2s isolated from the lung, while our human experiments used ILC2s isolated from the peripheral blood. It is a well-documented phenomenon that ILC2s exhibit a significant degree of tissue heterogeneity and their phenotypes are substantially molded by the environments in which they reside.³ While human lung ILC2s from deceased tissue donors perhaps could have allowed for the most appropriate inter-species comparison, this was not possible for our study due to practical and technical considerations. It is entirely possible that the ability of ILC2s to produce IL-10 in response to FFAR3 signaling is a reaction that is specific to the environment of the lung, and therefore human ILC2s taken from other tissues would not capture this biological feature. Limitations of our human ILC2 model have been added to the discussion with the following text:

“Another limitation is that there are notable differences in the response to FFAR3 signaling between ILC2s from our mouse and human experiments, especially relating to responses in cell number and IL-10 production. It is possible that this incomplete degree of phenotypic conservation can be explained by the different tissue sources of ILC2s for the experiments; our mouse experiments used lung ILC2s while our human experiments used ILC2s from the peripheral blood. Additionally, some of our mechanistic data supports the notion that the downstream responses of FFAR3 signaling in ILC2s (decreased Type 2 cytokines, increased cell number through apoptosis resistance, and increased IL-10) are uncoupled from one another, and different arms of this response may be variably conserved across species and ILC2 tissue types.”

8.) To strengthen novelty, the authors could compare the transcriptional signature of CC030 ILC2s to known ILC210 profiles, including those in studies that link IL-10 production with altered metabolism. For example :

- Seehus CR, Kadavallore A, Torre B, Yeckes AR, Wang Y, Tang J, Kaye J. Alternative activation generates IL-10 producing type 2 innate lymphoid cells. *Nat Commun.* 2017 Dec 1;8(1):1900. doi: 10.1038/s41467-017-02023-z. PMID: 29196657; PMCID: PMC5711851.
- Howard E, Lewis G, Galle-Treger L, Hurrell BP, Helou DG, Shafiei-Jahani P, Painter JD, Muench GA, Soroosh P, Akbari O. IL-10 production by ILC2s requires Blimp-1 and cMaf, modulates cellular metabolism, and ameliorates airway hyperreactivity. *J Allergy Clin Immunol.* 2021 Apr;147(4):1281-1295.e5. doi: 10.1016/j.jaci.2020.08.024. Epub 2020 Sep 6. PMID: 32905799.

R3C8: We thank the reviewer for their excellent recommendation on how we may further demonstrate the novelty of the ILC2 phenotype that we describe in this manuscript. We have followed the reviewer's recommendation and obtained the read counts for the RNA-sequencing experiments presented in the manuscripts by Seehus *et al.* and Howard *et al.*^{7,8} We used the normalized read counts of samples derived from IL-10+ ILC2s to compare with our AR420626-treated ILC2s. These new analyses are presented in **Figure 6.L**.

We investigated transcriptomic similarities and differences between the samples with a PCA analysis. Our method for this analysis is highlighted with the following text which has been added to our methods section:

“The PCA plot in Figure 6 was generated through comparison of the normalized count matrices for ILC2 samples from 3 experiments (IL-10+ ILC2s from Seehus et al. (2017, GSE81882), IL-10+ ILC2s from Howard et al. (2020, GSE158983), AR420626-treated ILC2s from this manuscript (GSE288176). Values in the normalized count matrices were \log_2 -transformed to stabilize variance. We corrected for inter-experiment batch effects with ComBat from the sva R package. A design matrix was specified to retain variation associated with IL-10+ vs. IL-10- status (condition) while adjusting for experiment (batch). The ComBat corrected expression matrix was then subset to IL-10+ samples. PCA analysis included group-wise convex hulls and centroids to highlight sample clustering by experiment.”

It is clear from our analysis that there are notable transcriptional differences between the anti-inflammatory, IL-10-producing state that we present and the other IL-10-producing states that have been reported in the literature. Our PCA analysis is particularly helpful in evaluating these differences, as we use a method to correct for intra-experiment batch effects by retaining variation associated with IL-10 status (made possible with the inclusion of IL-10- (Seehus and Howard) and vehicle-treated (our study) ILC2 samples). This reduces variation that exists due to differences in experimental approach or mouse strain and retains the unique features of the IL-10-producing state from each experiment in relation to the control samples. When grouping samples by this method, it is clear that the IL-10+ ILC2s from Seehus *et al.* and Howard *et al.* are transcriptionally very similar to one another, as seen by the close proximity of centroids of the two groups. In contrast, AR420626 ILC2s from our experiment demonstrate greater distance from the other two experiments on the PCA plot, suggesting a greater degree of transcriptional distinction from previously described IL-10+ ILC2 states. We appreciate the reviewer's suggestion, as these experiments have helped to further solidify the novelty of the anti-inflammatory ILC2 state that we describe in this manuscript.

9.) In vivo validation of targeting FFAR3 therapeutically would provide proof-of-principle of the authors' suggestion that targeting FFAR3 may mitigate allergic inflammation. Otherwise discussion needs to be tempered.

R3C9: We concur with the reviewer that targeting FFAR3 in vivo would provide important proof-of-principle for FFAR3 as a therapeutic option for allergic inflammation. We sought to directly target FFAR3 through our attempt at creating a global *Ffar3*^{-/-} mouse with a CRISPR/Cas9 editing method on the CC030 background. Successful generation of *Ffar3*^{-/-} on the C57BL/6J backgrounds has been reported in the literature, and so we were encouraged that global deletion on the CC030 line would be possible.^{9,10} Despite effective embryo recovery, editing, and transfer into 5 pseudo-pregnant females, only 3 deceased pups were recovered, all of which were positive for the targeted *Ffar3* deletion. These results suggest that global deletion of this gene might be embryonically lethal on the CC030 background. A full report of this gene deletion attempt is provided in **Supplementary Data 5**.

Given that we were not able to directly target FFAR3 in vivo, we have tempered discussion in the manuscript around FFAR3's therapeutic potential in allergic inflammation. We have removed the following text from the manuscript:

"ILC2s are critical mediators of Type 2 inflammation in allergen-exacerbated asthma, and a new signaling mechanism that modulates this inflammation may have therapeutic implications."

"These data suggest that ILC2s could be targeted with FFAR3-specific agonists, or even with SCFA supplementation, to reduce Type 2 inflammation in an appropriate clinical setting, such as asthma."

"In addition to the potential clinical utility,"

10.) With IL-10 production highlighted, testing other known IL-10-inducing factors would confirm FFAR3's role in driving this phenotype. Do they also upregulate FFAR3? How do they compare in inducing IL-10 head-to-head with FFAR3 agonist?

R3C10: This is an excellent suggestion by the reviewer that builds off previous questions regarding the uniqueness of the IL-10 producing state that we elicit with FFAR3 signaling. We showed in **Figure 6** that AR420626-induced IL-10 production is dependent on IL-2 signaling, which was similar to IL-10 producing ILC2s described in other reports. However, through our transcriptomic analyses, we demonstrated that our ILC2s possess a unique transcriptomic signature that distinguishes them from previously described IL-10+ ILC2s. Per the reviewer's suggestion, we investigated AR420626 head-to-head with retinoic acid, which is a well-known inducer of IL-10.⁷

We repeated our ILC2 culture system with IL-33, TSLP, IL-2 and DMSO vehicle, AR420626 (10µM) and retinoic acid (1µM) to assess their cytokine responses head-to-head with new data presented in **Figure 6.J&K**. We were intrigued to find that AR420626 and retinoic acid secreted similar quantities of IL-10, but we were most encouraged by the IL-5 data. While AR420626 decreased IL-5 production compared to vehicle as we have repeatedly seen, retinoic acid significantly increased IL-5 expression compared to vehicle. This suggests that while AR420626 and retinoic acid are both inducers of IL-10, they have opposite effects when it comes to Type 2 inflammatory cytokines like IL-5. As we previously investigated, these two processes are independent, and our data suggest that FFAR3 signaling causes a more comprehensive anti-inflammatory reprogramming than other known IL-10 inducers in ILC2s. We feel that this finding effectively supports the novelty of our finding, as we are unaware of other modulators that can simultaneously increase IL-10 expression while decreasing Type 2 cytokines.

We investigated the reviewer's question regarding whether retinoic acid upregulated FFAR3 on ILC2s. Given these differences in response, we hypothesized that retinoic acid acted in an FFAR3-independent manner and did not influence its expression. We extracted RNA from ILC2s treated with DMSO vehicle, AR420626, or retinoic acid, and we found that neither factor upregulated *Ffar3* expression in the ILC2s. Interestingly, it seems that ILC2 activation and culture for 6 days markedly reduces *Ffar3* expression in all conditions. This may suggest that the re-programming experienced by ILC2s from AR420626 may occur in the earlier phases of culture. The qPCR data for this experiment is presented below:

Response to Reviewers Figure 5. (A) Quantification of *Ffar3* by qPCR in ILC2s isolated by flow cytometry sorting from CC030 mice (naïve) and cultured with IL-33, TSLP, and IL-2 with DMSO vehicle, AR420626, or retinoic acid. Δ CT values for *Ffar3* in each sample were calculated to the housekeeping gene *Gapdh* in the naïve condition, but all cultured cells had no amplification for *Ffar3*. Relative expression was calculated with $2^{-\Delta\Delta CT}$. Significance was assessed with Student's t-test. ns=not significant. ****= $p < 0.00005$.

11.) The potential co-expression of amphiregulin also remains an intriguing avenue for CRISPR or siRNA-based mechanistic studies. Is amphiregulin and IL-10 co-expressed? Does knocking down amphiregulin decrease IL-10 production?

R3C11: The reviewer raises important questions regarding the proposed mechanism of amphiregulin and EGFR signaling on ILC2s. We demonstrate that AR420626 increases the expression of EGFR, and we hypothesized that ILC2s are secreting EGFR ligands, like amphiregulin, to activate EGFR in an autocrine and/or paracrine manner. We measured amphiregulin in our ILC2 culture system, and while it is expressed in both conditions, we found that AR420626 decreases amphiregulin secretion. While AR420626 treatment does not increase amphiregulin, it is possible that the large increase in receptor expression outweighs the decrease in ligand, thereby still causing greater overall EGFR signaling.

We hypothesized that amphiregulin was signaling through EGFR to induce IL-10 expression, so we blocked amphiregulin signaling in our culture with an α -amphiregulin antibody, expecting that this would decrease IL-10 expression compared to isotype control. This hypothesis was supported by our data, which showed that amphiregulin blockade significantly decreased IL-10 expression in AR420626 treated groups. This decrease is modest, and the individual paired samples are shown in **Figure 7.K**. It is possible that this effect is so modest due to the fact that EGFR signaling may continue through other EGFR ligands that are secreted by ILC2s, so while

amphiregulin may be neutralized, other secreted proteins are able to continue signaling. This would be an intriguing line of future investigation to evaluate which EGFR ligands these ILC2s are secreting in our culture system.

Collectively, these points suggest that deeper mechanistic and translational data will strengthen the manuscript. Addressing these concerns will provide a more robust foundation for the authors' conclusion that FFAR3 is a key regulator of anti-inflammatory reprogramming in ILC2s.

We concur with the reviewer that addressing these points has significantly deepened the mechanistic insight and overall robustness of the paper. We are grateful for the careful review and constructive suggestions.

References

1. Zaiss, D. M. W. *et al.* Amphiregulin enhances regulatory T cell-suppressive function via the epidermal growth factor receptor. *Immunity* **38**, 275–284 (2013).
2. Florentin, J. *et al.* Loss of Amphiregulin drives inflammation and endothelial apoptosis in pulmonary hypertension. *Life Sci Alliance* **5**, e202101264 (2022).
3. Spits, H. & Mjösberg, J. Heterogeneity of type 2 innate lymphoid cells. *Nat Rev Immunol* **22**, 701–712 (2022).
4. Thio, C. L.-P., Chi, P.-Y., Lai, A. C.-Y. & Chang, Y.-J. Regulation of type 2 innate lymphoid cell-dependent airway hyperreactivity by butyrate. *J Allergy Clin Immunol* **142**, 1867-1883.e12 (2018).
5. Seehus, C. R. *et al.* Alternative activation generates IL-10 producing type 2 innate lymphoid cells. *Nat Commun* **8**, 1900 (2017).
6. Reid, K. T. *et al.* Cell therapy with human IL-10-producing ILC2s limits xenogeneic graft-versus-host disease by inhibiting pathogenic T cell responses. *Cell Reports* **44**, (2025).
7. Seehus, C. R. *et al.* Alternative activation generates IL-10 producing type 2 innate lymphoid cells. *Nat Commun* **8**, 1900 (2017).
8. Howard, E. *et al.* IL-10 production by ILC2s requires Blimp-1 and cMaf, modulates cellular metabolism, and ameliorates airway hyperreactivity. *J Allergy Clin Immunol* **147**, 1281-1295.e5 (2021).

9. Ren, Z. *et al.* G protein coupled receptor 41 regulates fibroblast activation in pulmonary fibrosis via Gai/o and downstream Smad2/3 and ERK1/2 phosphorylation. *Pharmacol Res* **191**, 106754 (2023).
10. Li, J. *et al.* GPR41 deficiency aggravates type 1 diabetes in streptozotocin-treated mice by promoting dendritic cell maturation. *Acta Pharmacol Sin* **45**, 1466–1476 (2024).

Response to Reviewers – Round 2
Rusznak *et al.*

We greatly appreciate the opportunity to submit these final revisions. We thank the reviewers for their thoughtful recommendations and careful attention to our manuscript.

Reviewer #1 (Remarks to the Author):

The content has been significantly improved in the revised version. It can be easily read and the importance of FFAR3 in the anti-inflammatory reprogramming of ILC2 has been clearly described.

We greatly appreciate the reviewer's assessment that the revisions have significantly improved the manuscript.

I want the authors to correct one point. The potential physiological role of functional regulation of ILC2 by FFAR3 has not been mentioned in the present version. I assume that anti-inflammatory regulation by the short-chain fatty acid/FFAR3 system may be equipped to dampen inflammation and/or that microbes may secrete FFAR3 ligands to escape from the host defense system via FFAR3. The authors should mention the possibility quoting appropriate reference(s).

We agree with the reviewer's assessment that that manuscript would benefit from exploring the possible physiologic role of an SCFA-FFAR3 axis relating to ILC2s. We have added the following text to the discussion:

“This raises the question of what may be the physiological role of a signaling axis that makes ILC2 effector states responsive to microbe-derived SCFAs. We speculate that this may be a way in which ILC2s, an integral part of the innate Type 2 inflammatory response that evolved to combat helminthic infections, may tune their inflammatory readiness based on the state of the microbiome. Many helminths, such as *Ascaris lumbricoides* have life cycles that involve the gut and the lung, and infections are known to perturb the microbiome and modulate SCFA levels.^{83,84} One could imagine the SCFA-FFAR3 axis acting as an early warning system: in settings of microbiome eubiosis, high SCFAs levels shift ILC2s to an anti-inflammatory or maintenance effector state, but helminthic infection and subsequent microbiome dysbiosis may drop SCFA levels and shift pulmonary ILC2s to a pro-inflammatory state of high readiness where they are primed to respond in the event of systemic infection.”

Reviewer #2 (Remarks to the Author):

The authors have responded positively to the prior reviews and have addressed my concerns satisfactorily.

We thank the reviewer for the positive comments on our revision.

Minor comments

Please add the %variance to the x and y axis labels in Fig 6L

We have added % variance to the x and y axis labels in Figure 6L.

Reviewer #3 (Remarks to the Author):

I greatly appreciate the effort the research team put forward to address prior concerns. The revisions have been thoughtful and fully address prior comments. I believe the manuscript is now suitable for publication.

We are grateful that the reviewer found our revisions to be thoughtful and comprehensive.

I have only two minor requests:

1. Please include Response to Reviewer Figure 4, either within the Methods section or as a Supplemental Figure, and add one or two sentences in the main text describing these data, as they are important to document.

We have now included Response to Reviewers Figure 4 as a supplemental figure. We agree that highlighting the differences short-chain fatty acid levels between strains is important to the manuscript. It has been added as Supplemental Figure 9, and it has been referenced in the main text as below:

“Our results suggested that differences in baseline FFAR3 signaling on ILC2s between C57BL/6J and CC030 mice were attributable to differences in receptor expression, but we aimed to investigate possible in vivo differences in SCFAs, the endogenous FFAR3 ligands, as well. We quantified acetate, propionate, and butyrate in the serum of C57BL/6J and CC030 mice at baseline via mass spectrometry, and CC030 mice had significantly lower serum concentrations of acetate compared to C57BL/6J, while there were no differences in propionate or butyrate (Supplementary Figure 9). The lower concentration of FFAR3 ligands in the serum of CC030 mice suggested that the increased FFAR3 signaling on CC030 ILC2s is dependent on increased receptor expression, not SCFA concentration.”

2. I would suggest tempering the heading “FFAR3 signaling reprograms human ILC2s towards an anti-inflammatory effector state in vitro”, to “FFAR3 signaling partially reprograms human ILC2s towards an anti-inflammatory effector state in vitro”, to not overstate the conclusion.

We agree with the reviewer that the language in the headings should be tempered to reflect the partial nature of the conservation between mouse and human ILC2s with respect to FFAR3's effects. We have amended the text to read “partially reprograms” in the results heading and Figure 8 legend.